# Coarse mode mineral dust size distributions, composition and optical properties from AER-D aircraft measurements over the Tropical Eastern Atlantic

Claire L. Ryder[1], Franco Marenco[2], Jennifer K. Brooke[2], Victor Estelles[3], Richard Cotton[2], Paola Formenti[4], James B. McQuaid[5], Hannah C. Price[6], Dantong Liu[7], Patrick Ausset[4], Phil D. Rosenberg[5,8], Jonathan W. Taylor[7], Tom Choularton[7], Keith Bower[7], Hugh Coe[7], Martin Gallagher[7], Jonathan Crosier[7,9], Gary Lloyd[7], Eleanor J. Highwood[1], Benjamin J. Murray[5].

1 Department of Meteorology, University of Reading, Reading, RG6 6BB, UK
2 Met Office, Exeter, UK
3 Department de Física de la Terra i Termodinàmica, Universitat de València,46100 Burjassot, Spain
4 LISA, UMR CNRS 7583/Université Paris Est Créteil et Université Paris Diderot, Institut Pierre Simon Laplace, Créteil, France
5 School of Earth and Environment, University of Leeds, Woodhouse Lane, LS2 9JT, UK
6 Facility for Atmospheric Airborne Measurements, Cranfield, UK
7 Centre for Atmospheric Sciences, School of Earth and Environmental Sciences, University of Manchester, Manchester, UK
8 National Centre for Atmospheric Science, Fairbairn House, 71-75 Clarendon Road, Leeds, LS2 9PH, UK
9 National Centre for Atmospheric Science, University of Manchester, Manchester, UK

*Correspondence to*: Claire Ryder (c.l.ryder@reading.ac.uk)

## Abstract

Mineral dust is an important component of the climate system, affecting the radiation balance, cloud properties, biogeochemical cycles, regional circulation and precipitation, as well as having negative effects on aviation, solar energy generation and human health. Dust size and composition has an impact on all these processes. However, changes in dust size distribution and composition during transport, particularly for coarse particles, are poorly understood and poorly represented in climate models. Here we present new in-situ airborne observations of dust in the Saharan Air Layer (SAL) and the Marine Boundary Layer (MBL) at the beginning of its trans-Atlantic transport pathway, from the AER-D fieldwork in August 2015, within the peak season of North African dust export. This study focuses on coarse mode dust properties, including size distribution, mass loading, shape, composition, refractive indices and optical properties. Size distributions from 0.1 to 100 $\mu$m diameter (d) are presented, fully incorporating the coarse and giant modes of dust. Within the MBL, mean effective diameter ($d_{eff}$) and volume median diameter (VMD) were 4.6 $\mu$m and 6.0 $\mu$m respectively, giant particles with a mode at 20-30 $\mu$m were observed, and composition was dominated by quartz and alumino-silicates at d > 1 $\mu$m. Within the SAL, particles larger than 20 $\mu$m diameter were always present up to 5km altitude, in concentrations over $10^{-5}$ cm$^{-3}$, constituting up to 40% of total dust mass. Mean $d_{eff}$ and VMD were 4.0 $\mu$m and 5.5 $\mu$m respectively. Larger particles were detected in the SAL than can be explained by sedimentation theory alone. Coarse mode composition was dominated by quartz and alumino-silicates; the

accumulation mode showed a strong contribution from sulfate-rich and sea salt particles. In the SAL, measured single scattering albedos (SSAs) at 550nm representing d<2.5 µm were 0.93 to 0.98 (mean 0.97). Optical properties calculated for the full size distribution (0.1<d<100 µm) resulted in lower SSAs of 0.91-0.98 (mean 0.95) and mass extinction coefficients of 0.27-0.35 $m^2g^{-1}$ (mean 0.32 $m^2g^{-1}$). Variability in SSA was mainly controlled by variability in dust composition (principally iron), rather than by variations in the size distribution, in contrast to previous observations over the Sahara where size is the dominant influence. It is important that models are able to capture the variability and evolution of both dust composition and size distribution with transport in order to accurately represent the impacts of dust on climate. These results provide a new SAL dust dataset, fully representing coarse and giant particles, to aid model validation and development.

## 1    Introduction

Mineral dust plays an important role in the Earth's climate system. It is the most dominant aerosol species in the atmosphere, constituting 70% of the global aerosol mass burden and 25% of the aerosol optical depth (AOD) (Kinne et al., 2006). Once uplifted from arid regions, dust is transported thousands of kilometres across the Atlantic Ocean from the Sahara desert (Carlson, 2016), and across the Pacific Ocean from eastern Asian deserts (Li et al., 2010). The abundance and long-distance transport of mineral dust allow it to affect the climate system via different processes. Dust interacts with both solar and infrared radiation, exerting a direct radiative effect at the top of atmosphere and surface, which can alter atmospheric heating rates and stability, surface fluxes and temperatures, and thus influence regional circulation and precipitation (Lavaysse et al., 2011; Strong et al., 2018). Dust particles may influence cloud development on a microphysical level by acting as both cloud condensation nuclei and ice nuclei (Kumar et al., 2011; Hoose and Mohler, 2012), affecting cloud optical properties and lifetimes. Finally, dust has an impact on biogeochemical cycles since it provides iron to the ocean (Jickells et al., 2005) and phosphorous to the Amazon rainforest (Yu et al., 2015), which can lead to subsequent changes in atmospheric carbon dioxide absorption by the oceans and vegetation and associated climate feedbacks. Dust impacts anthropogenic activities by depleting solar radiation available for solar energy generation, both in the atmosphere (Charabi and Gastli, 2012) and by deposition to solar panels (Piedra and Moosmuller, 2017). It also negatively impacts aviation in dust laden regions due to a reduction in visibility (Middleton, 2017), and finally high dust loadings can negatively impact respiratory health (Prospero et al., 2014). Many of these processes are dependent not only on total mass of dust, but also its size (Mahowald et al., 2014).

Recent studies have revealed how poorly climate models are able to simulate uplift and transport of dust. Evan et al. (2014) find that CMIP5 climate models underestimate dust mass path (the mass loading of dust per square metre) by a factor of three, 66% of which is due to a bias in size distribution skewed towards smaller particles, and 34% of which is due to an underestimate in emission fluxes. As a result, these models systematically failed to reproduce basic aspects of dust emission and transport, casting doubt on their ability to simulate regional African climate and the response of African dust to future climate change. Kok et al. (2017) used an observationally constrained emitted dust size distribution in combination with global model

simulations to determine dust radiative forcing. They estimated a more positive radiative forcing (-0.48 to +0.20 Wm$^{-2}$) compared to previous estimates from the AeroCom models (-0.6 to -0.3 Wm$^{-2}$) which over represented smaller, more cooling particles, and under represented the coarser, more warming particles. Kim et al. (2014) compared AeroCom models to satellite data, and found disagreement in dust optical depth of up to a factor of 4, also finding that mass extinction coefficient (MEC, which is sensitive to particle size distribution) varied by 27% between models. These studies emphasize the sensitivity of model predictions of key parts of the climate system to representation of particle size, and the challenges of capturing observations in current climate models.

Coarse and giant mode dust (defined here as d>2.5 µm and d>20 µm, respectively) is of particular importance to its interaction with radiation. In the shortwave (SW) spectrum, larger particles (assuming fixed composition, shape and roughness) increase the amount of atmospheric absorption, thus decreasing the single scattering albedo (SSA or $\omega_0$) (Tegen and Lacis, 1996). For example, calculations have shown that including coarse and giant mode particles measured over the Sahara results in the SSA dropping from 0.92 to 0.80 at 500 nm. This resulted in the associated atmospheric heating rates increasing by up to a factor of three (Ryder et al., 2013b). In the longwave (LW) spectrum, coarse particles are equally important. Otto et al. (2011) show that including particles larger than 5 µm diameter more than doubles the dust LW AOD; and the magnitude of the LW radiative effect can act to change the sign of the net radiative effect of dust (Woodage and Woodward, 2014). Song et al. (2018) show that dust radiative closure assessed by satellite observations in both the shortwave and longwave spectra can only be achieved with a substantial coarse mode dust size distribution.

Part of the challenge in modelling the dust cycle is that it is not currently clear even from observations what all the mechanisms for the transport of coarse particles are. In theory, dry deposition depends on particle size, but is also influenced by particle shape, density and roughness (Li and Osada, 2007). However, observations have consistently shown that coarse dust particles are transported further than predicted by dry deposition theory (Maring et al., 2003; Ryder et al., 2013a; Weinzierl et al., 2017; Gasteiger et al., 2017; Denjean et al., 2016; Stevenson et al., 2015). Various suggestions as to the observed retention of the coarse mode during transport have been proposed, including solar absorption by dust generating convection and therefore additional vertical mixing in dust layers (e.g. Gasteiger et al. (2017)), turbulence within dust layers (Denjean et al., 2016), and electrostatic charging of dust (Harrison et al., 2018). Measuring and quantifying these processes remains challenging. In order to do so, high quality observations of dust properties at multiple stages throughout transport events are required.

Over the last 15 years, aircraft observations have made significant advances both in observing dust in increasingly remote regions of the Sahara near dust sources and in utilizing instrumentation to characterize the full dust size distribution. The major fieldwork campaigns since 2005 are shown in Table 1, including the measurement technique and maximum size measured. Size distributions can be measured inside aircraft cabins behind inlets, although this introduces restrictions enforced by inlet dependent size cuts which prevent the measurement of the coarser particles to varying degrees. In recent years, wing-mounted

sizing probes have been more routinely operated, removing any inlet sizing restrictions. Wing-probes have employed both light scattering and light shadowing techniques. Optical Particle Counters (OPCs) measure light scattering, and post-processing requires converting scattering cross section to particle size. This relationship, although dominated by particle size, is also impacted by aerosol composition (via the refractive index), and the scattering cross-section to particle size relationship is non-monotonic. Non-spherical particles may also impact the retrieved size distribution. These limitations, impacts and uncertainties are discussed in detail by Ryder et al. (2015). Optical Array Probes (OAPs) can be wing-mounted and utilize light shadowing techniques for particle sizing, and are not subject to the above uncertainties, and therefore present a valuable method for sizing of coarse dust particles, as demonstrated by (Ryder et al., 2013b), although sizing can still be subject to uncertainties such as from particle shape. All wing probes can also potentially suffer from modification of the flow around the probe housing and particle bouncing from the probe tips (Korolev et al., 2013; Weigel et al., 2016; McFarquhar et al., 2017).

The progressive airborne measurement of larger dust particle sizes has demonstrated the prevalence of coarse and giant particles both over desert regions and far from sources. For example, Weinzierl et al. (2009) detected particles larger than 40 µm in 20% of cases over Morocco during SAMUM1, even up to 5 km altitude. During Fennec, dust particles sized over 100 µm were measured at altitudes up to 5 km over Mali and Mauritania (Ryder et al., 2013b). During SAMUM2 over the Atlantic Ocean, although dust particles sized over 10 µm were present in 88% of cases, no particles sized over 30 µm were detected (Weinzierl et al., 2011). However, these observations were performed within the low altitude wintertime Saharan dust plume (under 1.5 km), whereas the summertime elevated dust plume within the Saharan Air Layer (SAL) is subject to different meteorological and dust uplift and transport mechanisms (Prospero and Carlson, 1972; Karyampudi et al., 1999; McConnell et al., 2008).

Here we present new airborne measurements of aerosol in the tropical Eastern Atlantic region, obtained during August 2015 as part of the AERosol Properties – Dust (AER-D) fieldwork campaign, which ran alongside the Ice in Clouds Experiment – Dust (ICE-D) project. We utilize size distribution instrumentation aboard the UK FAAM BAe146 research aircraft consistent with the Fennec campaigns of 2011 and 2012 (Ryder et al., 2015) and present dust properties measured during summer, the peak North African dust transport season (Doherty et al., 2008), with a particular focus on the properties of the coarse mode. New real time measurements of accumulation mode hematite content measured during several of the same flights, and their optical properties, are presented separately by Liu et al. (2018). Ice nucleating properties of the dust sampled are given by Price et al. (2018), and vertical structure is analysed by Marenco et al. (2018, in review).

## 2 Method

### 2.1 Flight Patterns

During August 2015, the UK BAe-146-301 Research Aircraft operated by the Facility for Airborne Atmospheric Measurements was stationed at Praia on the Cape Verde Islands, for the ICE-D and AER-D field campaigns. The AER-D project comprised six flights, focusing specifically on dust properties within and beneath the SAL. Each flight was performed 4 to 5 days apart, due to the transported dust in the SAL displaying intermittent character, typical for summertime dust export (Jones et al., 2003; Schepanski et al., 2017). Details of the flights are shown in Table 2 and flight tracks shown in Figure 1.

The events sampled revealed accumulation mode AODs at 550 nm (see Section 2.3.1) from 0.4 to 0.8, which is within the range expected from August climatology over the Eastern tropical Atlantic indicated by satellite imagery (not shown). The aircraft observations revealed mostly typical vertical dust structure (see Section 3.1). The exceptions were b923 and b924, a pair of flights which observed an intense dust outbreak with AODs of up to 2.5 at around 24N off the coast of West Africa, with in-situ and remote sensing observations showing a different and complex vertical structure compared to the conventional SAL model of an elevated dust layer (discussed in Marenco et al. (2018, in review).

Two flights (b928, b934) formed part of the Sunphotometer Airborne Validation Experiment in AER-D (SAVEX-D) project, focused on providing airborne measurements for comparison with two types of ground-based sunphotometer. Requirements for non-cloudy skies and moderate to high dust loadings, and proximity to the ground-based sunphotometers required that the first SAVEX-D flight (b928) took place close to Praia, while the second (b934) took place close to the island of Sal. Three flights (b923, b924 and b932) were aimed mainly at mapping the vertical and horizontal aerosol structure. These flights headed to the northeast in order to encounter heavy dust loads closer to the African coast on 12 August. Flight b920 was conducted near to the Cape Verde archipelago and co-located with the path of the International Space Station (ISS), in order to fully characterise the SAL and validate the Cloud-Aerosol Transport System (CATS) remote sensing lidar instrument onboard the ISS.

Each flight consists of a combination of sloped profiles (abbreviated to 'P', e.g. 'P1') and straight-and-level runs (known as 'SLRs' and abbreviated to 'R', e.g. 'R2') at various altitudes, generally selected to be within the aerosol layer of interest, or at altitudes appropriate for radiometric measurements. Here we present results from 31 profiles and 19 in situ aerosol sampling SLRs, as shown in Figure 1. Five SLRs were performed in the marine boundary layer (MBL) at 30 to 35 m above sea level (one per flight), and 14 SLRs sampled the SAL at altitudes between 1.8 to 4.1 km. Exceptions were an intermediate layer sampled during flight b920 containing dust at 1.2 km, and heavy dust sampled during flight b924 at 920 m. Full information about profile and SLR times and altitude are available from the Centre for Environmental Data Analysis (see Data Availability).

## 2.2 Determination of Dust Sources and Dust Age

Broad geographic dust source locations, where dust was uplifted before being sampled by the aircraft during AER-D, have been identified using Spinning Enhanced Visible and Infrared Imager (SEVIRI) 'dust RGB' (red-green-blue) thermal infrared satellite imagery product (Lensky and Rosenfeld, 2008), where dust events are identifiable by their bright pink colour (e.g. Brindley et al. (2012)). Dust events sampled by the aircraft are tracked backwards in time visually until uplift times and locations are identified, identical to Ryder et al. (2013b). The high temporal resolution (15 min) of the imagery enables easier tracking compared to other spaceborne sensors which may have only two overpasses per day. However, this method is subjective, and therefore we allow generous errors in terms of geographic dust uplift location, and dust uplift time. Additionally, no height resolution information is available from the imagery, so source areas are categorized for each flight, and not for each sampling altitude within flights.

Examination of the dust events sampled during AER-D reveals that the dust in every event was initially uplifted by a mesoscale convective system (MCS) and a resultant cold pool (haboob) which spread out radially. This is in keeping with recent findings that cold pools are the dominant mechanism for summertime Saharan dust uplift (Marsham et al., 2013; Allen et al., 2013). The dust events then took 1 to 5 days to reach the Atlantic Ocean where they were sampled by the aircraft. In three cases, additional uplift events could be observed merging with the initial dust event, and in these cases error bounds on dust age were set to include all possible uplift events, and were therefore very broad. In most cases, convection and clouds could be observed developing over the SAL during the afternoon in the top of the dust layers during their transport. Therefore it is possible that the dust observed has been processed by clouds (e.g. Ryder et al. (2015) Section 4.1.4; Diaz-Hernandez and Sanchez-Navas (2016)), though no clouds were present when sampling took place.

The SEVIRI imagery is not able to give altitude-resolved information, and can be subjective, particularly when dust loadings are light, at low altitude or in a moist environment, making dust appear less pink and more difficult to identify (Brindley et al., 2012). This is more evident in the dust tracked for flights b932 and b934 where dust loadings were lower. This introduces a small level of uncertainty into both the source locations and dust ages, which we account for by giving generous error bars to the dust uplift times and source locations. HYSPLIT back trajectories (Draxler and Hess, 1998; Stein et al., 2015) were also run for the AER-D dust events. In only one of the five dust events was the dust source location similar to that observed in the SEVIRI imagery. In every case the back trajectories indicated a transport path and transport time different to that shown by the SEVIRI imagery. Although the SEVIRI methodology has its limitations, the back trajectory method results were clearly not compatible with the information from SEVIRI. Therefore back trajectories are not used to determine source location or age here. Additionally, another limitation of back trajectories is that they only indicate when an air mass nears the surface, but do not reflect potential uplift conditions (e.g. surface wind strength or soil conditions). It has been shown that models and reanalyses are currently unable to adequately represent convective events and winds over the Sahara and Sahel, particularly

due to the challenges of representing cold pools. For example, Garcia-Carreras et al. (2013) examine the role of convective cold pools and suggest that "the misrepresentation of moist convective processes can affect continental-scale biases, altering the West African monsoon circulation." Many other publications have examined the misrepresentation of Saharan convective events (Marsham et al., 2011; Heinold et al., 2009; Sodemann et al., 2015; Trzeciak et al., 2017; Allen et al., 2015; Roberts et al., 2017; Engelstaedter et al., 2015). Since convective events are the drivers of dust uplift in all the AER-D cases we do not consider HYSPLIT back trajectories (with relatively low model resolution of half a degree) to be informative here due to the challenges the models face in representing Saharan circulation. Finally, we note that back trajectories are recommended to be used with caution for dust events over the summertime Sahara (Trzeciak et al., 2017).

### 2.3 Instrumentation

Much of the instrumentation operated during ICE-D was identical to that operated during the Fennec campaign, described in Ryder et al. (2013b) and Ryder et al. (2015). Relevant information is provided below, noting where instruments and processing differed. Throughout this article all particle sizes are referred to in terms of diameter (d), and optical properties are presented at 550 nm unless stated otherwise.

#### 2.3.1 Inlets, Scattering and Absorption Measurements

Scattering measurements were made by a TSI 3563 integrating nephelometer (at wavelengths of 450, 550 and 700 nm). Absorption measurements were made by a Radiance Research Particle Soot Absorption Photometer (PSAP) at 567 nm. Both instruments are situated inside the aircraft cabin, behind a modified Rosemount 102E inlet. PSAP filters were changed before every flight and spot size was measured. Standard corrections are performed on both instruments. For the PSAP these are taken from Turnbull (2010), which incorporates corrections necessary to the FAAM PSAP measurements based on the original work by Bond et al. (1999), and further clarifications to this publication described by Ogren (2010). For the nephelometer corrections are performed according to Anderson and Ogren (1998) assuming supermicron particles. This results in 11% and 15% uncertainty in extinction and single scattering albedo (SSA) respectively (Ryder et al., 2013b). Corrections for internal nephelometer temperature and pressure are included.

Accumulation mode AODs are calculated from aircraft profiles by integrating the scattering and absorption measurements between the minimum aircraft altitude (typically around 30 m above sea level) to the top of the profile (typically around 6 km). Therefore AODs represent both SAL and MBL aerosol.

Extensive experimental and theoretical efforts to characterize inlet and pipe losses were performed during Fennec (Trembath, 2012; Ryder et al., 2013b), and we apply those results here. It was found that the net impact of both Rosemount inlets and pipework supplying the nephelometer and PSAP led to number concentration enhancements by a factor of 1.5 for d<1.5µm, no net loss or enhancement at 2.5µm, net losses between 2.5 to 5 µm (50% efficiency at 3.5±0.5 µm), and no particles larger

than 5 µm sampled. The exact enhancement and losses vary between the nephelometer and PSAP (slightly different pipework) and with altitude (see Supplement Figure S1). Since it is clear that coarse mode particles do not reach the nephelometer and PSAP, henceforward the term 'accumulation mode' is used to describe the size distribution arriving at these two instruments, roughly representing particles sized d<2.5 µm. The enhancement and loss factors as a function of diameter are applied to the

size distribution described in Section 2.3.2 in order to replicate the accumulation mode size distributions reaching the nephelometer and PSAP.

### 2.3.2 Size Distribution Measurements

We present size distribution measurements from a combination of different wing-mounted instruments as shown in Table 3. We utilize optical particle counter techniques for the accumulation and coarse modes (Passive Cavity Aerosol Spectrometer

Probe (PCASP) and Cloud Droplet Probe (CDP)), and light shadowing measurement techniques for the giant mode (Two-Dimensional Stereo Probe (2D-S), Cloud Imaging Probe 15 (CIP15), Cloud Imaging Probe 100 (CIP100)). Size distributions are also provided for some SLRs from filter sample analysis (Section 2.4). This range of instrumentation allows us to take advantage of measuring a wide particle size range (0.1 to 6200 µm), though the processing and uncertainties associated with each must be considered carefully, as follows. Uncertainties due to flow distortion effects are not considered.

The PCASP was calibrated before and after the campaign, and the CDP was calibrated and cleaned before most flights during the campaign as described in Rosenberg et al. (2012). The PCASP and CDP employ light scattering in order to determine particle size, thus the nominal size bins have been adjusted for a refractive index (RI) of dust of 1.53 – 0.001i, informed by the refractive index closure results from Section 2.5, using the CStoDConverter software (Rosenberg et al., 2012). This results in

an increase in diameter of the largest size bins due to the more absorbing imaginary part of the refractive index as compared to PSL (polystyrene latex) used in the manufacturer's calibrations. The software also provides uncertainties in the bin centres and bin widths due to oscillations in the Mie scattering curve, which can cause ambiguities in the scattering–size relationship. This propagation of errors allows us to utilize data from the full size range while still maintaining a measure of the sizing uncertainty, and can be seen by horizontal error bars in Figure 2. The larger errors around diameters of 3.5 and 20 microns, for

example, relate to inflection points on the Mie scattering curves. These horizontal sizing errors are propagated through to calculations of number and volume size distribution errors. The first size bins are removed for both the PCASP and CDP since the lower edges are not well defined.

In a similar manner to Fennec, custom bin widths were set for the CDP. The smallest bin of the CDP was set much wider than

standard, with a lower minimum detection threshold in order to increase sensitivity to smaller particles, and subsequent processing of the particle-by-particle data allowed the smallest size bin to be split into 4 sub-bins. As a result, the bin sizes of the CDP increase due to the refractive index correction, but the size resolution at the small end of the spectrum is retained due to the custom bin specifications. This prevents too much of a gap developing between the PCASP and CDP size ranges, with

the first 4 size bins covering 3.36-3.87, 3.76-4.18, 4.35-4.81, 4.74-5.36 µm. Note that the bin sizes overlap due to the method used to determine bin edges in CStoDConverter.

Bin sizes also depend on the choice of refractive index applied. In this work, a complex refractive index ($n^{550} = m^{550} - ik^{550}$) of
1.53-0.001i was used to determine the PCASP and CDP bin sizes, as determined from Section 2.5, for 550 nm. Since the PCASP and CDP operate at wavelengths of 633 and 658 nm, we assume a constant refractive index across these wavelengths. This is supported by the relatively flat spectral refractive index shape at these wavelengths indicated in Figure S2. As additional sensitivity tests, size distributions were also computed using a real part, $m^{550}$, of 1.48 and 1.58 (1.53±0.05), and imaginary parts, $k^{550}$, of 0.002, 0.003 and 0.006. Particle sizes determined from the PCASP and CDP represent optically equivalent
diameters. Sensitivity in effective diameter ($d_{eff}$) to these changes from $m^{550}$ were less than 0.2 µm (<5%), and from changes in $k^{550}$ were less than 0.9 µm (<20%). However, within the likely range of $k^{550}$ (0.001 − 0.002) as derived in Section 2.5, changes in $d_{eff}$ were < 0.1 µm (<3%). Therefore we consider the uncertainty in $d_{eff}$ to refractive index to be a maximum of 5% reflecting the uncertainty due to likely values of both m and k.

The Optical Array Probes (OAPs: CIP15, CIP100, 2DS) measure particles in two perpendicular directions: the first aligned
with the photodiode array (x) and the second along the direction of aircraft motion (y), Thus providing a two-dimensional projection of a particle (McFarquhar et al., 2017). Although the OAPs do not require assumptions about refractive index to derive size, they can be subject to several systematic uncertainties associated with the processing/sizing method (McFarquhar et al., 2017). Thus to investigate some of these uncertainties, the 2DS data was processed in two different ways, using two different sizing metrics.

Firstly, data from all three OAPs were processed using the System for OAP Data Analysis (SODA-2) developed at NCAR. The SODA-2 removes out of focus images and various instrument artefacts including accounting for a stuck diode in the middle of the CIP-100 array. A 'centre-in' method was applied where particles are only counted when the centre of the particle falls within the photodiode array. The particle size was defined as the diameter of the smallest circle enclosing the particle (denoted 'CC' – circumscribing circle), and thus for non-spherical particles may lead to an overestimate in particle size.
Secondly, data from the 2DS instrument were analysed using the Optical Array Shadow Imaging Software (OASIS), developed by the National Centre for Atmospheric Science (NCAS) and DMT. Particle sizes were calculated using the mean of the x and y dimensions of each particle image (denoted 'XY') using an 'all-in' method where particles are only counted when they fall completely within the photodiode array. The x and y dimensions are measured along the probe array, i.e. the particle is not rotated to minimize or maximize either dimension.

The mean XY method is considered to give a more representative diameter for non-spherical particles than the CC metric. Area-equivalent diameters were not calculated because particles can sometimes appear hollow on the OAPs (McFarquhar et al., 2017) which would lead to undersizing. If the particle image is an ellipse, the mean XY diameter will be larger than an

ellipse area-equivalent diameter, as used by the filter sample analysis for example. However, the OAP images capture 2-D image projections of the particles in their atmospheric orientation, while the filter samples are will be collected with their largest surface lying parallel to the filter sample, and therefore may be oversized in this context. An additional motivation for testing this second processing option for the 2DS was for comparison with Fennec data (Ryder et al., 2013b) which were
processed using an all-in, mean XY method for the CIP15.

Since particles detected by the 2DS during AER-D mostly cover 1-7 pixels, the impact of centre-in versus all-in is considered small. The sample area is adjusted for the effective array width, which is different depending on whether 'all-in' or 'centre-in' is used (McFarquhar et al., 2017), and therefore the calculated number concentrations account for this. The sizing metric (i.e. XY versus CC) has the greatest impact on the final size distribution. Smallest and largest size bins were removed for the OAP
data, and data was excluded for size bins in SLRs where fewer than 4 particles were detected as an additional measure for removing noise (equating to number concentrations of around $10^{-5}$cm$^{-3}$ for the 2DS and CIP15 for typical sampling times of around 20 minutes, or 132 km on the BAe146). The CIP100 suffered from noise and did not detect any particles within its size range (d>200 µm following exclusion of the first size bin).

Uncertainties in number concentration for all size probes are propagated from 1Hz measurements, through to means over SLRs, through to the AER-D campaign averages. For all probes, random errors (due to counting and discretization error) and systematic errors (due to sample area uncertainty and bin size centre and width from Mie singularities) were accounted for in their contribution to total number concentration errors, and propagated by standard analytical error propagation. I.e. random error can be minimized by increasing the sample size (averaging across the campaign), while systematic error remains constant.
For the CDP at d < 20 µm bin size uncertainty was found to dominate the total uncertainty in dN/dlogD and dV/dlogD. Horizontal error bars in Figure 2 represent the maximum uncertainty in bin edges, derived from uncertainties in both bin centre and bin width. Uncertainties in bin size contribute to uncertainties in both dN/dlogD and dV/dlogD, and therefore the relative uncertainties do not change significantly between the two panels. All error bars represent maximum uncertainty.

Figure 2 shows an instrument comparison for the five instruments for the mean size distribution in the SAL. It can be seen that the CIP15 and 2DS cover similar size ranges, and the CDP overlaps both of these instruments at the smallest size bins. However, different number concentrations are detected by the CIP15 and 2DS, which becomes more evident as size approaches 100 µm. Greater number and volume distributions are seen from the CIP15 than the 2DS, although when the same processing and sizing assumptions are used (CIP15 CC, blue, and 2DS CC, orange), the size distributions agree within error bars. The
2DS XY processing (green) results in a lower number distribution and a smaller maximum size, as expected for non-spherical dust particles (section 3.2) because the CC metric will oversize a non-spherical particle. Therefore several choices are made in terms of how best to use data from each instrument in the analysis of size distributions and subsequent optical properties, as follows. We use the full size range of the PCASP for the accumulation mode, for the coarse mode the CDP is used up to 20

μm, and the giant mode is taken from the 2DS XY processing and metric since this is a better metric for non-spherical dust particles, and also consistent with the Fennec data processing. This size distribution is referred to as 'FULL PSD.'

Size distributions are summarized with commonly used metrics of effective diameter, $d_{eff}$, (Hansen and Travis, 1974) (sometimes known as the volume-surface diameter, Hinds (1999)), and volume median diameter (VMD, the diameter below which half of the volume size distribution lies, (Seinfeld and Pandis, 2006)). Additionally we present the maximum size detected, $d_{max}$, by the 2DS XY as a useful metric of the largest size present in the atmosphere. Weinzierl et al. (2009; 2011) present maximum size measured at a concentration above $10^{-2} cm^{-3}$. Here we choose to simply present the maximum particle size detected, since the number concentration detected depends on the width of the size bin employed by the particular instrument operated and is not directly comparable. The sizing instruments operated during AER-D do not have a minimum detection concentration level, but at low particle concentrations the sampling statistics simply become poor, introducing an effective detection limit. Therefore we remove cases where fewer than 4 particles are detected over a SLR as previously mentioned, with an implicit effect of removing data where concentrations are lower than $10^{-5} cm^{-3}$ for the 2DS and CIP15.

Some CIP15 AER-D data were also processed using a centre-in, mean XY metric, but unfortunately it was not possible to process data for all the SLRs with this method. Therefore this data was used to inform on instrumental differences between the 2DS and CIP15 when processed with the same size metric (XY mean). It was found that the impact on the full PSD was very small ($d_{eff}$ differed by under 1%), but that $d_{max}$ was up to 6% larger with the CIP15 XY compared to the 2DS XY. The upper uncertainty of 6% in $d_{max}$ was therefore propagated in combination with the other uncertainties in $d_{max}$.

### 2.3.3 Size Distribution behind Rosemount Inlets

In addition to the main size distribution combination described above, a further size distribution is calculated in order to represent the size distribution reaching the nephelometer and PSAP, accounting for inlet enhancements and losses as well as pipe losses (Trembath, 2012; Ryder et al., 2013b). In order to do this, the 'FULL PSD' is adjusted for the size-resolved, height dependent, loss or enhancement as shown in Supplement Figure S1. Since particles larger than 5 μm diameter do not reach the nephelometer or PSAP, the choice of instrument at sizes larger than this is irrelevant. There is a net loss of particles at d > 2-3 μm and an enhancement of the number of particles in the accumulation mode at d < 2-3 μm. We term this size distribution 'ACC PSD' since it predominantly represents the accumulation mode.

### 2.3.4 Filter Sample Collection

Aerosol filter samples were collected using the FAAM airborne filter collection system (Andreae et al., 2000; Formenti et al., 2008; McConnell et al., 2010) behind a separate inlet dedicated to filter samples (i.e. different to the Rosemount inlets). This is the same inlet used to collect ice nuclei particle samples presented in Price et al. (2018), in many cases samples were collected in parallel. The inlet passing efficiency has not been formally determined, though Andreae et al. (2000) found that while

mounted on the previous C-130 aircraft, the same filters inlet restricted measurements to 35% of the coarse mode mass (defined as d>1.4 µm) in sea salt aerosol compared to ground-based observations. Further details are available in Price et al. (2018).

Aerosol particles were sampled by filtration onto a stacked-filter unit (SFU) with three stages, though only one stage was used in AER-D. Aerosols were collected on 47mm diameter Nucleopore filters with pore sizes of 0.2 or 0.4 µm using either a standard plastic filter holder as used previously (e.g. Formenti et al. (2008)) or on a newly developed adjusted plastic filter holder with a lip on the edge removed for ease of loading. Depending on the holder and pore size used, different support grids were used – either plastic, Millipore fine mesh metal, or JSHoldings coarse mesh metal. In total 22 filter samples were collected during AER-D flights, although problems were encountered with many samples ranging from difficulties mounting and removing the support grids into the old filter holder, very low flow rates when combining the 0.2 µm Nucleopore pore sizes and Millipore grid, and unexplained silicon contamination on some samples. As a result, many of the filter samples did not display the expected number of particles upon laboratory analysis. Therefore we were forced to discard many samples, and four samples of the best data were analysed for size, shape and chemical composition analysis: 2 SLRs in the MBL (b920 R2, b928 R2) and 2 SLRs in the SAL (b920 R5, b932 R6), in order to give a snapshot of these properties during AER-D.

### 2.4 Filter Sample Analysis

Filter samples were analysed by Scanning Electron Microscopy (SEM, instrument model JEOL JSM 6301F) coupled to an X-ray energy-dispersive spectrometer (Silicon Drift X-Max 80 mm$^2$ Detector and Aztec Advanced-INCA350 analyzer, Oxford Instruments) to provide information on the coarse and fine fractions respectively. Analysis was performed on a portion of filter cut and mounted on an aluminum stub using a double-sided adhesive and then covered with a thin film of Platinium (Pt) by sputter coating (Jeol JFC 1100E). Particles were found to be evenly distributed across the filter sample. Images were acquired by a series of transects at two magnifications ($\times$2,000 (55.9 nm/pixel) and $\times$10,000 (11.0 nm/pixel)), scanning between 397 to 1116 particles per sample.

Particles were sized by processing the SEM images, which are essentially 2-D projections of 3-D particles. The 2-D projections were fitted with a circumscribed ellipse, to produce an ellipse area-equivalent diameter, where the major and minor axes ($d_{max}$ and $d_{min}$) are used to define the particle diameter d according to $d=(d_{max}d_{min})^{1/2}$. The aspect ratio was accordingly calculated using $d_{max}/d_{min}$.

It was not possible to use automated image contrast to calculate projected particle area because of a high degree of variability in particle contrast. Our filters sizing this technique may oversize the particle size for two reasons. Firstly, the area of a fitted ellipse may be larger than a projected particle area, though the particles were not noticeably jagged around their edges. Secondly, our method may oversize particle volume where the shape is a platy silicate and has a tendency to fall with its largest surface parallel to the substrate. For example, Chou et al. (2008) found the height of dust particles examined under SEM to be

around one third of their major axis length. Additionally, we tested the sensitivity of the filters PSD to using the mean XY and CC sizing methods applied to the OAP data (not shown). Using a mean XY method on the filters data did not produce significantly different results, while using the CC method was found to shift the PSD towards larger particles, similar to the findings from the OAP size metric comparisons.

Semi-quantitative elemental chemical analyses integrate the following elements: Na, Mg, Al, Si, P, S, Cl, K, Ca, Ti, Mn, and Fe . The results are expressed as a weight percentage of the associated oxides ($Na_2O$, $MgO$, $Al_2O_3$, $SiO_2$, $P_2O_5$, $SO_3$, Cl, $K_2O$, CaO, $TiO_2$, MnO, $Fe_2O_3$) normalized to 100%. The composition type of each particle is then identified based on these percentages into one of the following categories: alumino-silicates, quartz, sulfate, salt, thenardite, gypsum, Ca-rich, Ti-rich,

Fe-rich and others. These are interpretations based on the elementary chemical analysis. E.g. if we observe the simultaneous presence of calcium and sulphur (typically within 10% of one another), this could be several possible minerals: anhydrite ($CaSO_4$), bassanite ($CaSO_4 \cdot 0.5(H_2O)$) and gypsum ($CaSO_4$, $2H_2O$), though the most probable atmospheric form is gypsum. Sulfate is classified by the dominance of $SO_3$. Thenardite is classified when both $Na_2O$ and $SO_3$ are dominant. Salt is classified by percentages of both $Na_2O$ and Cl being high, indicating halite particles of marine origin. Alumino-silicates, quartz,

thenardite, gypsum and Fe-rich particles indicate terrigenous particles in these cases. Only one black carbon chain-like structure was observed during the analysis of over 6500 particles, and therefore this aerosol category is not included.

Although particles may frequently be a mix of several composition types, they were classified according to their component type which made up the greatest oxide percentage. This may particularly be an important assumption regarding the iron

component, which was consistently present in small amounts on most particle types but only classified as iron when iron was the dominant component; i.e. in general when iron oxide constituted over 50% of the composition. More complex methods of accounting for iron as a minor component are possible (e.g. Kandler et al. (2011a); Balkanski et al. (2007) but are beyond the scope of this work. Although composition information is available for all particles sampled under magnifications of ×2,000, many particles sampled under the high magnification did not provide a strong enough signal to provide composition data. Thus

the number of particles available for composition analysis in the fine fraction is much lower than that for the coarse fraction.

Filter sample analyses use subsets of data from each magnification in order to take advantage of the best counting statistics for each size range and optimal viewing at each magnification. Filter sample size distributions are calculated using fairly finely resolved size bins from 0.05 to 40 µm as shown in Figure 7. However, at each magnification, sensitivity to the smallest particles

detected is low due to a low SEM signal (fewer photons emitted for smaller volume particles). Therefore particles smaller than 0.1 and 0.5 µm are excluded for ×2,000 and ×10,000 respectively. Additionally, for the ×10,000 magnification, not many particles larger than 1 µm are counted in comparison to ×2,000, so particles larger than 1 µm are also excluded.

For aspect ratio, composition and refractive index analyses, particles sized 0.1 to 0.5 µm are taken from the ×10,000 magnification, while particles sized between 0.5 to 40 µm are taken from the ×2,000 magnification, and scaled appropriately to account for the different substrate area examined under the different magnifications. Size resolved composition data utilizes six size bins with edges at 0.1, 0.5, 1.0, 2.5, 5, 10 and 40 µm. Information for the full filters size distribution accordingly covers

0.1 to 40 µm. Additionally, bulk filter sample properties are calculated specifically for the accumulation mode, covering diameters 0.1 to 2.5 µm, in order to replicate the in-cabin measurements behind the Rosemount inlets, and also for the bulk sample from 0.1-40 µm to cover the full size distribution. Aspect ratios are presented as number fractions; composition is presented as volume fraction.

## 2.5 Derivation of Refractive Index

### 2.5.1 Refractive Index Iterated from Optical and Size Measurements

In order to determine the accumulation mode refractive index for each individual SLR, measurements of scattering and absorption from the nephelometer and PSAP are used in combination with Mie scattering code, taking input from the measured size distributions. Since the nephelometer and PSAP only measure scattering and absorption from the accumulation mode due to inlet effects, the refractive indices derived only represent the accumulation mode up to 2.5 µm, and not the coarse mode.

The method is identical to that described in Ryder et al. (2013b): a Mie scattering code is used to generate optical properties at 550 nm, using the ACC PSD, with refractive index of $m^{550}=1.53$ and $k^{550}$ incrementing in steps of 0.0005 from 0.0005 to 0.006, but with an additional smallest value of 0.0001 which was required for the MBL SLRs. The value of $k^{550}$ which produced best agreement with the SSA from the nephelometer and PSAP was selected for each SLR. The resulting values of $k^{550}$ are shown in Figure 3, with modal values of 0.001i for SAL SLRs and 0.0001i for MBL SLRs. This set of refractive index data is

referred to hereafter as the 'iterated refractive indices.'

The derived iterated refractive indices were then used in two ways. Firstly, a value of 1.53-0.001i was used to correct all size distributions from the PCASP and CDP. Secondly, iterated refractive indices derived for each SLR were selected to generate optical properties for the FULL PSD and ACC PSD described in Section 2.6.

### 2.5.2 Refractive Index from Filter Samples

We use the composition volume fraction and spectral refractive index data of 7 aerosol/mineral components in order to compute refractive indices. These are alumino-silicates, quartz, carbonates, gypsum, iron-rich, sodium chloride (sea salt) and sulfate-rich. Alumino-silicate is represented by a mean of illite and kaolinite refractive indices, carbonates by calcite refractive indices, and iron-rich by a mean of hematite and goethite refractive indices. Sea-salt is represented with NaCl refractive indices. The

sulfate-rich category is represented by ammonium sulfate refractive indices, assumed to be the most likely atmospheric composition. Thenardite is also included in the sulfate-rich category as no refractive index data are available for it, similar to

Kandler et al. (2009). Literature data was taken as follows: illite (Egan and Hilgeman, 1979; Querry, 1987), kaolinite (Egan and Hilgeman, 1979; Glotch et al., 2007), quartz (Shettle and Fenn, 1979; Peterson and Weinman, 1969), calcite (Querry et al., 1978; Long et al., 1993), gypsum (Long et al., 1993), hematite (Shettle and Fenn, 1979; Bedidi and Cervelle, 1993; Marra et al., 2005), goethite (Bedidi and Cervelle, 1993; Glotch and Rossman, 2009), sodium chloride (Toon et al., 1976), ammonium sulfate (Toon et al., 1976). Where no spectral data are available, values are linearly interpolated across wavelengths. Refractive indices are then calculated by weighting each mineral component by its volume fraction given from the composition analysis in Section 2.4, a method which assumes that particles are internally, homogenously mixed. Although it was evident from the SEM analysis that the particles are externally mixed, and it is known that internal and external mixing can result in different optical properties (McConnell et al., 2010). We employ this method partly for simplicity and partly for consistency with previous work (e.g. Kandler et al. (2009; 2011a); Klaver et al. (2011); Formenti et al. (2014). These refractive indices are then used with the size distribution data from the wing probes to generate optical properties using Mie scattering code.

## 2.6 Calculation of Optical Properties

Optical properties (SSA, MEC and g) are calculated using a Mie scattering code, where dust particles are assumed to be spheres. Size distributions used are taken from the wing probes, representing either the FULL PSD or the ACC PSD. Refractive indices are taken from either the iterated RI method (Section 2.5.1) or from internal or external mixing RI values calculated from the filter samples (Section 2.5.2).

When the iterated RI values are used, composition, and therefore $k^{550}$, is assumed to remain constant with particle size. This allows the iterated RI for the accumulation mode to be applied to the full size distribution. For all cases $m^{550}$ was assumed to be 1.53. Iterated values of $k^{550}$ specific to each SLR were used. For the MBL $k^{550}$ varied between 0.0001 to 0.0005, and for the SAL $k^{550}$ varied between 0.0005 to 0.0025 (as shown in Figure 3). Although a value of 1.53 for $m^{550}$ is higher than that produced by the filter sample composition results (Section 3.5, 1.47-1.49) the filters result is likely biased low due to the reasons discussed in Section 3.5. We also performed a sensitivity test to using $m^{550}$ of 1.48 and 1.58, and found that $d_{eff}$ changed by up to 5% and SSA by under 1%.

Where composition from filter samples are used, RI is allowed to vary as a function of particle size according to the composition results. For internal mixing assumptions the refractive indices calculated according to Section 2.5.2 are used. For external mixing assumptions, size distributions of each mineral component are calculated by weighting the size distribution from the wing probes for each mineral component using its number fraction. Scattering properties for the size distribution of each mineral are then computed using Mie scattering code and the same literature refractive indices as described above. Summing scattering and absorption over all minerals then provides the total optical properties for the external mixing case.

Although it is clear that dust particles are not spherical, sensitivity of SSA to shape was tested by Otto et al. (2009) and Johnson and Osborne (2011), who found that SSA changed by under 1% and 2% respectively when non-spherical particles were assumed. This is less than our uncertainty in SSA of 5% due given above due to refractive index, and therefore we consider this an acceptable assumption.

## 3 Results

### 3.1 Dust Sources and Vertical Structure

Figure 1 shows the dust uplift locations (sources) determined using SEVIRI RGB imagery for dust sampled during each AER-D flight. Table 4 shows the corresponding meteorological event driving the uplift, and also the age of dust when sampled by the aircraft. Dust sampled by the first two flights, b920 and b923/924 originated from large scale dust events uplifted in easily identifiable single events. The b920 dust was uplifted over southern Algeria (blue circle in Figure 1), and subsequently transported northwards and then westwards. The dust emerged over the Atlantic from southern Morocco, being sampled by the aircraft after around 4 days' transport. The b923/b924 dust was uplifted further west, over northern Mali (green oval in Figure 1), and was also subsequently transported northwestwards, emerging over the Atlantic over southern Morocco and sampled by the aircraft after a shorter time of 2 days due to the more direct transport path.

During the days leading up to flight b928, there was a change in the dominant meteorological dust export mechanism, as described by Liu et al. (2018). First, dust was uplifted over northern Mauritania (orange circle number 1) underneath widespread altocumulus cloud, which moved slowly to the southwest. Here, it became mixed with dust which had originated from northern Mali (orange circle number 2) where it had been uplifted by an MCS and haboob. This moved rapidly westwards, driven by strengthened 700 mb winds, which characterized the second phase of ICE-D (Liu et al., 2018). The two dust events became mixed together and subsequently transported dust over the Atlantic via the Mauritanian coastline, thus taking a more direct southerly transport route than the dust exported and sampled by the first two flights of the campaign (b920 and b924). Thus the dust sampled during b928 was a mixture of dust from two different source regions. The final two flights, b932 and b934, consist of dust uplift events which subsequently overpassed secondary (or tertiary) uplift events, and are therefore considered as dust from a mixture of the identified sources. The dust sampled during b932 was initially uplifted by a MCS and haboob close to the Mali/Algerian border, and some small scale reinvigorated convection caused additional uplift over northern Mali, before the dust was transported southwestwards towards the Mauritanian coastline. Dust encountered during b934 was also initially uplifted in almost the same region along the Mali/Algeria border by a MCS and haboob, and on each subsequent evening a new convective cell developed over the transported dust with a new haboob – over the Mali/Mauritanian border (purple circle number 2), and then over western Mauritania (purple circle number 3). Therefore the range of possible dust ages for b934 is very large (16 hours to 3 days). The range of sources identified during AER-D are consistent with well-known source regions in the literature (Engelstaedter et al., 2006; Formenti et al., 2011; Evan et al., 2016; Scheuvens et al., 2013),

particularly the Mali/Algeria border hotspot downwind of the gap between the Atlas and Ahaggar Mountains (Potential Source Area 3 in Formenti et al. (2011)), which contributed to 4 of the 5 AER-D dust events.

Figure 4 shows the vertical structure of the five dust events sampled during AER-D from in-situ aircraft measurements. Four events (b920, b928, b932 and b934) display typical SAL structure, with elevated dust from a base of between 0.5 to 1.5 km up to an upper bound of 5 km (6 km in one case) overlying the MBL. Accumulation mode extinction coefficients vary from around 100 to 500 Mm$^{-1}$ in this layer, sometimes with fairly constant values in the vertical (b920, b934). At other layers were more sinuous (b932, b928), but always had accumulation mode scattering angstrom exponents (SAE; 700 to 450 nm) of fairly constant value around -0.5 – a clear indication of coarser particles. Within the SAL, water vapour mixing ratios were low (under 10 g/kg) and potential temperature increased slowly with altitude. Relative humidities were low at 30-50% at altitudes where SLRs were performed. Contrastingly, in the MBL relative humidities were high (>90%) as expected, with the MBL capped by a temperature inversion. SAE values in the MBL were variable, sometimes exhibiting a jump to positive values (b920, b934) suggesting a dominance of smaller particles, whereas sometimes hardly displaying any difference to the overlying dust (b928, b932). In the former two cases, the PCASP size distribution confirms a greater relative contribution from fine (d<0.3μm) particles in comparison to those sized 0.3 to 3 μm. This vertical structure is as expected for the region in summer, and in keeping with the elevated and dusty SAL above the MBL (Prospero and Carlson, 1972).

In contrast, the dust event sampled on flights b923 and b924 displays very large dust loadings and different vertical structure. The upper SAL, from 2.5 to 5 km, displays roughly constant potential temperature and water vapour mixing ratio, with extinction coefficients of around 140 Mm$^{-1}$. Between 0.5 to 2.5 km, dust loadings become extremely high reaching 2122 Mm$^{-1}$ at 1.2 km, contributing to the very high AOD of 2.54 for the accumulation mode only (including coarse particles would increase the estimated AOD further). To the authors' knowledge, this is the highest aircraft-measured value of dust-related extinction measured for dust transported over the ocean, and is explored further by Marenco et al. (2018, in review). (Note that lidar-derived AODs and extinction shown by (Marenco et al., 2018, in review) are slightly lower than those shown here, which may be due to different extinction properties of the dust at the lidar wavelength of 355 nm, the Rosemount inlet enhancement effects shown in Figure S2, or the differences between a lidar curtain and sloped aircraft in-situ profile). Moisture levels here decrease with decreasing altitude (8-12 gkg$^{-1}$) and are larger than observed within dust during the other flights. The higher levels of moisture are perhaps not surprising given that a MCS-haboob uplifted this dust two days earlier, and one day earlier Saharan Boundary Layer (SABL) convection can be seen impacting the dust over northern Mauritania (consistent with Marsham et al. (2013); and recycling of moisture within the dusty SABL in Ryder et al. (2015)). Note that these values of extinction do not include the coarse mode, and therefore actual extinction values will be even higher. Beneath the thick, low altitude dust layer was a shallow MBL extending up to 500 m. This unusual vertical structure of intense, thick dust within the bottom half of the SAL is not in keeping with the conventional SAL model (Prospero and Carlson, 1972) of well-mixed

elevated dust throughout the whole SAL, and is therefore of additional interest. This is discussed in detail by Marenco et al. (2018, in review).

### 3.2 Size distributions and shape

Figure 5 shows the wing-probe size distributions for each SLR, separated into those measured in the SAL and MBL. The size distribution in the SAL displays a broad shape which does not change with increasing or decreasing dust load, but simply shifts between higher or lower volume concentrations. For example, for b924 under large dust loadings where lidar-derived AODs approached 2.0 (Marenco et al., 2018, in review), the volume concentrations are markedly larger than other flights in AER-D (green points in Figure 5), although the size distribution shape is much the same. This is in contrast to measurements over land close to dust sources during Fennec (Ryder et al., 2013b) where the absence or presence of the coarse and giant modes had a strong impact on the overall shape of the size distribution, since the relative proportion of giant particles was observed to increase. The peak of the volume concentration during AER-D was constantly between 5-10 µm diameter.  We observe a fine mode of aerosol at 0.1-0.3 µm, which is evident when dust concentrations are lower, but becomes eclipsed by the accumulation mode dust during flights with larger dust loadings (e.g. b924). Section 3.4 shows that the composition of d < 0.5 µm aerosol in the SAL during AER-D was dominated by sulfates and salts, thus explaining the different behaviour of this mode. Liu et al. (2018) examine the composition and behaviour of the accumulation mode during ICE-D in more detail.

In the MBL, a distinct giant mode is evident between 20 to 60 µm during three flights (b924, b928 and b932: red, green and orange in Figure 5b). For flight b928, filter sample analysis (Section 3.4) confirms that this giant mode is composed of dust, rather than sea salt. Higher wind speeds in the MBL, which may be an indicator of sea salt abundance, are not correlated with the presence of this giant mode (10-11 ms$^{-1}$ for b924 and b932; 3-5 ms$^{-1}$ for b920, b928 and b934). It is possible that turbulent mixing from the SAL to the MBL is more likely given a weak inversion and higher turbulence. Therefore we also examined whether the strength of the temperature inversion (calculated using the vertical gradient of potential temperature) and the strength of turbulence at the inversion (indicated by the variance of vertical velocity) could be related to the giant particles present in the MBL. However, we found no obvious connection in these cases. We note that giant mode particles in the MBL were only present during flights when there was a significant presence of giant mode particles in the SAL above: there is a giant mode in Figure 5b (flights b924, b928, b932; green, orange red) only when a higher concentration of particles in the SAL in the same size range was measured (Figure 5a, same colours). When few particles are measured in the SAL in this size range (b920 and b934, blue and purple), the MBL giant mode is also absent. Thus the observations suggest that the giant MBL mode may be dust being deposited from the overlying SAL towards the ocean. This has also been suggested by Jaenicke and Schutz (1978) from aerosol surface observations at Sal, Cape Verde, where giant particles (d > 40 µm) were observed to arrive at the site a day after that of coarse dust particles (6 < d < 60 µm).

Despite careful error analysis and selection of instrumental data, there is still a large degree of noise as a function of diameter in the wing probe size distributions shown in Figure 5, which was also not indicated in the filter sample PSDs. Therefore in Figure 6, lognormal curves using 4 lognormal modes are fitted to the wing probe instrumental data using a least squares regression (Markwardt, 2008) with mode parameters from Table 5. The lognormal curves represent a best-fit across the full size range for the instruments available. Differences between the effective diameter calculated with the best-fit lognormal curves and the observed PSDs are between 10-15%, resulting largely from deviations between the observations and best-fit curve in the CDP size range (3 to 20 µm).

It can be seen that the volume concentrations in the SAL are larger than those in the MBL, as expected due to the higher concentrations of elevated dust, and that the SAL size distribution has a notably different structure. Dust in the SAL displays a broad size distribution with contributions from particles over a wide range of sizes (0.3 to 100 µm), peaking at 5 - 10 µm. The MBL size distribution has a narrow peak at d~5 µm and a giant mode contribution at around 20 to 60 µm, where volume concentrations in the MBL mean are actually larger than the SAL mean. Compared to PSDs over the desert measured with aircraft during Fennec and SAMUM1 (Ryder et al., 2013b; Weinzierl et al., 2009), where the volume distributions peak at diameters larger than 10 µm, AER-D has a smaller giant mode contribution, as would be expected. The AER-D PSD is more in keeping with other aircraft observations of transported dust, where the volume distributions peak at diameters between 3 to 10 µm (measured during GERBILS (Johnson and Osborne, 2011), ADRIMED (Denjean et al., 2016), SAMUM2 (Weinzierl et al., 2011) and SALTRACE (Weinzierl et al., 2017)). Despite some agreement here, the variation of size across the 3 to 10 µm size range can still have a large impact on the dust radiative effect (Tegen and Lacis, 1996). MBL PSDs under the influence of dust advection were measured by Kandler et al. (2011b) at Cape Verde. Their observations revealed a sharp mode at 10 µm, declining steeply at larger sizes, which contrasts to our modes centred at d~5 µm and 20 to 60 µm. They also found that when air masses had a maritime origin (but still dominated by dust) the PSD was broadly flat between around 10 to 80 µm, similar to the AER-D MBL PSDs.

Size distributions have also been derived from the 4 filter samples, and are contrasted to the wing-probe size distributions in Figure 7. Such comparisons have previously been shown to be challenging due to the different nature of measurement from each instrument (e.g. Chou et al. (2008); McConnell et al. (2008); Price et al. (2018)) and discrepancies are common, particularly with non-spherical particle geometry. We note that particles measured by the PCASP will be randomly orientated due to passing into the PCASP nozzle. Larger particles sampled by the CDP and those measured by the 2DS and CIP15 may be aligned horizontally in the atmosphere (e.g. Ulanowski et al. (2007)), and measured in this orientation. Also,, each technique allocates size using a different methodology. Additionally filter sample viewing is likely to preferentially view the larger cross-section of plate-like particles as they fall flat on the filter substrate.

Figure 7 shows that in most cases, the filters size distribution is greater than that from the wing probes. In the size range 0.5-1 µm the best agreement is found. In 3 of the 4 SLRs, the filters coarse mode volume distribution exceeds that of the wing probes by an order of magnitude or more. At d<0.5 µm the filters size distribution shows a much more distinct fine mode than that seen from the PCASP. Solid lines for the filter samples (blue and orange) in Figure 7 indicate volume distribution calculated assuming spherical shape using diameter calculated from the area-equivalent diameter of the fitted ellipse. It is possible that the filter samples overestimate size, and therefore also volume, if particles are plate-like and fall flat on the filter substrate. We test whether accounting for this by including a representation of particle height, using a height:maximum axis ratio of 1:3 (Chou et al., 2008) can improve the agreement with the wing probes. This sizing metric is shown by the dashed lines in Figure 7. Although accounting for preferential particle orientation on the filters makes some differences to the derived size distribution, it is not sufficient to allow agreement with the wing probe size distributions. It is possible that there were problems with the filters flow rates measured during AER-D resulting from some combinations of filter pore sizes and filter supports. SLRs with higher flow rates (b920 R2 and R5) show better agreement with the wing probes (Figure 7a and c) and also used filter samples with the larger pore sizes (0.4 µm), compared to the other two SLRs (Figure 7b and d) which have worse agreement. We note that for all the SLRs shown, the shape of the coarse mode size distribution is the same for both filters and wing probes, even if they are offset. For example, the broad shoulder of the coarse mode size distribution can be seen in b928 R2 (Figure 7b), and the sharper drop off of the coarse mode in b920 R5 and b934 R6 (Figure 7c and d), for both filters and wing probe size distributions. Additionally diameters of the peak size distribution (5-10 µm) are consistent between filters and wing-probes for the SAL (c and d). We also note that the filters PSD is much smoother than that from the CDP.

Examination of the metric of effective diameter is useful, since it takes into account the contribution of a range of particle sizes. Figure 8a shows that the effective diameter of the full size distribution computed from the wing probes is fairly constant with altitude with no discernible trend, as expected for a well-mixed SAL. A similar picture is seen for the VMD but with values of around 5 to 6 µm (not shown). No evidence is found for larger particles being more abundant closer to the base of the SAL, as would be expected due to gravitational settling. For the SAL the mean (minimum, maximum) $d_{eff}$ value is 4.0 µm (3.6, 4.7) while for the MBL the mean is 4.6 µm (3.4, 5.5). Variation within the MBL is much greater due to the absence or presence of the giant mode shown in Figure 6. Mean (minimum, maximum) $d_{eff}$ for the accumulation mode is smaller at 1.7 µm (1.4, 2.0) for the SAL, and 1.1 µm (0.7, 1.4) for the MBL, in agreement with Liu et al. (2018), reflecting the enhanced fine mode in the MBL size distribution from 0.1 to 0.3 µm diameters shown in Figure 5. We also note that $d_{eff}$ for flight b924 in the thick dust layer was 4.3 µm – not at all different from other $d_{eff}$ values in typical dust loadings, despite total volume concentration being much larger. Mean AER-D size parameters of $d_{eff}$ and VMD are given for the SAL and MBL in Table 6.

In addition to $d_{eff}$, we also show the maximum size ($d_{max}$) detected by the 2DS XY at concentrations greater than $10^{-5}$ cm$^{-3}$ (see Section 2.3.2), as a useful indication of transport of the largest sizes, which can contribute substantially to the mass fraction and are therefore important to dust biogeochemical cycles. Figure 8b shows maximum size detected during AER-D. The largest

value of 80 µm at 900 m comes from flight b924, during the intense dust event. Within the SAL, $d_{max}$ varied between 20 to 80 µm, and the same range was found within the MBL. There is no clear trend of $d_{max}$ decreasing with altitude. Particles sized 20 µm (40 µm) or larger were detected in 100% (36%) of the AER-D dust layers investigated, and in 100% (80%) of MBL layers. The lack of decrease of $d_{max}$ with altitude during AER-D is similar to that observed over the desert during SAMUM1 by

Weinzierl et al. (2011).

The prevalence of coarse particles shown in Figure 8b is greater than predicted due to settling velocities alone: a 20 µm particle should fall 5 km in 1.4 days (Li and Osada, 2007), and a 40 µm particle would take 13 hours for the same distance. Therefore with the dust age range in AER-D estimated at 17 hours to 4.6 days, we would not expect any 40 µm particles to be present at

all, and would only expect 20 µm particles at altitudes below 2.4 km, yet 40 µm sizes were measured over 3 km and 20 µm sizes were measured at altitudes over 4 km.

Aspect ratio derived from the filter samples, defined as the ratio of the major to minor fitted-ellipse axes, is shown in Figure 9. The results are somewhat noisy in the larger size ranges due to the relatively small number of particles analysed, and samples

where n<10 have been excluded. As expected, larger particles are more non-spherical in general, with higher median aspect ratios (between 1.30 and 1.51) for the 5 to 10 µm and 10 to 40 µm size ranges. This contrasts to the 0.5 to 5 µm range where median aspect ratio varied from 1.3 to 1.44 and modal values are 1.3. This is particularly notable for b928 R2 in the MBL where there were enough giant mode particles (10-40 µm) counted, showing much larger modal aspect ratios of 1.5 (median of 1.50). For flight b920 the smallest particles (0.1-0.5 µm) were more spherical (modal aspect ratios 1.0-1.2, median values

1.08 and 1.23), though b928 R2 and b934 R6 contrast to this with much greater fractions of higher aspect ratios for smaller particles (mode aspect ratios 1.2-1.4, median values of 1.27 and 1.13, a larger tail in the aspect ratio distribution). This is explained by the composition of these latter 2 SLRs being more strongly dominated by sea salt with a cuboid shape. Data representing the accumulation mode and full PSD strongly shadow the smallest size bin shown, since the data are dominated by smallest particles with the highest number concentrations. Our median values are slightly lower than the majority of those

reported in the literature. E.g. SAMUM1 values were around 1.6 for d>0.5 µm and 1.3 for smaller particles (Kandler et al., 2009), while at Praia, Cape Verde, Kandler et al. (2011a) found values of 1.6-1.7 at d>0.7 µm and values below 1.4 at d<0.7 µm, and Chou et al. (2008) found median values of 1.7 during AMMA. Contrastingly Rocha-Lima et al. (2018) found lower modal values of 1.3 from ground-based samples during Fennec.

### 3.3 Mass Loading

In order to facilitate comparisons with model data, where typical output is in terms of dust mass loadings, Figure 10 shows dust mass loadings from all the 31 AER-D in-situ aircraft profiles, calculated from the measured size distributions and assuming a typical dust density of 2.65 gcm$^{-3}$ (Tegen and Fung, 1994) and spherical particles. The intense dust profiles encountered during certain sections of flights b923/b924 are highlighted in orange, where dust was elevated further north

around the Canary Islands, and in red where the dust was found at lower altitudes at around 23N, since they both show notably higher dust mass loadings. Due to the presence of a strong coarse and giant mode, mass loadings are generally high, and typically 300-1000 $\mu gm^{-3}$ in the elevated SAL (Figure 10a, black lines) when AODs were low to moderate (0.2 to 0.5) (black lines). Exceptionally high values were found during the intense dust event on 12 August 2015 when values of mass loading exceeded 1000 $\mu gm^{-3}$ and reached a maximum value of around 4600 $\mu gm^{-3}$. These values are of the order of a factor of ten larger than those observed in the region previously measured up to a maximum size of 20 $\mu m$ (Collins et al., 2000; Garrett et al., 2003).

Figure 10b shows that the mass contained within the accumulation mode (d<2.5 $\mu m$) is around a factor of ten lower than the total mass, indicating that sub-sampling dust properties behind size-limiting inlets significantly under-samples dust mass. Specifically, we find that on average 14% (10th and 90th percentiles at 6 and 28%) of total mass is contained within the accumulation mode between altitudes of 1.5 to 4 km, values in agreement with estimates by Kok et al. (2017). Figure 10c and d further illustrate the impact of size on the mass loading by showing the fraction of mass contained at diameters greater than 5 $\mu m$ and 20 $\mu m$ (c and d respectively). These values are selected since 5 $\mu m$ is the diameter at which models begin to under represent dust mass concentration compared to observations (Kok et al., 2017), and few models represent dust particles larger than 20 $\mu m$ (Huneeus et al., 2011). On average, around 60% of the mass is found at sizes greater than 5 $\mu m$ and 0-12 % at sizes greater than 20 $\mu m$ in the SAL between altitudes of 1.5-4 km where most of the mass is found. Within the MBL a greater fraction of mass is found at large sizes: 70-80% greater than 5 $\mu m$ and 10-20% greater than 20 $\mu m$. In the extreme, up to 90% of dust mass can be found at sizes greater than 5 $\mu m$ and up to 40% at sizes greater than 20 $\mu m$.

There appears to be a trend with altitude shown in Figure 10c: the mean mass fraction at d > 5 $\mu m$ decreases steadily from 0.75 at the surface to 0.23 at 5 km altitude. A decrease is also evident in panel d with the largest fractions being found towards the bottom of the SAL (excluding the MBL). These decreases with dust mass as a function of altitude are somewhat in contrast to the homogeneous distribution of $d_{eff}$ throughout the SAL shown in Figure 8. This may be due to the data shown in Figure 10 coming from profiles rather than SLRs, such that more data is available, and also that although $d_{eff}$ represents the full size distribution, as such it is relatively insensitive to smaller changes in the coarse and giant particle concentration. Either way, there is clearly evidence of coarser dust particles being more prevalent towards the bottom of the SAL (and also in the MBL), indicating deposition processes occurring.

Additionally, we have calculated dust mass path (DMP, also known as integrated column mass loading) values from the mass profiles, where DMP is the vertically integrated mass of dust per unit area (Evan et al., 2014). For AER-D low to moderate AODs, mean DMP is 1.0 $gm^{-2}$ (minimum and maximum values of 0.2 and 2.4 $gm^{-2}$). These values are higher than, but within the bounds of error and variability of those derived from satellite retrievals in the same geographic region (Evan et al., 2014). However, DMPs produced by CMIP5 models are much lower (0.05 to 0.46 $gm^{-2}$ with a multi-model median of 0.26 $gm^{-2}$) –

falling at or below the lower edge of the AER-D values, furthering the argument that models underestimate dust mass loading due to poor representation of dust coarse mode. For the intense dust event on 12 August DMP values are extremely large, from 3.1 to 6.2 gm$^{-2}$.

### 3.4 Dust Composition

Figure 11 shows the size resolved and bulk (full PSD and accumulation mode, d<2.5 µm only) composition results for the 4 filter samples analysed. In general, across all the samples and all size ranges above diameters of 0.5 µm, the particles are dominated by alumino-silicates and quartz, with alumino-silicates forming over 80% of the composition volume, with quartz typically forming around 10% of the volume, though sometimes being up to 20%, consistent with Price et al. (2018). Other components are generally low in volume percentage, with calcium-rich particles providing up to 15% of volume content in

some samples. Particles dominated by iron are present in low quantities (and therefore displayed separately on the right hand axis), although their contribution is extremely important in controlling shortwave refractive indices (section 3.5). Fe-rich particles are present in higher quantities in the SAL cases (0.5-2.5%) compared to the MBL (0-0.9%). Error bars are noticeably large for Fe-rich particles due to the relatively low counting statistics. Although there is some variation in the iron content as a function of particle size, no distinct pattern is displayed across all the samples, and the uncertainties prevent definitive

conclusions from being drawn. Even when the uncertainties are reduced by looking at the full PSD vs accumulation mode only (black circle and triangle), iron content is higher for the full PSD than the accumulation mode in the SAL for b920 R5 (1.7% vs 1.3%), whereas for b934 R6 also in the SAL, iron content is higher in the accumulation mode than for the full PSD (1.3% vs 0.6%). These values agree well with the range of hematite content proposed by Balkanski et al. (2007) where values span 0.9 to 2.7%, though our values will be low-biased as we only include iron when iron was the dominant component of a particle,

and it is evident that iron is present as a portion of almost all larger particles. Iron is detected across the full size range in variable amounts, thus where absorption is increased due to large particle sizes, it will be further increased due to elevated iron content.

The size range 0.1 to 0.5 µm (shown in dark blue) displays notably different composition, in keeping with the different size

distribution displayed for the fine mode (Figure 5 and Figure 7) indicating a different aerosol type. Although some fraction of this fine mode is composed of dust particles (alumino-silicates and quartz), there is always a contribution from sulfate particles (10-40%). Contributions from sea salt in the MBL fine mode (25 and 55%) are higher than in the SAL, as expected. Samples B928_R2 and B934_R6 show noticeably higher thenardite concentrations, possibly indicative of dry saline lake origins (e.g. Formenti et al. (2003)) or sea salt reacted with sulfuric acid. The high contribution of sea salt in the fine mode in b928 R2

(Figure 11b) impacts the aspect ratio distributions for this size range shown in Figure 9b, where these small particles are much less spherical and present a rectangular shape in the SEM imagery, inferred to be a cuboid shape.

Figure 11 shows that even within the MBL, the aerosol content was dominated by dust particles in both cases examined here. Additionally, during b928 R2 (orange points in Figure 5b, and Figure 11b), one of the SLRs when a substantial giant mode was present, the composition data confirms that this giant mode (purple points) was composed of dust, being dominated by alumino-silicates, quartz and calcium rich-particles (most likely calcite).

Kandler et al. (2009; 2011a) show that the quartz fraction for particles increases with particle size, particularly for d>20 µm. Our data do not show that quartz content appreciably increases with size, though this may be due to the relatively large errors on the size-resolved data, although even the bulk data for the full PSD and accumulation mode PSD with smaller error bars do not show significant differences in quartz content. It may also be due to our filters data not extending to such large size ranges (up to 200 µm) as Kandler et al. (2009).

Additionally, in contrast to Liu et al. (2018), we do not detect any black carbon on the filter samples, which they find present predominantly between sizes of 0.1 to 0.6 µm. During the analysis of 6500 particles, only one black carbon chain structure was observed. This difference is unlikely to be due to the pore size of the filter samples (0.2 and 0.4 µm) allowing small BC particles to pass through them, since our filters collect efficiently over a size range extending below this by diffusion and impaction (e.g. Lindsley (2016)) and the size distributions in Figure 7 clearly show that this size range is collected. BC loadings for the ICE-D flights shown in Liu et al. (2018) are very low (0.05-1.0 µgm$^{-3}$). The most likely explanation is that BC particles were not present in sufficient concentrations to be sampled by the filters.

### 3.5 Refractive Indices

Size resolved and bulk complex refractive indices at 550 nm calculated from the filter samples' composition are shown in Table 7, and full spectral variability (solar and terrestrial) is shown in the supplement, figures S2 and S3. Real values in Table 7 representing the full PSD and accumulation mode PSD are generally 1.47 to 1.49 and do not vary substantially. These values are relatively low compared to that expected for dust, and may be influenced by the lower real part of illite in determining the 'alumino-silicate mean' refractive index which is an average of illite and kaolinite. Other refractive indices such as those of Kandler et al. (2009; 2011a) solely apply kaolinite refractive indices and produce higher real values. The low values may also be influenced by the internal volume mixing rule applied here, and are consistent with those from Formenti et al. (2014). Additionally, the percentages of kaolinite and illite by mass respectively have been shown to vary between 71.1±13.5% and 16.4±11.8% for dust originating from Southern Mali/Algeria, to 30.7±3.2% and 54.2±6.0% for dust from Mali/Mauritania/Western Sahara (Formenti et al., 2014). Real values are also distinctly higher for the smallest size class, though this has no impact on the bulk value due to the low volumetric contribution from these small sized particles.

Much more variability is seen in the imaginary part, largely influenced by the iron content and its absorbing influence via hematite and goethite. For the two samples in the SAL, k is the same for both in the accumulation mode (0.0023), but increases

or decreases to 0.0030 or 0.0012 when the coarser particles are included. To explain this, inspecting the size resolved data, it can be seen that for sample b920 R5 k increases with size from 0.0020 to 0.0034 reflecting the increasing iron content with size seen in Figure 11c. However this is not the case for the second SAL sample (b934 R6) where iron content and k are highest (0.0023 to 0.0039) in the mid-range sizes from 0.5 to 5 microns (Figure 11d), while lower k values (0.0003 to 0.0009) are found for d < 0.5 µm and d>5 µm due to their lower iron content.

Imaginary parts for the MBL are much smaller than those for the SAL (0.0004-0.0005 compared to 0.0012 to 0.003), reflecting the lower iron content across all size ranges. Real parts in the MBL are not notably different to those in the SAL.

Longwave spectral refractive indices have also been calculated and are shown in Figure S3 in the supplement. Here the main controlling factor is the fraction of sea salt and sulfate-rich particles, which is important for d<0.5 µm and occasionally up to d=1 µm in the MBL, and the relative proportion of alumino-silicate and quartz is important for sizes d>0.5 µm. The refractive indices of sea salt used here have zero absorption in the longwave spectrum, so higher salt content lowers k in the smallest size range. All samples show an increase in absorption within the atmospheric window between around 8.5 to 10 µm. The quartz absorption peak occurs at 9.4 µm while the alumino-silicate peak occurs at 9.6 µm. Thus the relative proportions of these two minerals control the height of each of these peaks, but since no significant size resolved change in the quartz:alumino-silicate ratio was found in Section 3.4, any size resolved changes in these peaks in the longwave refractive index are negligible.

The longwave imaginary parts are substantially higher between 9-10 µm, by 2 to 3 times, than those of some of the literature where k is less than 1.0 (Di Biagio et al., 2017; Fouquart et al., 1987; Volz, 1973; Hess et al., 1998; Balkanski et al., 2007), although our values similarly high to those of both Otto et al. (2009) and Formenti et al. (2014) who used the similar internal mixing volume calculations. We purport this to be due to the assumption of an internally mixed dust aerosol to be inappropriate, knowing that the dust actually is externally mixed (excepting the complex nature of iron existing within mixtures), as in Formenti et al. (2014), though why this assumption produces worse results in the longwave spectrum is unclear. Other ways of calculating the refractive index based on iron being internally mixed, while quartz and alumino-silicates are externally mixed are possible (e.g. Balkanski et al. (2007)), as are other more complex ways of representing internal and inhomogeneous mixing (Lindqvist et al., 2014; Nousiainen, 2009) but are beyond the scope of this work.

### 3.6 Optical Properties

Figure 12 shows how SSA varied with height for the accumulation mode and the full size distribution. SSA was fairly constant with altitude within the SAL. Measured mean (minimum, maximum) accumulation mode SAL SSAs (black) were 0.97 (0.93, 0.98). MBL values were all greater than 0.99. Mie-calculated values of SSA for the accumulation mode (orange) agree with measurements (black) within error bounds, since here agreement in SSA is tuned by refractive index iterations. Once the full size distribution had been accounted for, calculated SSA values (green) decrease to 0.95 (0.91,0.98) for the SAL and to 0.99

(0.97,0.99) for the MBL. Since larger particles are more absorbing for a fixed refractive index, this decrease is expected, and the magnitude of the decrease is dependent on the amount of coarse particles, and also the refractive index. In the MBL, SSA values do not decrease much since $k^{550}$ is small (~0.0001i); the addition of a coarse mode of negligible absorption makes little difference to SSA. However, in the SAL, where derived $k^{550}$ varies from 0.0005 to 0.0025i, the addition of the full coarse mode causes SSA to drop noticeably.

Since it is clear from the composition results that aerosol in the SAL is dominated by dust across all sizes, applying the same $k^{550}$ for the accumulation mode to the coarse particles may be appropriate. However, some variation of dust composition with size is still evident (Section 3.4) which is not taken into account in Figure 12. The low $k^{550}$ of 0.0001i in the MBL is representative of highly scattering sea salt or sulfates which dominate the fine mode in the MBL (Section 3.4), although it is clear that dust particles dominate at larger sizes in the MBL. Therefore extending the low value of $k^{550}$ to the coarse mode in the MBL is a less reasonable assumption. Therefore it is likely that the SSA values in the MBL for the full size distribution shown in Figure 12 are overestimates. SSA values calculated using a RI representative of dust for the coarse mode from filter samples are discussed later on.

Theoretically, the variability of the optical properties of dust in the SAL may be determined by either the dust composition, the dust size distribution, or both. Other elements may also influence the optical properties, such as particle shape, roughness, hygroscopic growth and mixing, but are not considered here as there are few observational constraints available on these properties from AER-D. In order to determine the contributions from both particle size (assessed via $d_{eff}$ from the measured size distribution) and composition (assessed via $k^{550}$ calculated from two different methods described in Section 2.5) to the variability of SSA, we show both $d_{eff}$ and $k^{550}$ as a function of SSA in Figure 13, separated into the ACC PSD and FULL PSD size ranges.

It is clear that during AER-D, it was the variability of dust composition that controlled the variability of SSA, rather than the variability of size distribution. In Figure 13a for the SAL (orange) points, it can be seen that the direct observations for the accumulation mode (orange asterixes, where nephelometer and PSAP scattering and absorption are used to calculate SSA) are in agreement with Mie simulations using the ACC PSD (small orange circles). When the coarse mode size distribution is then included (large orange circles), effective diameter increases and there is a slight decrease in SSA. Although $d_{eff}$ does impact the magnitude of the SSA, within AER-D the shape of the PSD was relatively stable, with a relatively constant $d_{eff}$. This meant that the small variation of size distribution shape had minimal impact in determining the variability of SSA.

Figure 13b shows the variation in SSA as a function of composition, represented by $k^{550}$. Here we show the relationship between these two variables calculated with different RI datasets: firstly for RI from the Mie scattering iterations (circles, Section 2.5.1) and secondly for RI calculated for filter sample composition (diamonds, Section 2.5.2). Optical properties are

shown for the accumulation mode only (small data points), and for the full PSD (large data points). For dust in the SAL, Figure 13b shows a consistent decrease in SSA with the imaginary part of the refractive index. This relationship becomes more negative when the full coarse mode PSD (large symbols) are included because for a fixed RI, the larger particles exert more absorption. Although there is some variability between the results from different RI datasets, overall they show the same trend.

Contrastingly to Figure 13a, Figure 13b shows that the SSA variability was strongly influenced by the variability in composition. This is the case for both accumulation mode observations of SSA, and for the full size distribution. It is not surprising that variability in $k^{550}$ influences absorption and therefore SSA. However, the SSA can be influenced by several factors, including the PSD. Our aim is to investigate which factors influence the variability of the SSA. Therefore it is notable that there is so little variation in the PSD during AER-D that the composition (or $k^{550}$) is the main factor contributing to the variability of the SSA. This finding is notably the opposite from that found during Fennec, where the size distribution was the dominant controller of optical properties. Liu et al. (2018) show that hematite content is important in the ICE-D/AER-D samples as a controlling factor on optical properties. Moosmuller et al. (2012) and Caponi et al. (2017) also show dependencies of refractive index on iron content. This is consistent with our findings that the calculated refractive index from the filter samples is strongly influenced by the iron content and its absorbing properties. It appears that over the Sahara, variations in the PSD (affected by dust age) have an important impact on SSA, while over the ocean the impacts of composition (perhaps either by chemical aging or by sampling dust from different sources) become more important.

Finally, we compare optical properties calculated with composition data from the filter samples using both internal and external mixing assumptions against observations in Figure 14 for the ACC PSD (a) and the FULL PSD (b). Observations are available only for the ACC PSD size range. Figure 14a shows that compared to the observations of SSA, the external mixing assumption provides closer agreement with the observations, confirming that the internal mixing assumption used to derive refractive indices shown in Section 3.5 overestimates the absorption, consistent with previous publications (Formenti et al., 2014). Figure 14b shows the same results but for the full PSD, and also those for the iterated RI for the full PSD. The internal and external mixing calculations, when applied to the full size distribution, allow for the composition of larger particles being dust in the MBL, contrasting to the iterated RI method which assumes constant composition across all sizes. For the two MBL SLRs the SSAs drop from 0.99 for the ACC PSD down to 0.98 and 0.97 for external mixing when the coarse mode and its composition is accounted for – thus producing SSAs in the MBL similar to those of the overlying SAL. This is only achieved through analysis of the coarse mode aerosol composition within the MBL. SSAs from internal mixing RIs are also 0.98 and 0.97 for the MBL SLRs, but these are much the same as those for the accumulation mode. For b934 R6 there is not much variation between the three methods, while for b 920 R5 the SSA is lower at 0.89 for internal mixing, compared to 0.96 for the other two methods. This is because b934 R6 contained more iron in the coarse mode than the other SLRs (see Figure 11c) which had a strong impact on lowering the SSA for an internal mixing assumption. This is not reflected in the external mixing value due to the non-linearity of the scattering and absorption properties.

Campaign mean optical properties representing the full size distribution in the SAL and MBL are summarized in Table 6. In AER-D, MEC in the SAL is higher (0.27-0.35 $m^2g^-$) compared to Fennec observations over the Sahara (0.15-0.23$m^2g^{-1}$, Ryder et al. (2013a)) as a result of fewer coarse and giant particles. AER-D SAL SSA values (0.91 to 0.98, mean 0.95) are somewhat

higher than those of Fennec (0.86 to 0.97) for the same RI (Ryder et al., 2013b), and closer to previously published higher values, for transported dust. For example, (Haywood et al., 2003) values of 0.95-0.98 during SHADE and Chen et al. (2011) values of 0.97±0.02 during NAMMA. AER-D SAL Asymmetry parameters are large at 0.74, unsurprising given the presence of coarse particles, contributing to forward scattering.

## 4    Conclusion

Dust in the SAL during the AER-D airborne field campaign from six different flights, 19 in-situ SLRs and 31 profiles has been characterized. The flights were performed in August 2015 between the Cape Verde and Canary Islands. In particular, a strong focus is given to the presence and contribution from coarse and giant dust particles through operating wing-mounted instrumentation intended to sample the full size distribution from 0.1 to 100 µm diameter. This work fills a research gap by providing in-situ coarse mode dust observations which firstly cover the full size range, and secondly by providing these

observations in the SAL close to the African continent during the peak dust transport season.

Dust sources contributing to the events sampled were located in southern Algeria, Mali and northern Mauritania, with a well-documented dust hotspot along the Mali/Algeria border contributing to dust sampled in 4 out of 5 cases. Several events sampled dust which had been uplifted from up to three separate source regions. Dust age at sampling was determined to be 17 hours to

4.6 days. Dust transport pathways, ages and source locations assessed with SEVIRI dust RGB satellite imagery were different to paths indicated by back-trajectories, in keeping with Trzeciak et al. (2017) that back-trajectory models struggle over the convective summertime Sahara. Vertical structure of the dust was consistent with the conventional model of the SAL, with elevated dust (~1-5 km) overlying the MBL, except in one intense dust event with an AOD of 2.5, with thick dust concentrated between 500 m to 2.5 km altitude.

Size distributions spanning 0.1 to 100 µm were assessed by combining wing-mounted optical particle counters and shadow probes. Mean $d_{eff}$ for the SAL (minimum, maximum) was 4.0 µm (3.6, 4.7 µm) while for the MBL the mean was 4.6 µm (3.4, 5.5 µm). The campaign mean VMD was 5.5 µm. The shape of the measured size distribution did not vary significantly between dust layers (reflecting $d_{eff}$ values which were relatively constant both at different altitudes and in different dust events). Volume

size distributions consistently peaked between 5 to 10 µm, even though total volume concentrations were found to increase or decrease with dust loading. This contrasts to the Fennec results over desert where the contribution from the coarse mode was highly variable, as was $d_{eff}$ (Ryder et al., 2013b). Within the SAL dust layers, particles sized 20 µm diameter or larger were

detected in 100% of cases, and particles 40 µm or larger were detected in 36% of cases, at concentrations over $10^{-5}$ cm$^{-3}$. Based on dust age at sampling time, more coarse and giant particles were present than expected due to gravitational sedimentation alone.

Giant particles (d>20 µm) were found in the MBL on 3 out of 5 flights. Filter sample analysis for one of these cases confirms that these giant particles were dust. The shape of the size distribution indicates similarity to the dust layers above in the SAL, and therefore suggests a high likelihood of dust being deposited from above. Despite this, the only size metrics which showed evidence of increasing particle size towards the bottom of the SAL were the mass fraction of particles sized over 5 and 20 µm.

Size distributions from vertical profiles were also used to calculate size resolved mass loadings. Very large values were found, between 300 to 1,000 µgm$^{-3}$ in the SAL for low to moderate AODs (0.2 to 0.5), and up to 4600 µgm$^{-3}$ in an intense dust event. Only 14% of mass was found to reside in sizes beneath 2.5 µm, while 60% resided in sizes larger than diameters of 5 µm and 0-12% resided at sizes above 20 µm. The latter two diameters are important, representing sizes where models begin to underestimate the size distribution and where models typically exclude larger particles respectively (Kok et al., 2017; Huneeus
et al., 2011). Thus it is clear that a large proportion of mass resides in the larger size ranges, which will impact biogeochemical cycles in models if underestimated. Dust Mass Paths (DMPs) were also calculated for these profiles giving values between 0.2 to 2.4 gm$^{-2}$ (mean of 1.0 gm$^{-2}$), in agreement with satellite-derived values of Evan et al. (2014) within error bounds.

Analysis of four filter samples provided information on size-resolved aspect ratio, composition, and refractive indices. Modal
aspect ratios were 1.2 to 1.4, lower than typically found in the literature. In the SAL, alumino-silicate particles dominated the composition at sizes above 0.5 µm, followed by quartz, although sulfates and sea salt were present in significant quantities at sizes beneath 0.5 µm. In the MBL the situation was similar, with particles sized over 1 µm being predominantly dust, though the contribution of sulfates and sea salt extended up to 1 µm diameter. The iron rich fraction was small in the SAL (0.5 to 2.5% by volume fraction), and even smaller in the MBL (0 to 0.9%). Although iron content varied with particle size, there was
no consistent behaviour across the small number of samples analysed.

Full spectral complex refractive indices were calculated from the filter samples. At 550 nm, the imaginary part of the refractive index k$^{550}$ for the full PSD was 0.0012 to 0.0030 in the SAL, strongly influenced by the volumetric iron content. Real parts were relatively low at 1.48 due to the low real values contributed by the literature kaolinite data. In general, refractive indices
at 550 nm calculated from two different methods agreed well. For the full spectrum of data, iterated k$^{550}$ values representing the accumulation mode covered 0.0005 to 0.0025 with a modal value of 0.001. Refractive indices calculated from composition data and internal mixing assumptions were found to overestimate absorption, while external mixing assumptions provided the best agreement with observations.

SSAs for the accumulation mode were measured, and SSAs for the full size distribution were calculated using measured size distributions and derived refractive indices at 550 nm. Within the SAL, measured SSAs at 550 nm for the accumulation mode were 0.93-0.98 (mean 0.97), and calculated values for the full size distribution dropped to 0.91-0.98 (mean 0.95). During AER-D the SSA still showed a reasonable amount of variability. The contribution of both PSD and k to the variability of SSA was

investigated. Both the shape of the PSD and $d_{eff}$ varied little, despite total dust concentrations being variable. Therefore during AER-D, variability in PSD did not have a strong effect on SSA. This allowed variations in composition (via the imaginary part of the refractive index) to control the variability of the SSA. This contrasts to Fennec where size was strongly the controlling factor. Liu et al. (2018) show that hematite variability within the accumulation mode was an important control on SSA during ICE-D. Within the MBL, aerosol in the accumulation mode was extremely scattering with SSA values above 0.99. However,

once accounting for the coarse mode particles and coarse mode-specific composition, SSAs within the MBL were found to be more absorbing and representative of mineral dust.

Over the Atlantic, a significant coarse mode of dust is still present, and contributes to the overall optical properties of dust. Particles larger than expected from sedimentation processes alone are found. Additionally, the transport of mass is dominated

by the larger particles, which is important to biogeochemical cycles via deposition of nutrients to the ocean. However, we find that variability in the optical properties is controlled principally by the variability in composition. Therefore in order to appropriately model the transport of dust and its associated optical properties and mass impacts, dust models must attempt to capture both the broad size range of particles detected by measurements, and also the variability in composition, particularly that from absorbing iron oxides.

**Data Availability**

Facility for Airborne Atmospheric Measurements (2015): UK ICE-D: atmospheric measurements dataset collection. Centre for Environmental Data Analysis, July 2018. http://catalogue.ceda.ac.uk/uuid/d7e02c75191a4515a28a208c8a069e70.

**Appendix: Acronyms**

2DS Two Dimensional Stereo Probe

APS Aerodynamic Particle Sampler

CAS-DPOL Cloud and Aerosol Spectrometer with Depolarization Detection

CDP Cloud Droplet Probe

CIP Cloud Imaging Probe

FSSP Forward Scattering Spectrometer Probe

OAP Optical Array Probe

OPC Optical Particle Counter

PCASP Passive Cavity Aerosol Spectrometer Probe

SID Small Ice Detector

SLR Straight and Level Run

## 5 Acknowledgements

Airborne data from the BAe146 were obtained using the BAe-146-301 Atmospheric Research Aircraft operated by Directflight Ltd and managed by FAAM, which is a joint entity of the NERC and the UK Met Office. ICE-D was supported by NERC grant number NE/M001954/1 and the Met Office. SAVEX-D flights were funded by EUFAR TNA (European Union Seventh Framework Programme grant agreement 312609) and other AER-D flights were funded by the Met Office. C.L.R. 
10 acknowledges NERC support through Independent Research Fellowship NE/M018288/1. SAVEX-D was supported by projects PROMETEUII/2014/058 and GV/2014/046 from the Valencia Autonomous Government, and CGL2015-70432-R from the Spanish Ministry of Economy and Competitiveness - European Regional Development Fund. BJM, JBM and HCP thank the European Research Council (ERC) (240449 ICE and 648661 MarineIce) for funding. CEDA are acknowledged for their efforts in storing and archiving ICE-D FAAM data. We thank Claudia Di Biagio for providing mineral refractive index 
15 data and useful discussions, Konrad Kandler for providing calcite refractive index data, Helen Dacre for dust transport discussions and Graeme Nott for discussions about wing probe flow distortions.

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

## Tables and Figures

| Campaign | Acronym | Fieldwork Date | Location | Measurement upper size limit, µm | Instrument type | In-cabin or wing-mounted | Details | Publication |
|---|---|---|---|---|---|---|---|---|
| Dust And Biomass burning Experiment | DABEX | 2006 | Niger | 10 | OPC | In-cabin | PCASP-X, behind a counter-flow virtual impactor with significant pipework; loss of majority of coarse particles | Osborne et al. (2008) |
| | | | | 10 | Filter samples | In-cabin | Inlet restricted measurements to 35% of coarse mode (d>1.4 µm) | Chou et al. (2008) |
| Dust Outflow and Deposition to the Ocean 2 | DODO2 | Aug 2006 | Tropical Eastern Atlantic | 40 | OPC | Wing-mounted | CDP measurements on a few flights only; otherwise size distributions up to 3 µm | McConnell et al. (2008) |
| African Monsoon Multidisciplinary Analysis | AMMA | Jun-Jul 2006 | Niger and Benin | 20 | OPC | In-cabin | Grimm OPC behind isokinetic inlet with 50% passing efficiency at 9 µm | Formenti et al. (2011a) |
| NASA AMMA | NAMMA | Aug-Sep 2006 | Tropical Eastern Atlantic | 5 | APS | In-cabin | APS behind an inlet with 50% sampling efficiency at 5 µm | Chen et al. (2011) |
| Saharan Mineral Dust Experiment 1 | SAMUM1 | May-Jun 2006 | Morocco | 30/100 | OPCs | Wing-mounted | FSSP-300/FSSP-100 | Weinzierl et al. (2009) |
| Geostationary Earth Radiation Budget Intercomparison of Longwave and Shortwave radiation | GERBILS | Jun 2007 | Mali, Southern Mauritania | 60 | OPC | Wing-mounted | SID-2. PSDs represent aged, transported dust events with light dust loadings | Johnson and Osborne (2011) |
| Saharan Mineral Dust Experiment 2 | SAMUM2 | Jan-Feb 2008 | Tropical Eastern Atlantic | 30 | OPC | Wing-mounted | FSSP-300 | Weinzierl et al. (2011) |
| Fennec – The Saharan Climate System | Fennec | Jun 2011 | Mali, Mauritania | 50/60/930 | OPCs and OAPs | Wing-mounted | CDP/SID2/CIP15 | Ryder et al. (2013b) |
| Aerosol Direct Radiative Impact on the regional climate in the MEDiterranean region | ADRIMED | Jun-Jul 2013 | Mediterranean Sea | 20 | OPC | Wing-mounted | FSSP-300 | Denjean et al. (2016) |
| Saharan Aerosol Long-range Transport and Aerosol-Cloud-Interaction Experiment | SALTRACE | Jun-Jul 2013 | Tropical Western Atlantic | 50/100 | OPCs | Wing-mounted | CAS-DPOL/FSSP-100. Some measurements additionally taken over the Eastern Tropical Atlantic | Weinzierl et al. (2017) |
| AERosol Properties – Dust | AER-D | Aug 2015 | Tropical Eastern Atlantic | 100 | OPCs and OAPs | Wing-mounted | CDP, CIP15 and 2DS | This article |

**Table 1: Airborne campaigns measuring size distributions of Saharan mineral dust since 2006, showing maximum particle size measured and size restrictions by inlets where instruments were located inside the aircraft cabin. OPC size ranges are nominal diameters. See text for acronym details. Instrument acronyms can be found in the Appendix.**

| Flight Number | Date | Times of in-situ sampling (UTC) | Accumulation Mode 550 nm AOD | General Flight Aims and Conditions |
|---|---|---|---|---|
| b920 | 7 Aug 2015 | 15:00 – 17:00 | 0.4 | In-situ and remote sensing, CATS underflight |
| b923 | 12 Aug 2015 | | 1.8* | High-level remote sensing of dust, mapping of intense dust event, Cape Verde to Fuerteventura, Canary Islands |
| b924 | 12 Aug 2015 | 15:30-16:30 | 2.5 | In-situ and remote sensing of intense dust event, Fuerteventura, Canary Islands to Cape Verde. |
| b928 | 16 Aug 2015 | 15:00-18:00 | 0.8 | SAVEX-D flight 1; in-situ and remote sensing close to Praia ground site |
| b932 | 20 Aug 2015 | 11:00-12:00 | 0.7 | In-situ and remote sensing |
| b934 | 25 Aug 2015 | 15:00-17:45 | 0.6 | SAVEX-D flight 2; in-situ and remote sensing close to Sal ground site |

Table 2: Dates of AER-D flights and times of intensive in-situ sampling SLRs, and accumulation mode AOD at the region of in-situ sampling. *No in-situ sampling, AOD is provided at the Canary Islands.

| Instrument | Abbreviation | Wavelength, nm | Scattering angle, degrees (primary, secondary) | Nominal size range measured, μm diameter | Corrected size range, μm diameter | Measurement method |
|---|---|---|---|---|---|---|
| Passive Cavity Aerosol Spectrometer Probe 100-X | PCASP | 632.8 | 35-120, 60-145 | 0.1-3.0 | 0.12-3.02 | Light scattering |
| Cloud Droplet Probe | CDP | 658 | 1.7-14 | 3-50 | 3.4-95.4 | Light scattering |
| Cloud Imaging Probe 15 | CIP15 | 642 | n/a | 15-930 | n/a | Light Shadowing |
| Two-Dimensional Stereo Probe | 2DS | | n/a | 10-1280 | n/a | Light Shadowing |
| Cloud Imaging Probe 100 | CIP100 | | n/a | 100-6200 | n/a | Light Shadowing |

Table 3: Wing mounted Size Distribution instrumentation operated during AER-D.

| Flight | Uplift number | Event driving uplift | Time and date of uplift | Uplift longitude centre | Uplift latitude centre | Age at aircraft sampling, days |
|--------|---------------|----------------------|-------------------------|-------------------------|------------------------|--------------------------------|
| b920 | n/a | MCS and haboob | 2 Aug 15:00 to 3 Aug 07:00 | 3.0 | 23.5 | 3.9 to 4.6 |
| b923/b924 | n/a | MCS and haboob | 10 Aug 10:00 to 10 Aug 19:00 | -1.0 | 21.0 | 1.9 to 2.2 |
| b928 | 1 | Convection under widespread patchy cloud | 13 Aug 12:00 to 14 Aug 10:00 | -8.5 | 23.0 | 2.2 to 3.1 |
| | 2 | MCS and haboob | 14 Aug 19:00 to 15 Aug 05:00 | -2.2 | 20.4 | 1.4 to 1.8 |
| b932 | 1 | MCS and haboob | 17 Aug 10:00 to 18 Aug 01:00 | -1.5 | 21.5 | 2.4 to 3.0 |
| | 2 | Small scale local convection | 18 Aug 12:00 to 18 Aug 14:00 | -5.0 | 24.5 | 1.9 to 2.0 |
| b934 | 1 | Small scale convection and haboob | 22 Aug 21:00 to 23 Aug 03:00 | -1.3 | 22.5 | 2.6 to 2.9 |
| | 2 | Small scale convection and haboob | 23 Aug 22:00 to 24 Aug 01:00 | -6.0 | 20.0 | 1.6 to 1.7 |
| | 3 | Small scale convection and haboob | 24 Aug 20:00 to 24 Aug 23:00 | -13.0 | 19.5 | 0.7 to 0.8 |

Table 4: Details of dust uplift events determined from SEVIRI RGB imagery driving dust sampled by the aircraft. Uplift numbers correspond to primary, secondary or tertiary uplift, also indicated in Figure 1.

| | | | Mode 1 | Mode 2 | Mode 3 | Mode 4 |
|---|---|---|---|---|---|---|
| **SAL** | $d_{pg}$ | Mean | 0.105 | 0.851 | 1.580 | 32.527 |
| | $\sigma_g$ | | 2.200 | 1.181 | 1.928 | 1.528 |
| | $N_{tot}$ | | $3.91 \times 10^2$ | $8.39 \times 10^0$ | $1.16 \times 10^1$ | $1.38 \times 10^{-4}$ |
| | $d_{pg}$ | Min | 0.142 | 0.838 | 2.176 | 10.643 |
| | $\sigma_g$ | | 1.658 | 1.262 | 1.585 | 1.300 |
| | $N_{tot}$ | | $1.14 \times 10^2$ | $2.80 \times 10^0$ | $1.52 \times 10^0$ | $4.44 \times 10^{-3}$ |
| | $d_{pg}$ | Max | 0.089 | 0.576 | 1.571 | 15.421 |
| | $\sigma_g$ | | 2.200 | 1.500 | 1.957 | 1.877 |
| | $N_{tot}$ | | $1.14 \times 10^3$ | $2.32 \times 10^2$ | $5.47 \times 10^1$ | $3.75 \times 10^{-3}$ |
| **MBL** | $d_{pg}$ | Mean | 0.148 | 0.487 | 3.675 | 7.651 |
| | $\sigma_g$ | | 1.437 | 1.900 | 1.392 | 2.000 |
| | $N_{tot}$ | | $3.14 \times 10^2$ | $1.42 \times 10^1$ | $1.38 \times 10^0$ | $1.08 \times 10^{-2}$ |
| | $d_{pg}$ | Min | 0.133 | 0.686 | 3.288 | 10.457 |
| | $\sigma_g$ | | 1.483 | 1.500 | 1.500 | 1.300 |
| | $N_{tot}$ | | $1.35 \times 10^2$ | $2.39 \times 10^0$ | $6.60 \times 10^{-1}$ | $1.27 \times 10^{-3}$ |
| | $d_{pg}$ | Max | 0.151 | 0.458 | 3.144 | 7.651 |
| | $\sigma_g$ | | 1.423 | 1.872 | 1.491 | 2.000 |
| | $N_{tot}$ | | $4.86 \times 10^2$ | $2.81 \times 10^1$ | $2.88 \times 10^0$ | $2.32 \times 10^{-2}$ |

**Table 5: Lognormal Mode Properties for the number size distribution. Diameters are given in microns, number concentrations in ambient cm$^{-3}$.**

| | SAL Mean (min, max) | MBL Mean (min, max) |
|---|---|---|
| MEC, $m^2g^{-1}$ | 0.32 (0.27,0.35) | 0.25 (0.22, 0.29) |
| SSA | 0.95 (0.91,0.98) | 0.99 (0.97,0.99) |
| g | 0.74 (0.74, 0.74) | 0.73 (0.71, 0.74) |
| $\sigma_e$, Mm$^{-1}$ | 220 (38, 1148) | 77 (27, 139) |
| $d_{eff,}$ µm | 4.0 (3.6, 4.7) | 4.6 (3.4, 5.5) |
| VMD, µm | 5.5 (5.0, 6.3) | 6.0 (5.7, 6.3) |

**Table 6: Optical and Size Properties for the AER-D SAL and MBL campaign minimum, mean and maximum: Optical properties of Mass Extinction Coefficient (MEC), single scattering albedo (SSA), asymmetry parameter (g) and Extinction coefficient ($\sigma_e$) are given at 550nm. Effective diameter ($d_{eff}$) and Volume Median Diameter (VMD) are calculated directly from the 2DS XY PSD; optical properties are calculated using the same PSDs and iterated RIs for each SLR.**

| Size range, µm diameter | Sample ID | B920 R2 | B928 R2 | B920 R5 | B934 R6 |
|---|---|---|---|---|---|
| | Layer type | MBL | MBL | SAL | SAL |
| FULL PSD | m | 1.46 | 1.48 | 1.48 | 1.48 |
| | k | 0.0004 | 0.0005 | 0.0030 | 0.0012 |
| ACC PSD | m | 1.47 | 1.49 | 1.48 | 1.48 |
| | k | 0.0014 | 0.0010 | 0.0023 | 0.0023 |
| 0.1-0.5 | m | 1.49 | 1.54 | 1.52 | 1.53 |
| | k | 0.0002 | 0.0005 | 0.0021 | 0.0009 |
| 0.5-1.0 | m | 1.48 | 1.50 | 1.49 | 1.51 |
| | k | 0.0007 | 0.0004 | 0.0026 | 0.0039 |
| 1-2.5 | m | 1.47 | 1.48 | 1.48 | 1.48 |
| | k | 0.0015 | 0.0010 | 0.0023 | 0.0023 |
| 2.5-5.0 | m | 1.47 | 1.48 | 1.49 | 1.49 |
| | k | 0.0003 | 0.0016 | 0.0031 | 0.0027 |
| 5-10 | m | 1.45 | 1.48 | 1.49 | 1.47 |
| | k | 0.0004 | 0.0003 | 0.0034 | 0.0003 |
| 10-40 | m | 1.47 | 1.47 | n/a | n/a |
| | k | 0.0003 | 0.0003 | n/a | n/a |

**Table 7: Complex refractive indices at 550 nm derived from filter sample composition assuming internal mixing. Real part (m) and imaginary part (k). Values are given for the full size distribution (full PSD), accumulation mode PSD (d<2.5 µm), and as a function of diameter range given.**

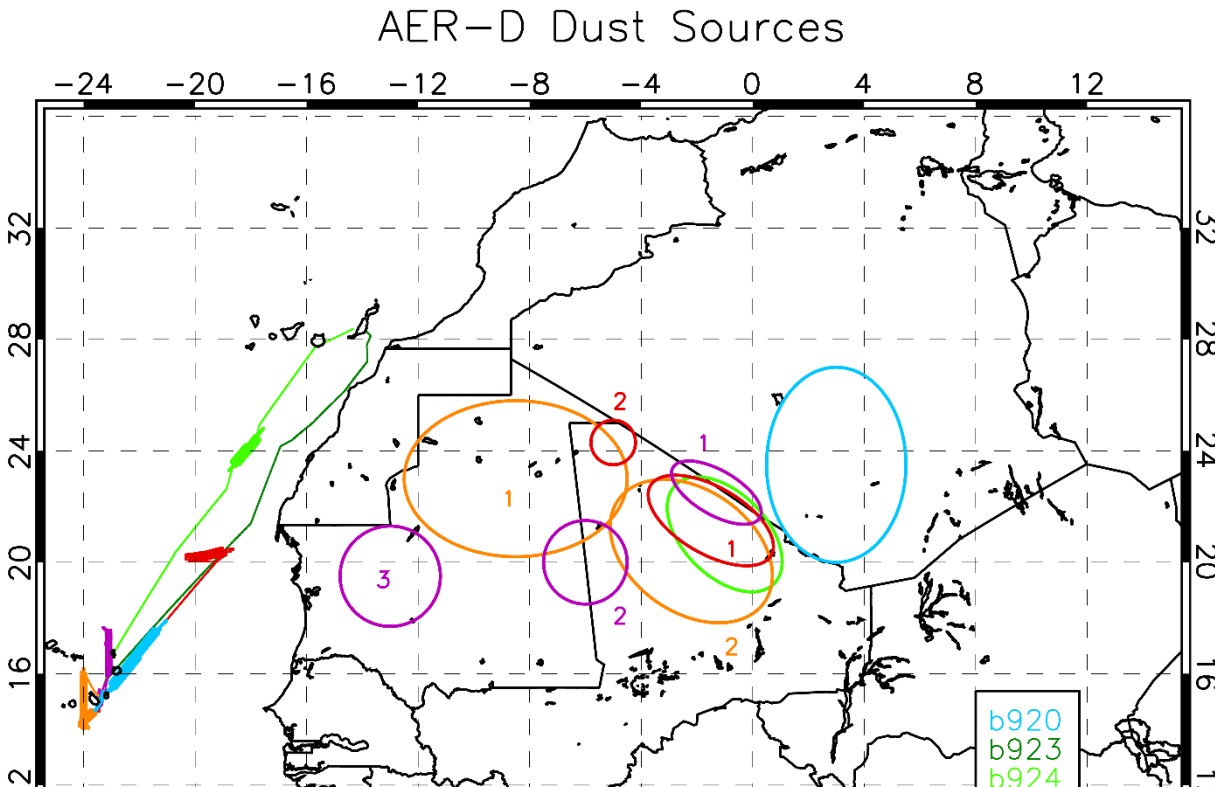

**Figure 1: Map showing the location of the AER-D flights, based out of Praia on Cape Verde. Thin lines show full flight tracks, bold sections indicate in-situ sampling SLRs analysed here. Note that within each flight there were several SLRs at different altitudes, which overlay each other here. Circles indicate dust source locations. Numbers indicate primary, secondary and tertiary dust uplift (see Table 4) events. Note that flights b923/b924 sampled the same dust event.**

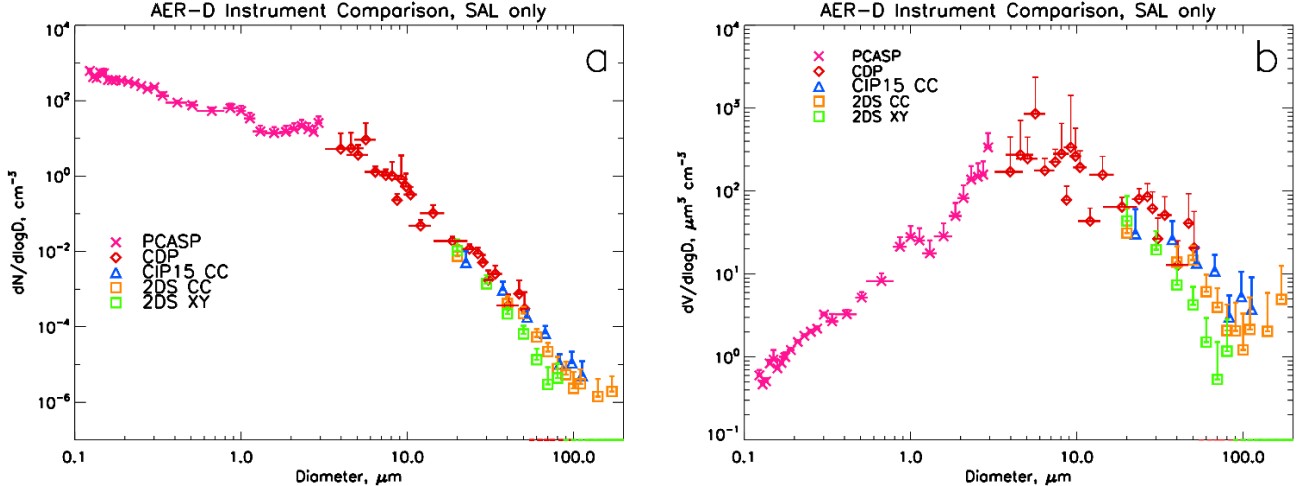

**Figure 2: Comparison of mean size distributions in the SAL during AER-D from different instruments and different sizing metrics. For clarity only upper error bounds are shown. Horizontal error bars represent maximum bin width due to uncertainties in both bin centre and bin width.**

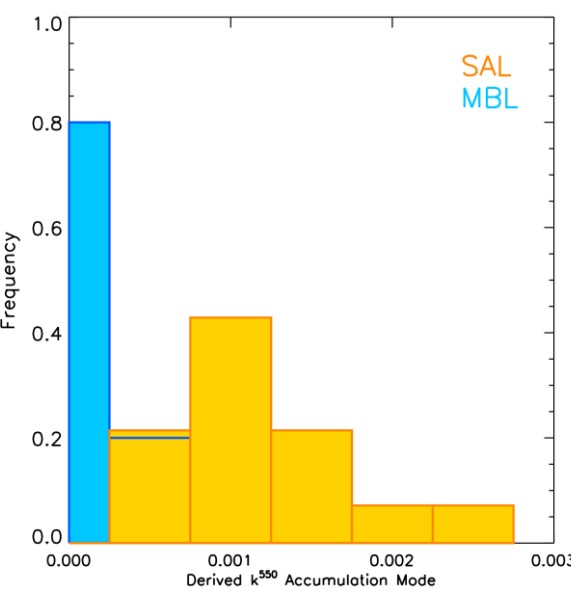

**Figure 3: Histogram of derived imaginary part of the refractive index at 550 nm ($k^{550}$) from iterations for the accumulation mode, shown separately for SLRs in the SAL and MBL.**

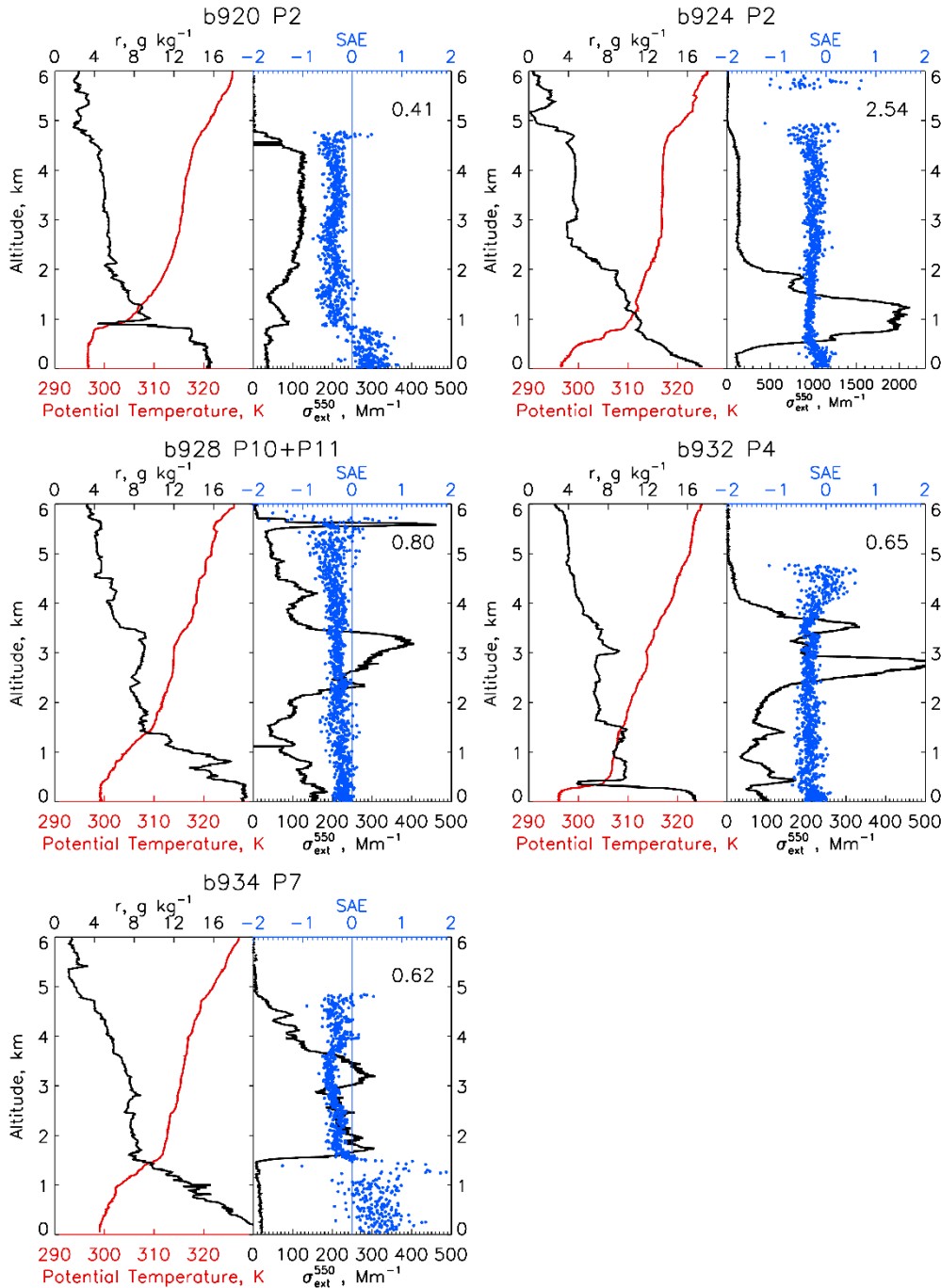

**Figure 4: Vertical structure observed during the 5 flights from in-situ measurements during aircraft profiles, in the region where the SLRs were performed. Potential temperature (K; red), water vapour mixing ratio (r, g/kg, black), accumulation mode extinction (Mm$^{-1}$, black, note different x-axis range for b924) and scattering angstrom exponent (SAE) for the accumulation mode between 450 and 700nm (blue). Numbers at the top right of each panel indicate accumulation mode AOD at 550nm.**

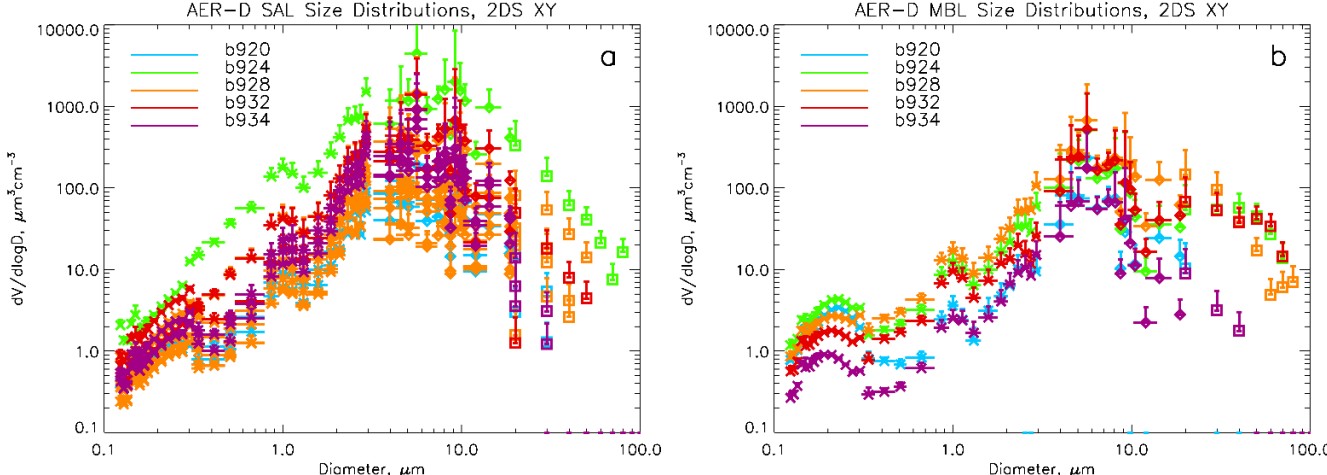

Figure 5: Size distributions for SLRs (a) in the SAL and (b) in the MBL, for the PCASP, CDP and 2DS XY. Errors combine systematic and random errors. For clarity only upper error bounds are shown. The following numbers of SLRs were performed per flight in the SAL and are shown in panel (a): b920 (2), b924 (1), b928 (6), b932 (2), b934 (3). One SLR per flight was performed in the MBL as shown in panel (b).

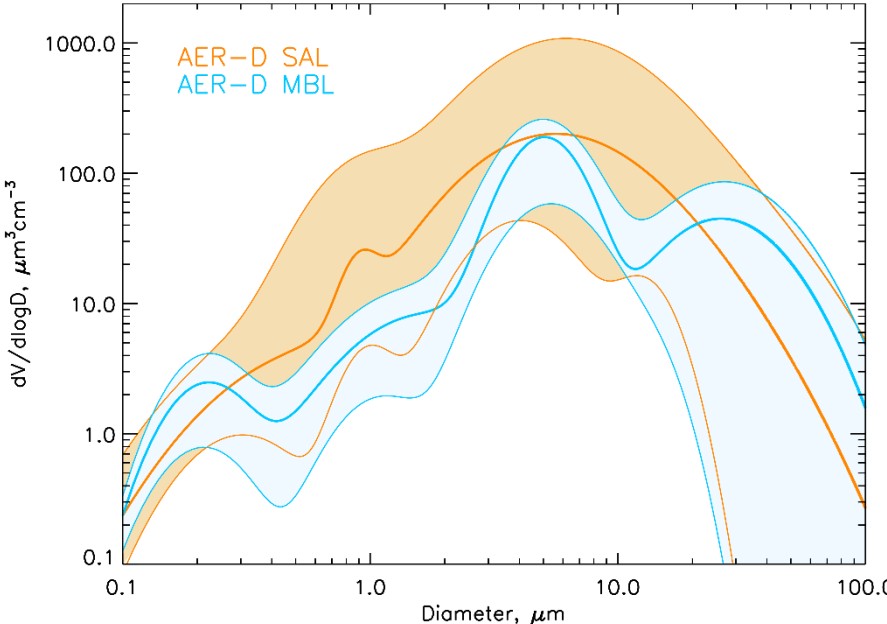

Figure 6: AER-D mean logfit size distribution from the MBL (blue) and SAL (orange). Shading indicates the range between minimum to maximum values, and central solid line shows the mean.

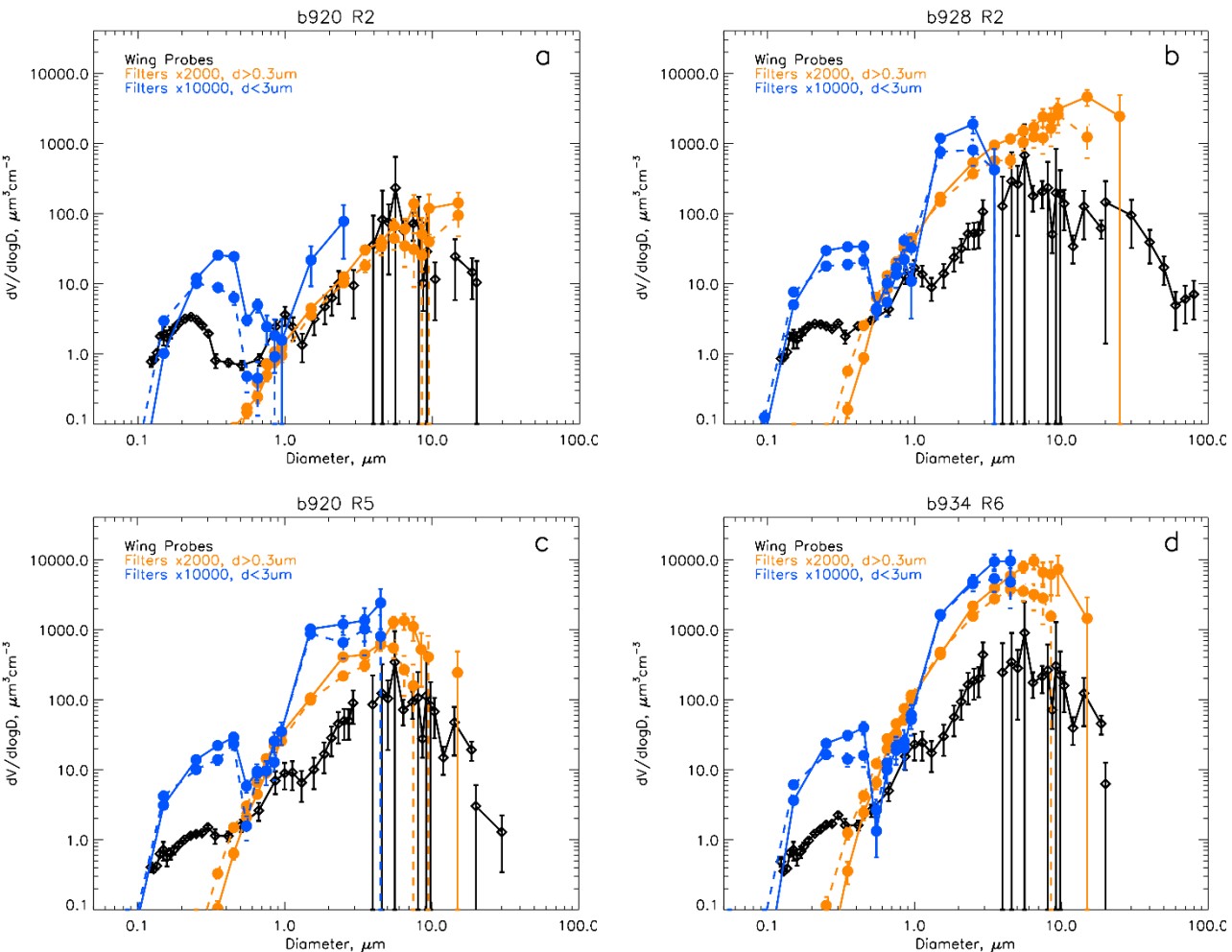

**Figure 7: Comparison of wing-probe size distributions with size distributions from filter samples at 2 magnifications (2,000 and 10,000) for 4 SLRs. (a) and (b) show PSDs in the MBL without (a) and with (b) giant mode present; (c) and (d) show PSDs in the SAL. For filter PSDs, solid lines indicate volume distribution calculated assuming spherical shape using an area-equivalent diameter. Dashed lines indicate volume distribution calculated using a height:maximum axis ratio of 1:3 (see text for more details).**

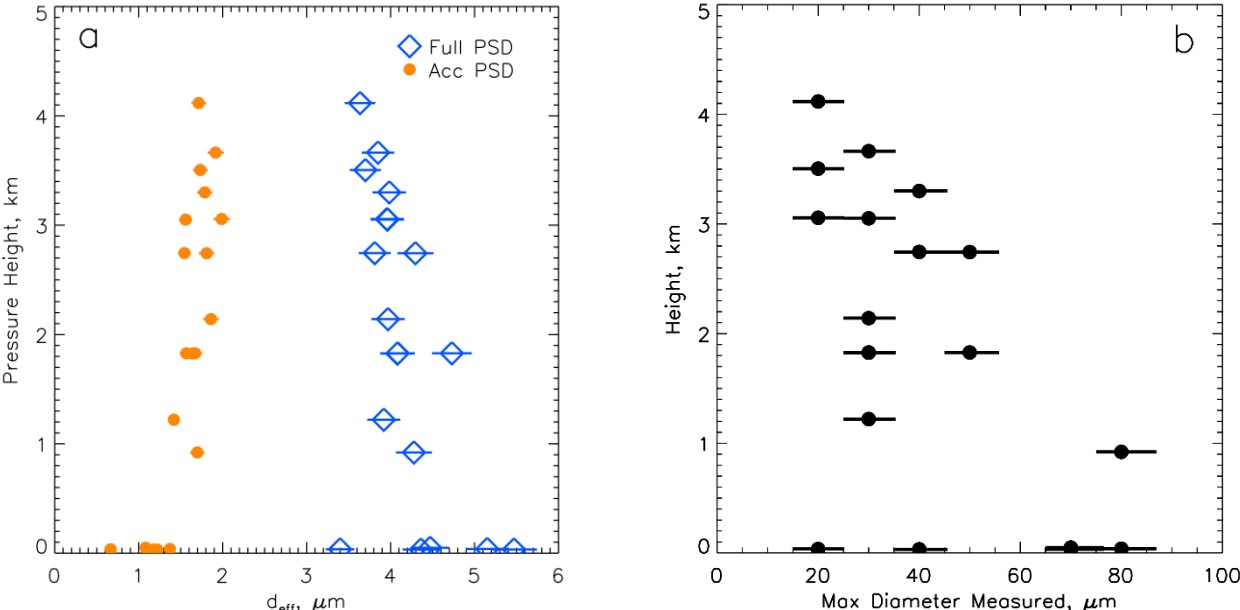

**Figure 8: Variation of size with altitude for SLRs: (a) effective diameter; (b) maximum diameter measured by the 2DS XY instrument. Orange circles represent accumulation mode only, blue diamonds represent the full PSD.**

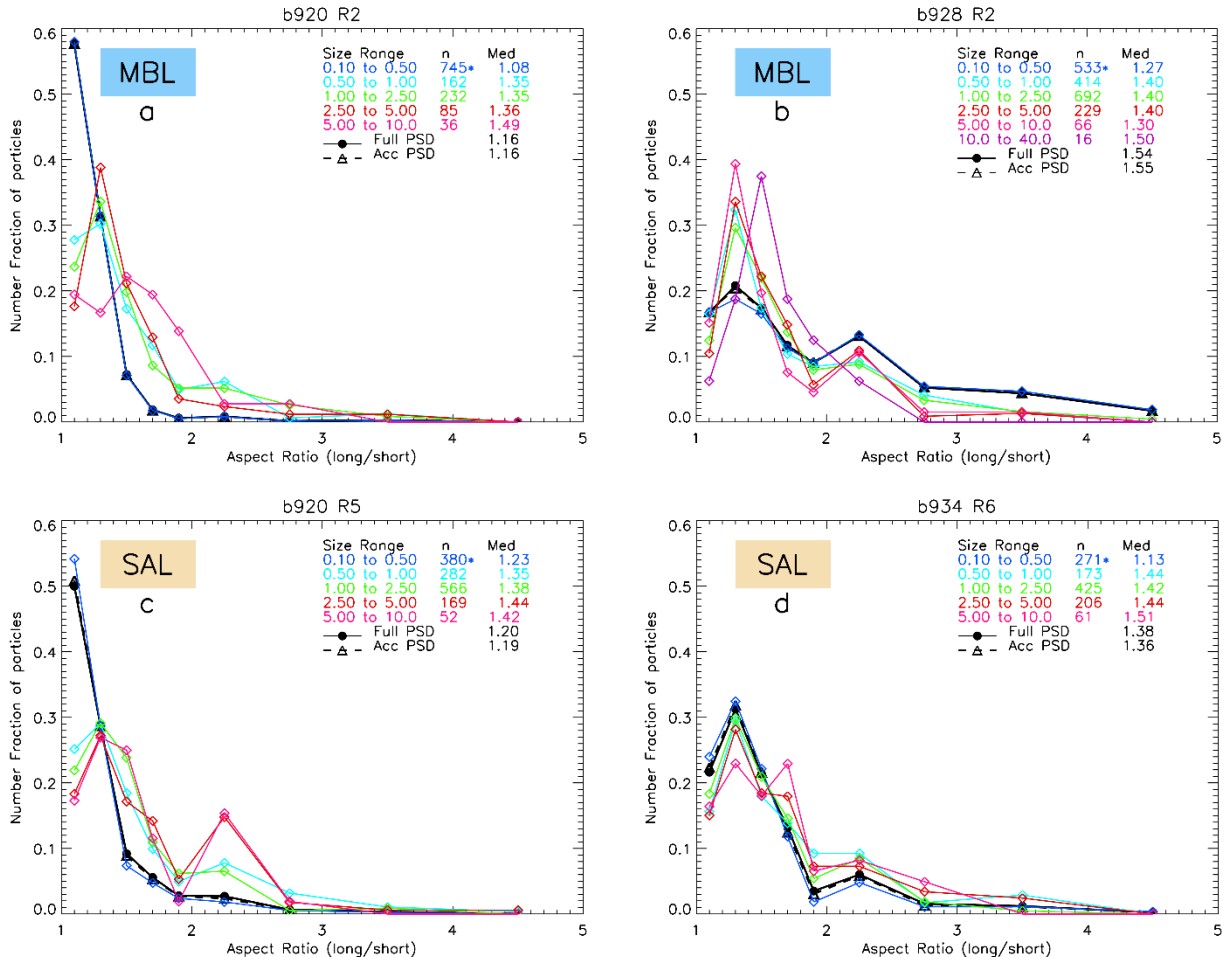

**Figure 9: Number fraction of particles as a function of aspect ratio from filter sample analysis for the same 4 SLRs as shown in Figure 7, as a function of size (colours), for the full PSD (0.1-40 µm, solid black), and accumulation mode PSD (0.1-2.5 µm). * indicates numbers were scaled-up from value shown to allow for different substrate areas at the higher magnification. Top row are samples from the MBL without (a) giant particles and with (b) giant particles. Bottom row are SAL samples. Data are not shown where the number of particles in a size range is under 10. In each panel the size range (microns diameter) and the associated number of particles counted and median aspect ratio are shown.**

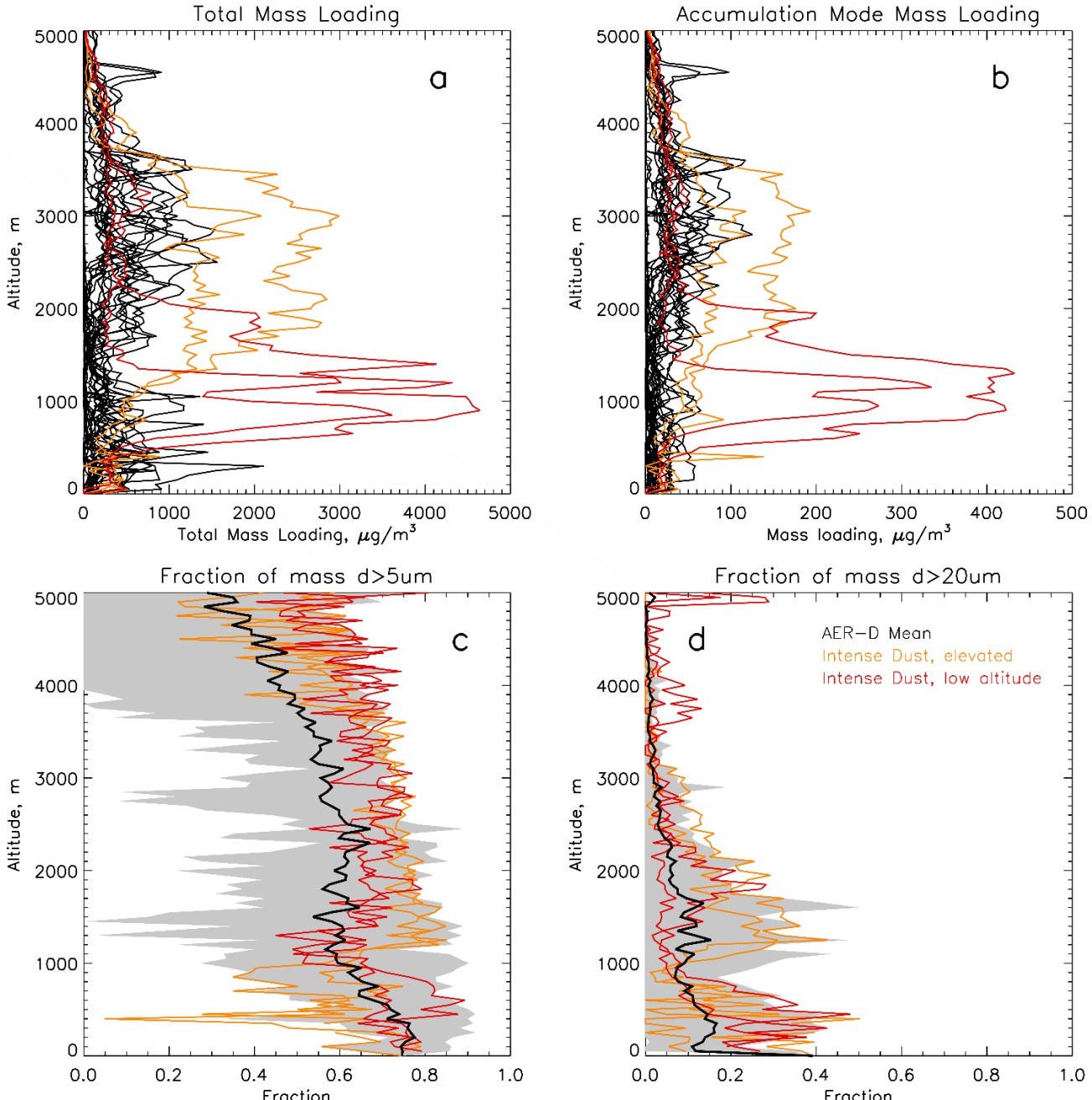

**Figure 10: Profiles of aerosol mass loading calculated from in-situ size distributions assuming a dust density of 2.65gcm$^{-3}$. (a) Total mass loading for the full size distribution; (b) accumulation mode mass loading (d<2.5 μm); (c) Fraction of mass at sizes greater than 5 μm diameter; (d) fraction of mass at sizes greater than 20 μm diameter. Black lines in (a) and (b) represent all AER-D profiles under low-medium AODs, red/orange represent the intense dust event with AODs ~2.0 (b923/b924) separated by locations where the dust was elevated (orange) or at lower altitudes (red). In (c) and (d) grey shading represents AER-D 10th to 90th percentile range; black line represents AER-D mean. Dust Mass Path values are given in the text.**

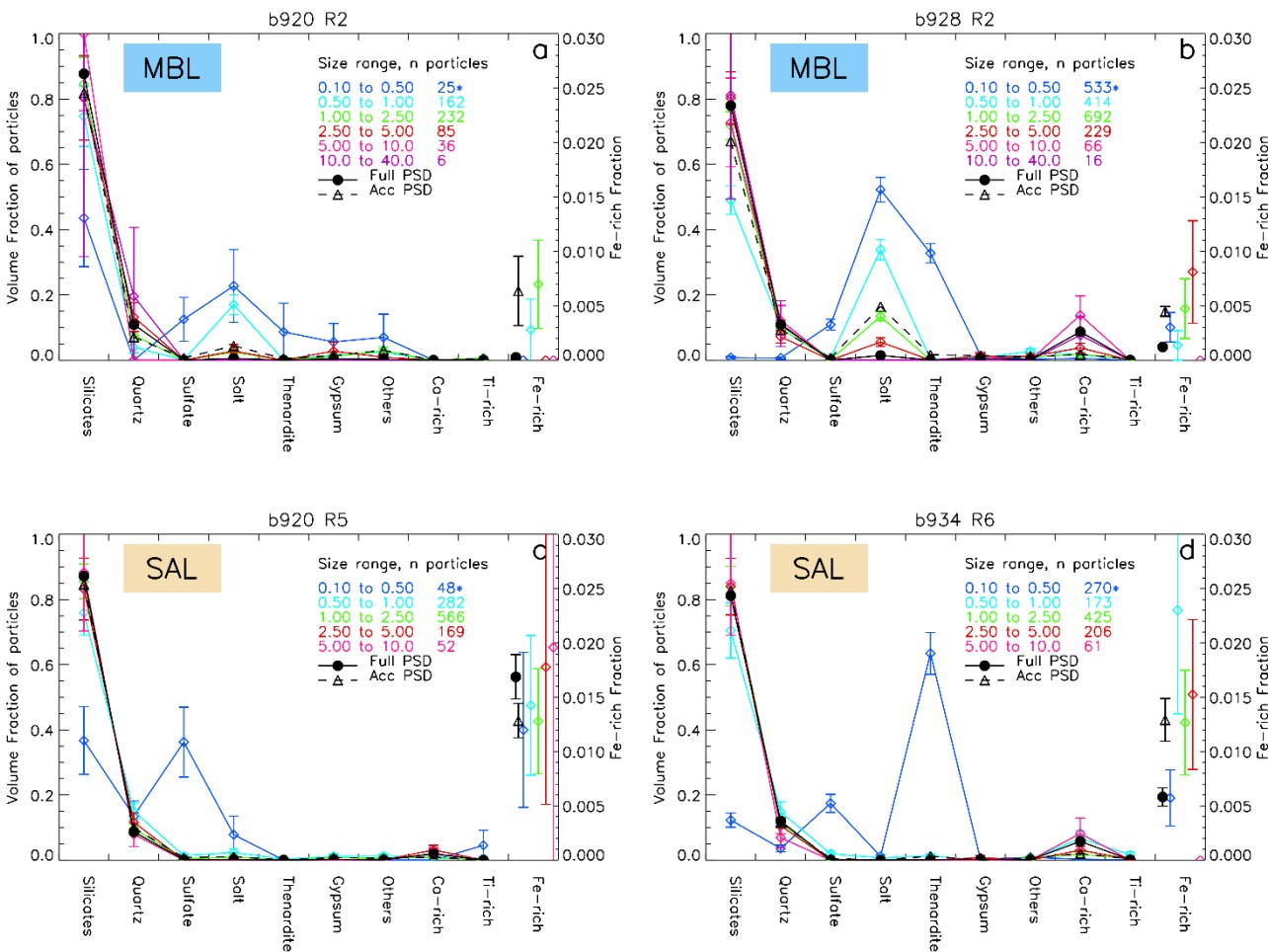

**Figure 11 Composition as volume fraction from filter samples in the MBL (top panels) and SAL (bottom panels). B928 R2 (top right) contained a substantial giant mode in the MBL (e.g. see Figure 5 and Figure 7). Individual panels indicate number of particles sampled; * indicates numbers were scaled-up from value shown to allow for different substrate areas at the higher magnification. Fe-rich volume fraction is provided on the right axis with different size ranges offset. Data is also shown as bulk properties for the full PSD and accumulation mode (d<2.5 μm) and size resolved. Data is only shown when sample size is greater than 5. Error bars are counting uncertainties.**

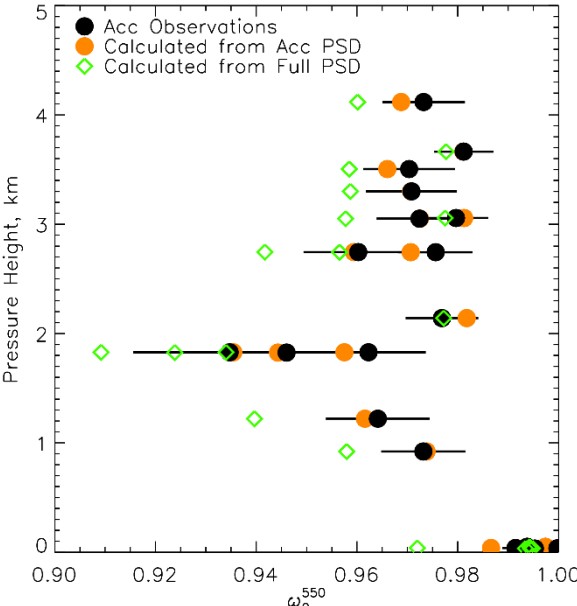

**Figure 12: Variation of SSA at 550nm with altitude for SLRs. Black circles indicate direct measurements taken in-cabin behind inlets and therefore represent the accumulation mode (ACC PSD) only. Orange circles indicate calculated properties, representative of the same conditions in-cabin behind inlets for the ACC PSD. Green diamonds indicate values calculated using the 2DS-XY FULL PSD covering size ranges 0.1 to 200 um. Green and orange data points are from Mie calculations, black represents measurements.**

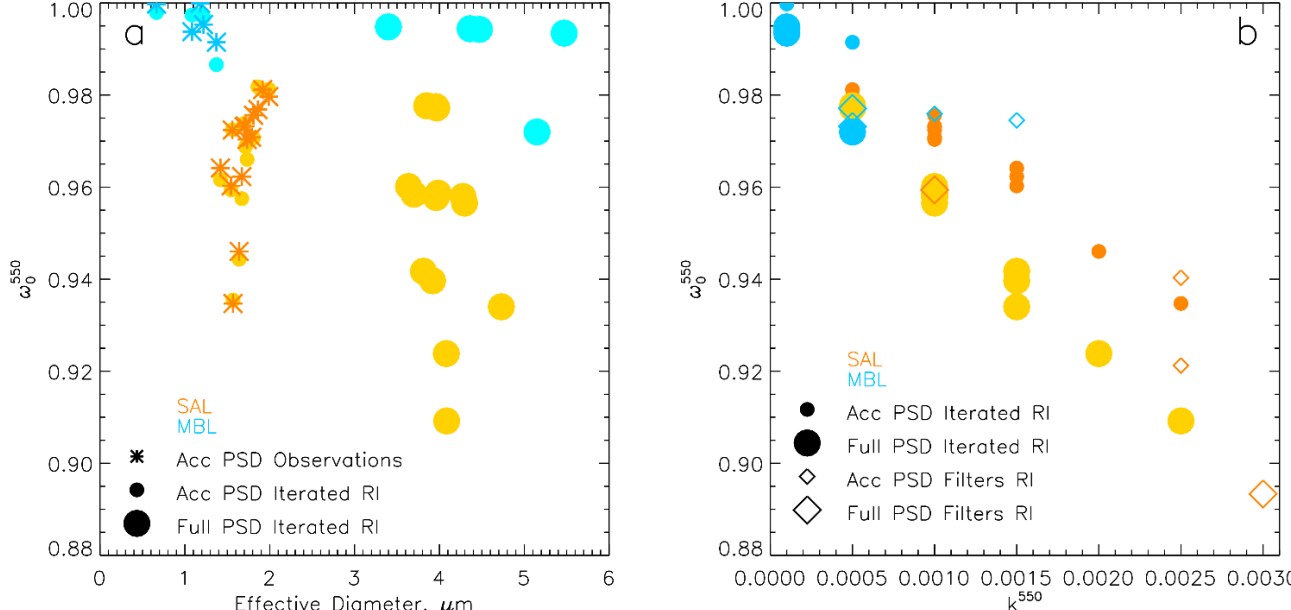

**Figure 13: Contribution to single scattering albedo from particle size (a: SSA vs d$_{eff}$ ) and from composition (b: SSA vs k$^{550}$), separated by SLRs in the SAL (orange and yellow) and MBL (blue). Small data points represent the ACC PSD only, large data points represent the FULL PSD. Asterixes in (a) are direct observations behind inlets. Circles represent calculations using the RI derived from Mie iterations; diamonds represent calculations using RI derived from composition data from filter samples assuming internal mixing (4 samples). For the FULL PSD with the iterated RI, the same accumulation mode RI is extended to the coarse mode. For the filters RI, size-specific RI is used for the accumulation mode and the full PSD (as given in Table 7).**

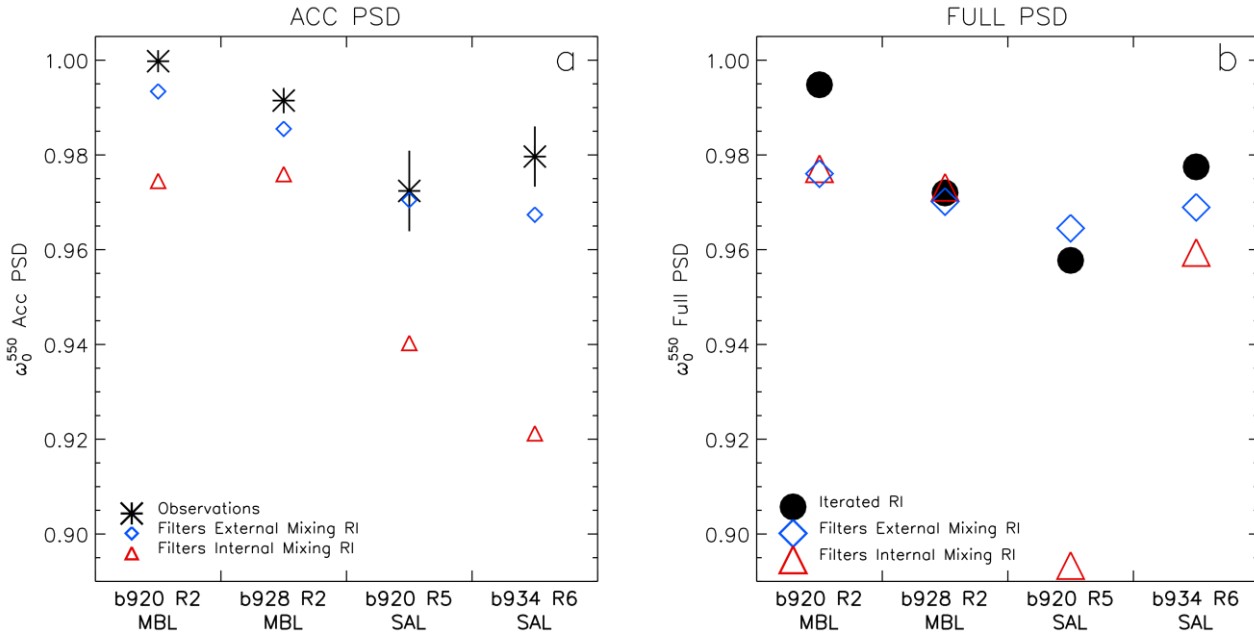

**Figure 14: Comparison of SSAs calculated using different RI methods for (a) the ACC PSD; (b) the FULL PSD. Observations are only available for the accumulation mode, shown in (a). SSA is calculated using RI derived from filter samples assuming internal and external mixing, or iterated RI.**

