# Peer review of "Coarse mode mineral dust size distributions, composition and optical properties from AER-D aircraft measurements over the Tropical Eastern Atlantic"

_Atmospheric Chemistry and Physics, 2018_

## Referee Comment (RC1) · K. Kandler (Referee) · 4 Sep 2018

Review of "Coarse mode mineral dust size distributions, composition and optical properties from AER-D aircraft measurements over the Tropical Eastern Atlantic" by Ryder et al.

The manuscript describes airborne measurements of mineral dust size distributions close to the source. Representation of aerosol size distributions inside atmospheric transport models is still based on many assumptions, and new data for evaluations and

input is highly welcome. The manuscript adds new data for the region close to the source at the beginning of the trans-oceanic transport. It also adds additional evidence to the importance of studying particles larger than 10 $\mu$m, which are frequently omitted in aerosol research, and which are difficult to study. Furthermore, optical properties for the dust aerosol are reported, which can also serve the community, e.g., as model input or for remote sensing interpretation. The authors show that variation in physical as well as compositional parameters can have a considerable impact on the optical properties, and that in their samples, the compositional variation is dominating the variation in single scattering albedo.

The topic is suitable for ACP. The paper is well-written and clearly structured. References are mostly made where appropriate, some additional comparison with previous work would enhance the general significance. Some issues - particular on the error handling - should be addressed and questions answered before publication.

==========================================

Remarks and questions

P3 L32: Are really any inlet size restriction removed? Given the high speeds at which particles are collected, a minimal deviation from perfect alignment might easily introduce boundary layers and re-circulations.

P6 L32: Should this diameter then considered as optical equivalent diameter? Please state.

P7 L10: Have experimental or theoretical approaches been used to characterize the losses?

P8 L6 and Figure 2: It looks like the sizing uncertainties from the number size distribution were not propagated into the volume size distribution (e.g., the CDP point at 20 $\mu$m seems to have a factor of 3 uncertainty in size, which should propagate to around a factor of 10 in volume, but a much smaller error bar is shown in the volume plot). Instead it looks like the same vertical error bars are shown in both plots. Please explain or modify.

Figure 2 is hard to understand also from another aspect: Either the measurement points are all at the lower boundary of the error, or the error always includes zero. If they are at the lower boundary, please explain why. In particular with a non-zero counting error (which is necessarily two-sided), this seems impossible. If the error always includes zero, then the statement in P9 L27, "agree within the error bars" is meaningless, as all data could be zero.

P8 L31 and P9 L4: if a particle was not counted for the given reasons – how was this underestimation of the concentration then accounted for? Decreasing proportionally the sample volume?

P9 L6: For what reason the projected area equivalent diameter was not used, which appears to be a common standard in shadowing techniques (i.e. light microscopy)?

P9 L11-21: Please be more explicit on the error handling. Was is done by analytical error propagation, by Monte Carlo simulation, ...? What were the distribution assumptions (Poisson?)? How was the counting error treated, if measurement points with only 4 particles were regarded? If error bars are given: do they show a certain confidence interval, one (or more) standard deviations, or maximum errors?

P9 L30-35: Are there any arguments beyond consistency with previous measurements for disposing (if I understood correctly) the CIP15 CC data? It is stated that the 2DS XY metric is better – better in terms of what? Please explain.

P10 L1: This type of effective diameter seems usually (Hinds 1999) to be called the Sauter mean diameter (apparently defined by Sauter 1926) or mean volume-surface diameter (Hinds 1999). As different effective diameters exist, I suggest referring to one of the mentioned denominations and referring in addition to a more fundamental paper (McFarquhar et al. 1998).

P10 L4-7: Given the high counting uncertainty associated with a single count of the maximum-sized particle, wouldn't there be a better option, e.g., fitting a log-normal distribution or power law into the data? At least, both mentioned approaches should be compared to allow for consistent comparison.

P11 L12 (and below): As magnifications are relative to the used screen, a measure like nm per pixel or pixels per smallest particle would be more meaningful.

P11 L14-19: Wouldn't it make more sense using the same metrics as described for the shadowing instruments above?

P11 L25-29: Please state where you draw the lines between thenardite/sulfate and gypsum/sulfate.

P16 L18-20: This behavior was described and a similar explanation was given for the same geographical region and season a while ago (Jaenicke et al. 1978), and also some model approaches seem to predict that (Garrett et al. 2003), so maybe a short comparison with previous finding would make sense here and enhance the general significance of the findings.

P16 L28-30: This finding is surprisingly similar to the volume distributions shown for Cape Verde in winter time, comparing dusty and marine situations (Kandler et al. 2011), so a comparison could show a broader relevance.

P17 L14-18: This would mean that the flow through the filter would have been ten times as high as the one used for calculating the size distributions – is this still in a physical probable range? Can there exist any aerosol concentration effects due to high velocity gradients during sampling?

P17 L22: Smoothness of the curves could be related to different size intervals. The wing probes seem to have more size intervals (with higher counting uncertainties) in Figure 7.

P18 L23-32: Probably a median aspect ratio, instead of a modal value, would make

interpretation less dependent on a single interval. In particular, as in the following the values are compared with median values from other sources, which otherwise can't be compared.

P20 L24-25: How about sea-salt reacted with sulfuric acid?

P21 L7-12: How would black carbon have been identified in this work? Were all particle images manually inspected for fractal-like structures (doesn't come clear from the method section).

P22 L20: In particular for the internal mixing, there is plenty of room with respect to complexity for calculating an effective value (Nousiainen 2009; Lindqvist et al. 2014).

P24 L6: In addition, also measured refractive indices show the dependency on iron (Moosmüller et al. 2012; Caponi et al. 2017), so it appears to be consistent in general.

Figure S2: Where does the 'step function' of the imaginary part at 700 nm wavelength derived from?

==========================================

Corrections

General: The order of the figures does not correspond with the order of their references in the text.

General: The table numbers in the text don't fit with the table numbers at the end.

P3 L8-11: including into calculation/model ?

P4 L22: Why time-of-flight? Wasn't it an SP2 instrument?

P4 23-25: While the link to the optical property measurement is clear, the reference to the ice nucleation measurements and to the modeling work doesn't seem to add something useful here.

P4 26-30: I suggest removing the paragraph, as the structure is standard.

P5 L15-22: If results from SAVEX and CATS are not discussed here, these explanations should be removed.

P8 L2: "ambiguities" instead of "singularities"?

P9 L9: What does "Instead, ... sizing metric" refer to? Aren't the previous sentences the section about the sizing metric?

P10 L8-9: "do not have a minimum detection concentration level, but at low particle concentrations the sampling statistics simply become poor": In fact, this is the detection limit. With poor sampling statistics the (counting) error becomes high in comparison with the signal. If it is decided to omit data below a certain signal/error ratio, the detection limit is introduced (which is a common procedure).

P10 L14: "PSD' size distribution" – doubling "size distribution"

P12 L10: "low SEM signal" low image contrast?

P15 L5: increased?

P16 L24: There seems to be no (b) in Figure 6.

P18 L5: In Figure 7, in most cases the volume maximum is below 10 $\mu$m, in one above (filter size distribution). "Dominating" therefore doesn't seem to be appropriate for the largest particles with respect to the mass.

P18 L12: Sentence ends with a comma.

P24 L28: Most of section 4 is rather a summary than conclusions, so the section should be termed accordingly.

Table 1: Different reference formats. Maybe explain instrument abbreviations, e.g., as addition in Table 3?

Table 3: Check caption

Table 4: different time formats between table 2 and 4 (?)

Table 5: 'IT' number format (E+01 etc.)

Figure 2: Both y axes should have the same number format.

========================================

References

Caponi, L., P. Formenti, D. Massabó, C. Di Biagio, M. Cazaunau, E. Pangui, S. Chevaillier, G. Landrot, M. O. Andreae, K. Kandler, S. Piketh, T. Saeed, D. Seibert, E. Williams, Y. Balkanski, P. Prati, J. F. Doussin (2017): Spectral- and size-resolved mass absorption efficiency of mineral dust aerosols in the shortwave spectrum: a simulation chamber study. Atmos. Chem. Phys. 17(11), 7175-7191. doi: 10.5194/acp-17-7175-2017

Garrett, T. J., L. M. Russell, V. Ramaswamy, S. F. Maria, B. J. Huebert (2003): Microphysical and radiative evolution of aerosol plumes over the tropical North Atlantic Ocean. J. Geophys. Res. 108(D1), AAC 11-11-AAC 11-16. doi: doi:10.1029/2002JD002228

Hinds, W. C. (1999): Aerosol Technology. Properties, behavior, and measurement of airborne particles. Second edition. New York, USA, Wiley Interscience.

Jaenicke, R., L. Schütz (1978): Comprehensive Study of Physical and Chemical Properties of the Surface Aerosols in the Cape Verde Islands Region. J. Geophys. Res. 83(C7), 3585-3599. doi: 10.1029/JC083iC07p03585

Kandler, K., L. Schütz, S. Jäckel, K. Lieke, C. Emmel, D. Müller-Ebert, M. Ebert, D. Scheuvens, A. Schladitz, B. Šegvić, A. Wiedensohler, S. Weinbruch (2011): Ground-based off-line aerosol measurements at Praia, Cape Verde, during the Saharan Mineral Dust Experiment: Microphysical properties and mineralogy. Tellus 63B, 459-474. doi: 10.1111/j.1600-0889.2011.00546.x

Lindqvist, H., O. Jokinen, K. Kandler, D. Scheuvens, T. Nousiainen (2014): Single scattering by realistic, inhomogeneous mineral dust particles with stereogrammetric

shapes. Atmos. Chem. Phys. 14(1), 143-157. doi: 10.5194/acp-14-143-2014

McFarquhar, G. M., A. J. Heymsfield (1998): The Definition and Significance of an Effective Radius for Ice Clouds. J. Atmos. Sci. 55(11), 2039-2052. doi: 10.1175/1520-0469(1998)055<2039:tdasoa>2.0.co;2 Moosmüller, H., J. P. Engelbrecht, M. Skiba, G. Frey, R. K. Chakrabarty, W. P. Arnott (2012): Single scattering albedo of fine mineral dust aerosols controlled by iron concentration. J. Geophys. Res. 117, D11210. doi: 10.1029/2011JD016909

Nousiainen, T. (2009): Optical modeling of mineral dust particles: A review. J. Quant. Spectrosc. Ra. 110, 1261-1279.

Sauter, J. (1926): Determining size of drops in fuel mixture of internal combustion engines. National Advisory Comitee for Aeronautics, Technical Memorandum 390.

---

## Referee Comment (RC2) · Anonymous Referee #2 · 4 Sep 2018

Ryder et al. discuss airborne measurements of Saharan dust performed between Cabo Verde and the Canary Islands in August 2015 with focus on several aspects of the dust, for example the coarse mode size distribution. I think the paper is interesting and fits well into the scope of ACP. Overall, the paper is in a useful shape but there are many, mainly minor, things the authors should improve before the paper is ready for final publication.

[In the following, e.g. page 6 and line 44 is referred to as p6l44.]

[Figure]

General comments:

Abstract: Maybe one sentence with results for the MBL could be added.

On p3l6 you define coarse and giant mode dust. However you do not follow this definition, e.g. at p7l15, p9l32, p12l27, p25l25. Please make sure that the paper is self-consistent.

Often the citation type, e.g. '(Ryder et al., 2013)' vs. 'Ryder et al. (2013)', is not correct.

You did not define the flight legs. At p11l5 you mention 'R2' for the first time, but you did not introduce this properly. In Fig. 4 an undefined 'P2' etc appears, probably also refering to flight legs. I suggest to add a proper introduction of the nomenclature and to add a table (maybe in the supplement) with more details about the flight legs considered in the paper (e.g. name of leg, start time, end time, duration, height, etc. ).

You write SSA in the text but use $\omega\_0^{550}$ in the figures. Please use only one of these.

Sometimes for the imaginary part values a 'i' was added after the value, sometimes not. This should be made consistent, perferably removing 'i' everywhere because the 'imaginary part' is a real value; see also your definition at p8l18.

Often, the main text refers to the wrong table number.

Specific comments:

p1l24: In my view 'during' should be replaced by 'at'.

p1l28: 'constituting up to 40% of dust mass': As you mention this number in the abstract, it should also be mentioned in Sect. 3.3.

p2l3f.: It is unclear what 'this complex evolution' refers to. Suggestion: '... to capture correctly both the dust composition and the size distribution including their changes during transport in order ...'

p3l27: 'preceeding the AER-D flights' comes a bit surprisingly. Please reformulate without refering to AER-D which is introduced only later.

p3l33: I think 'however' could be removed here.

p4l1: 'Mie theory conversion': it is not very clear what is meant. Please reformulate.

p5l7f.: 'The dust events sampled 550 nm AODs from 0.4 to 0.8' should be reformulated.

p5l19: 'aerosol structure' is a bit unclear. You probably mean the vertical distribution (structure) of the aerosol.

p5l20: 'nearer the ...' could be replaced by 'closer to the'.

p5l26: 'Figure 1b' does not exist.

p5l28: 'flight' could be added before b920 and b924.

p7l3: The wavelength list may be a bit confusing. Maybe you can just write the wavelengths in parentheses after the instruments?

p7l25: 'aerosol' should be replaced by 'particle' to make the sentence more general. Particles with 6.2mm are usually not aerosol particles but much more likely some kind of hydrometeors.

p8l18: PCASP and CDP do not operate at 550nm. This should be mentioned here including your assumption that the refractive index does not change between 550 nm and the instrument's wavelengths.

p8l28: You could write ' ... in two different ways, resulting in different sizing metrics.' This would help the readers in the subsequent paragraph.

p9l3f.: Were the particles rotated such that one dimension is minimized and the other maximized? Or were x and y measured for each imaged particle without such rotation?

p9l6: '... though diameters will be lower than an area-equivalent diameter for example, if the particle is an ellipse.' looks wrong. For example, assume x=1 and y=2. Then

D_XY=(x+y)/2=1.5. The area-equivalent diameter however is D_area=(1*2)ˆ0.5=1.414 which is smaller than 1.5. This could be a reason why also the mean XY method somewhat overestimates the 'real' particle size.

p9l8: I wonder if there is a reason why you don't use the mean XY method (instead of the CC method) for the CIP15 in this discussion paper?

p9l12: Which flight leg length do these 10ˆ-5 cmˆ-3 correspond to, approximately?

p9l20: It is unclear what '... errors due to bin size from ...' means. Please reformulate this sentence.

p9l28f.: 'as expected when the particles are non-spherical (section 3.2)' could be replaced by 'as excepted for dust'.

p10l9f.: This looks quite similar to p9l10ff. Maybe you could bring both together or at least refer here to the previous text (e.g. 'Therefore, we remove, as mentioned, cases where fewer ...').

p10l17: 'of around 0.2 to 1 $\mu$m' should be replaced by 'smaller than 2-3$\mu$m' when considering Fig S1.

p11l5: 'R2', 'R5' are not defined.

p11l15: The areas of the ellipses were larger than the areas of the particle 2D projections because you used circumscribed ellipses. How large is this difference? Why didn't you use the area of the projection itself to determine the area-equivalent diameter?

p12l29: Shouldn't '0.0001' be replaced by '0.0005'? Otherwise I don't understand this description.

p13l1: 'Figure 8b' shows something else.

p13l9: 'between 0.0015i to 0.0025i' doesn't fit to the mode value of 0.001.

p13l11: Why do you use 'volume fraction' here and 'number fraction' for the external mixing case (p13l30)? How big is the difference between both cases?

p15l16: I suggest to briefly discuss the difference from Marenco et al., who find maximum AODs of 2.0.

p15l28: 'each flight leg': As Fig. 5 shows only a single leg inside the SAL for each flight I assume that there was only one flight leg in the SAL for each flight? However, then Fig. 2 and 5 seem to not fit to each other as mentioned in a separate point below. In general, the legs should be described better.

p15l33: I do not really understand this sentence. Would the absense or presence of the coarse mode not always have an effect on the overall shape of the size distribution?

p16l1: 'peak volume concentration' is unclear. I suggest to write 'The peak of the volume distribution during ...'.

p16l8f: 'Figure 5b; green, orange and red'

p17l3: ' may be aligned horizontally in the atmosphere ... ': I suggest to add here a reference to Ulanowski et al. (2007), DOI:10.5194/acp-7-6161-2007, who made some simulations on this topic (see e.g. Fig. 9 of that paper).

p18l10: How does 'Particles sized over 20 $\mu$m diameter were detected in 100%' fit to Fig. 8b which shows that there are cases with D_max=20$\mu$m?

p18l14: 'Figure 8c' does not exist.

p19l4: particles

p19l10: 4643 has probably too many significant digits. I suggest 'around 4600'.

p19l12: 'decreases' is maybe the wrong word here. I suggest 'is lower'.

p19l15: 'PM2.5' in not defined. As it is used nowhere else, I suggest to just write 'the accumulation mode'.

p20l16: 'as they only include iron when detectable as single-iron particles': As far as I understand this sentence, 'they' should be replaced by 'we' and 'detectable as single-iron particles' by something like 'iron was the dominant component of a particle'.

p22l17: There is one 'is' too much.

p22l28: 'coarse mode present' could be replace by 'coarse particles'.

p22l29: 'so' could be removed.

p23l1f: The sentence could be improved by removing 'same' and adding 'also' after 'mode'.

p23l4: 'as dominate' is unclear.

p23l7: 'RI' is not defined.

p23l10: Suggestion: 'The variability of the optical properties of dust in the SAL is probably mainly determined by ...'.

p23l14: 'the variability of the' should be inserted before 'optical'.

p23l26: I suggest to write 'the variation of the SSA as function of composition, represented by k550', because this order is more logical and also better fits to Fig. 13 considering that the vertical axis usually shows the dependent variable (y=f(x)).

p24l1: 'optical property' could be replaced by 'SSA' to be more specific.

p24l13ff: It is not clear how this fits to p13l29f where you write that you use the same size distribution (only number-weighted) for all components. In addition, you could consider a size dependency not only in case of external mixtures but also in case of internal mixtures.

p24l21: 'In contrast to Fennec observations of the full PSD and associated optical properties over the Sahara,' could be removed. Maybe the information about the location could be added somewhere on line 22.

p24l30: I suggest to delete 'during August 2015' and to add instead a new sentence like 'The flights were performed in August 2015 between Cape Verde and the Canary Islands.'.

p25l7: 'to be' could be removed.

p25l14: 'Deff for the SAL the mean (minimum, maximum) was' should be reformulated.

p25l26: 'giant MBL mode particles' could be reformulated.

p25l30: calculate

p26l6: 'slightly lower' is an understatement because the 'base value' of the aspect ratio is 1.0. Then your value is only about half of the literature value.

p26l6: 'and quartz' could be removed when considering Fig. 11.

p26l28: 'was extremely scattering' should be replaced by 'was only very weakly absorbing'.

p26l33: I suggest to start a new sentence after 'dust' and to write 'Particles larger than expected from sedimentation processes alone are found.'.

Table 1: The reference style is not consistent.

Table 1: During 'SALTRACE' also the 'CAS-DPOL' instrument was used, measuring upto $50\mu$m (Weinzierl et al., 2017).

Table 2: The 'General Flight Aims and Conditions' do not very well fit to Table 1 of Marenco et al. Furthermore, you write 'b923', and Marenco et al. 'B923'. Maybe this could be more harmonized between both papers.

Table 4: Negative latitude values don't make sense here.

Table 4: Longitude and latitude values for b923/b924 do not fit to Fig. 1.

Table 6: Sometimes you write 'D_eff' and sometimes 'd_eff'.

Table 6: With 'derived RI' you mean the refractive index you iterated to fit the optical measurements? If yes, 'iterated RI' would be more specific.

Table 7: What means 'assuming internal mixing' here? In my understanding, the mixing state (internal/external mix) is only relevant for optical calculations but not for the derivation of the refractive index from filter samples.

Figure 4: What is 'SLRs'?

Figure 5a: The dV/dlogD value for b924 (green) at the largest three size bins is more than ten times higher than the corresponding average value shown in Fig. 2. However, there are only five flights and SAL flight lags. How do these figures relate? How did you calculate the average in Fig. 2? See also my comment on p15l28.

Figure 6: You write '6(a)' while there is no '6(b)'.

Figure 7: The dashed lines are not very well visualized and the description is missing in the legend (at a reference to p17l12 should be added).

Figure 9: 'Aspect ratios histograms as a function of number fraction of particles' is not clear. You mean 'number fraction of particles as function of aspect ratio'?

Figure 10c: It looks like there is a height dependence of the fraction of $D>5\mu m$ particles within the SAL. I think this height dependence should be briefly discussed in Section 3.3.

Figure 11: Relative 'n particles' for '10.0 to 40.0' shows no big difference between 'b920 R2' and 'b928 R2', so it is a bit unclear why you mention here 'B928 R2 (top right) contained giant mode MBL particles.' which should also be true for b920.

Figure 11: The last sentence could be 'Errorbars are counting uncertainties.'.

Figure 13: 'size-specific RI is used': Is this explained in the main text?

---

## Referee Comment (RC3) · Anonymous Referee #3 · 8 Sep 2018

The study characterizes the dust properties during the beginning of trans-Atlantic transport of dust particles. It presents new airborne measurements of dust size distribution, composition, shape, and optical properties within the Saharan Air Layer (SAL) and the Marine Boundary Layer (MBL) taken during the AERosol Properties – Dust (AER-D) fieldwork campaign in August, 2015. In their 6 flights, the authors used wing-mounted optical particle counters and shadow probes to measure dust sizes between 0.1 and 100 $\mu$m diameter, a nephelometer and an absorption photometer to measure dust optical properties, and an in-cabin filter collection system to collect dust samples.

[Figure]

The focus of the study is to highlight the presence and contribution of coarse and giant mode dust particles to the dust size distribution, mass loading, shape, composition, refractive indices and optical properties. The authors found that within the SAL, dust particles with diameter (D) greater than $20\mu$m are detected in 100% of the cases, and those with D>$40\mu$m are detected about 36% of the cases. Of the dust particles detected, 14% of the masses are for dust particles with size D<$2.5\mu$m, 60% for size D>$5\mu$m, and about 10% for D>$20\mu$m.

In addition, the authors also found the following: the shape of the measured particle size distribution does not vary significantly between dust layers; the modal aspect ratios are in between 1.2 to 1.4; the real part of dust refractive index in both SAL and MBL is within 1.47 to 1.49, but the imaginary part is between 0.0012 - 0.003i in the MBL and between 0.0004 - 0.0005i within the SAL. They also found that the single-scattering albedo (SSA) at 550nm decreases in the SAL when the measured coarse and giant dust particles are included in the calculation. However, they concluded that the variability of the SSA is not controlled by the dust size distribution, but by the variability in dust composition, contrary to previous studies.

Observational datasets for the coarse and giant dust particles, reported in this paper, are very important to better constrain dust properties in climate models. Current climate models over-estimate the fine-mode dust particles and under-estimate the coarse-mode particles, leading to uncertainties in the estimation of dust optical properties. This is largely due to inadequate observational constrains, and only few similar measurements of size-resolved dust properties are publicly available, with few obtained during the summer time period. Hence, high-quality measurements with a wider particle size range, like those reported in this study, are needed.

The paper is generally well written, and I believe it also meets the ACP standards. I recommend it for publication, if the authors can address the following comments:

1. Reading through the paper, some parts of it are rather confusing. This is primarily

because some of the sentences are too long, making the reading of the paper a bit tiring. The long sentences also sometimes obscure the point the author may want to pass across. I encourage the author to look more closely into each sentence, separating the long ones to multiple short sentences, where necessary. While few of these sentences are highlighted below, I cannot point to all the instances and I hope the author will do the due diligence in addressing this comment throughout the paper.

Pg 14 Lines 6-8, 14-16. Pg 15 Line 1-4. Pg 17 Line 1-4, 10-12. Pg 18 Line 22-26. Pg 20 Line 2-5. Pg 25 lines 16-19

2. Pg 6: The authors should provide a more objective assessment of the dust source areas. While HYSPLIT back-trajectory understandably are associated with uncertainty at the trajectory endpoints, it is still a reasonable method to determine the age of the dust particles, especially when the alternative is subjective. This is particularly useful for the dust particles in the SAL, where such trajectory can easily be estimated along a constant potential temperature surface, therefore avoiding possible influence of the convective events within the boundary layer. Doing it this way, may give a more close and objective approximation of the dust age, to which the SEVIRI images can eventually confirm. Free-tropospheric dust aerosols generally preserve their temperature for a considerable distance from the source region. Isentropic trajectories are therefore suitable above the boundary layer (e.g. Merrill et al., 1986).

From the HYSPLIT website (https://ready.arl.noaa.gov/HYSPLIT_traj.php), the figure below shows an example of the isentropic back-trajectory for flight #b932 starting on 20/Aug/2015 at 12Z for an arbitrary height of 2800 m above sea level. This height corresponds approximately to the highest extinction in your Fig. 4. The figure is a 3-day back-trajectory and it appears to suggest that the starting point after 3 days is approximately in the same area as suggested by SEVIRI in you Figure 1. This calculation can be repeated for different height within the SAL, and can also be combined with the SEVIRI images to give a more objective estimate of the dust sources, the age and the starting location.

[Figure]

In addition, the figure below uses the NCEP reanalysis dataset. It may be useful, however, to use a better quality meteorological dataset, like ERA-Interim with relatively higher resolution, to drive the HYSPLIT back-trajectories. ECMWF assimilates meteorological data from radiosondes that launch from few but important stations over north Africa. This may reduce the uncertainty even further, giving some more credence to the methodology.

3. The authors should either carefully justify the application of the Lorenz-Mie theory for dust particles larger than $\sim20\mu$m or use a more appropriate methodology for this size range. The manufacturer-provided size bin diameters were calibrated against polystyrene latex spheres, which the authors corrected to diameter of dust using Lorenz-Mie method (on PCASP and CDP). But Lorenz-Mie theory is only valid when the particle size is comparable to the wavelength (Bohren and Huffman, 1983). For coarse and giant dust particles with diameter larger than $\sim20$ $\mu$m, the application of Lorenz-Mie theory is no longer valid, and instead the geometric optics method may be useful (see Bi et al., 2009).

Specific Comments:

Pg 5, Line 7. Pg 7, Line 20. Pg 14, Line 4. Pg 16, line 25. Pg 18, line 2. Pg 21, line 15: The table numbers referenced here are wrong. Please check all other reference in the paper.

Pg 3, Line 9-10: Re-write for clarity.

Pg 9, Line 8-9: I wonder if this difference between the "all-in" and the "center-in" is actually quantified. This text referenced here appear to be an assumption as suggested by the use of word "considered". If the latter is the case, I suggest this sentence should be re-written to clarify this point.

Pg 16, Line 24: There is no need for "6a", there is just one figure. Please also correct this in other places of the manuscript.

Pg 18 line 14: Figure 8c is not provided.

There is no definition of some acronyms – an example is the "SLR" acronym in the text or in Fig. 4. I suggest the author look through the paper and make sure every acronym is defined before use.

References:

Bi, L., Yang, P., Kattawar, G. W. and Kahn, R.: Single-scattering properties of triaxial ellipsoidal particles for a size parameter range from the Rayleigh to geometric-optics regimes., Appl. Opt., 48(1), 114–126, doi:10.1364/AO.48.000114, 2009.

Bohren, C. F. and Huffman, D. R.: Absorption and scattering of light by small particles, 1st ed., Wiley-VCH., 1983.

Merrill JT, Bleck R, Boudra D. 1986. Techniques of Lagrangian trajectory analysis in isentropic coordinates. Mon. Weather Rev. 114: 571–581.

[Figure]

**Fig. 1.**

---

## Referee Comment (RC4) · Anonymous Referee #4 · 8 Sep 2018

This manuscript presented a study of Saharan dust based on airborne observations made over the Eastern Tropical Atlantic near the western African coast. The measurements were targeted to characterize dust microphysical, chemical and optical properties, including size distribution, particle shape, mass loading, composition, refractive indices, and SSA. This study contrasted the dust properties in Saharan Air Layer and the marine boundary layer. The authors highlighted several important findings which will advance the current understanding and benefit later modeling studies. The manuscript is logically organized and well written. It is noted that the authors provided meticulous

details about the instrument, data reduction, and uncertainty analysis. This reviewer believes that this manuscript shall be published after the authors considering a few suggested minor changes, which will not alter the major finds of this study.

Minor Comments:

Page 4, Line 4: Please clarify that while light shadowing techniques are not impacted aerosol composition or Mie theory conversion issues, they still can be impacted by non-spherical particles.

Page 5, Line 7: Change "Table 1" to "Table 2"

Page 5, Line 8, please describe the AOD and clarify if the AOD is calculated over the dust layers. Please also make changes to the table caption so that it will be consistent with the text

Page 5, Line 26: Change "Figure 1b" to "Figure 1"

Page 6, Lines 18-19: The authors note that visually identifying and tracking dust plumes is subjective, difficult, and potentially error-prone. Would it be possible to instead obtain the underlying satellite data and apply an objective threshold?

Page 7, Line 5, please add a brief discussion on the choice using PSAP correction by Turnbull (2010) and difference between this correction and that by Virkkula, AS&T, 44:706-712, 2010

Page 11, line 31, please provide a more quantitative criteria to define the word "dominant"

Page 13, Line 1: Figure 8b is unrelated to SSAs; perhaps Figure 13b was intended?

Page 13, Line 8: Please restate the rationale to hold the real part of the refractive index at 1.53, in the context that in Section 3.5, the real part is found to be 1.47-1.49 based on the filter sample composition.

Page 13, Lines 14-20: This information might be better suited to a table, which could also include the actual refractive index used for each substance.

Page 14, Line 4: Change "Table 3" to "Table 4"

Page 16, line 25. Please provide a brief discussion on how the "best-fit" compare to observed volume size distribution and number size distribution and Change "Table 4" to "Table 5"

Page 18, Line 2: Change "Table 5" to "Table 6"

Page 18, Lines 8 and 11: These two statements regarding a potential decrease in dmax with height seem contradictory. Please clarify to make them consistent.

Page 18, Line 14: There is no Figure 8c

Page 19, Line 14: Please supply a reference for the dust density value.

Page 21, Line 15: Change "Table 6" to "Table 7"

Page 23, Line 26 (and Figure 13b): Some readers may wonder if finding a good agreement between the imaginary part of the refractive index and the SSA is expected, given the relationship between k, absorption, and SSA. The authors should consider the significance of confirming that the relationship exists in this case.

Page 24, Line 20: Change "Table 5" to "Table 6"

Figure 4: This figure suggests the flight b924 and b934 did not have extensive sampling in MBL, please make changes in text accordingly

Figure 6: The blue shading was very faint on my screen. Perhaps a darker shade, or even hatching, could be used instead. Also, as there are no other parts to this figure, "Figure 6(a)" should be changed to "Figure 6" (as well as in the associated text).

---

## Author Comment (AC1) · 9 Nov 2018

**Response to Reviewers**

**K. Kandler (Referee #1)**

The manuscript describes airborne measurements of mineral dust size distributions close to the source. Representation of aerosol size distributions inside atmospheric transport models is still based on many assumptions, and new data for evaluations and input is highly welcome. The manuscript adds new data for the region close to the source at the beginning of the trans-oceanic transport. It also adds additional evidence to the importance of studying particles larger than 10 m, which are frequently omitted in aerosol research, and which are difficult to study. Furthermore, optical properties for the dust aerosol are reported, which can also serve the community, e.g., as model input or for remote sensing interpretation. The authors show that variation in physical as well as compositional parameters can have a considerable impact on the optical properties, and that in their samples, the compositional variation is dominating the variation in single scattering albedo.

The topic is suitable for ACP. The paper is well-written and clearly structured. References are mostly made where appropriate, some additional comparison with previous work would enhance the general significance. Some issues - particular on the error handling - should be addressed and questions answered before publication.

We would like to thank Konrad Kandler for this detailed review and comments which we hope have led to an improved, clearer paper. We have added information regarding the different sizing metrics used, additional explanations of error handling, and expanded on comparisons to previous work, all of which are detailed below. We apologize for incorrect cross-referencing of tables and figures, which have all now been corrected.

========================================

Remarks and questions

P3 L32: Are really any inlet size restriction removed? Given the high speeds at which particles are collected, a minimal deviation from perfect alignment might easily introduce boundary layers and re-circulations.

It is true that variations in the airspeed and flow angle through the sample area may occur due to distortions of the airflow about the wing probe housing and arms (Korolev et al. (2013), Weigel et al. (2016), McFarquhar et al. (2017)). We do not attempt to correct for these factors, and now state so in Section 2.3.2, and mention them in the introduction.

In terms of the alignment, the canisters are set at a set vertical angle to accommodate the 'normal' angle of attack (AoA) of the aircraft. There will be up to several degrees of misalignment as the AoA changes with altitude. The average and standard deviation of the canister angles is 3.25±0.14 deg (negative is nose-down). The range of AoA is 3.5-6 degrees nose up so there will generally be a couple of degrees misalignment between the probes and the airflow assuming no aircraft perturbation of the flows. Korolev et al. (2013) show that the velocity between the arms of the CIP has an unperturbed region of approximately half the arm width. This suggests that a small misalignment (only several degrees) is unlikely to narrow this unperturbed region sufficiently to impact the size-dependent concentrations.

Measurements behind inlets will systematically omit a portion at the upper size range of the PSD, and while wing probes may be subject to various uncertainties relating to local flow distortion, they are at least able to measure the full size range of dust particles because no inlet is used. Additionally, the bigger the particle the less they are effected by flow distortion because of their larger momentum. This is the opposite of the pipe bend problem which imposes a cut size.

P6 L32: Should this diameter then considered as optical equivalent diameter? Please state.

Not in this case, since various 'types' of diameter are considered throughout the paper. However, we have added, "Hence particle sizes determined from the PCASP and CDP represent optically equivalent diameters," to section 2.3.2 where this is discussed.

P7 L10: Have experimental or theoretical approaches been used to characterize the losses?

Both – Trembath (2012) performed OPC measurements behind different inlets on the aircraft, and followed these up by theoretical calculations of pipe losses. We added "experimental and theoretical" to this sentence.

P8 L6 and Figure 2: It looks like the sizing uncertainties from the number size distribution were not propagated into the volume size distribution (e.g., the CDP point at 20µm seems to have a factor of 3 uncertainty in size, which should propagate to around a factor of 10 in volume, but a much smaller error bar is shown in the volume plot). Instead it looks like the same vertical error bars are shown in both plots. Please explain or modify.

The error processing takes place in two stages, which may be easier to follow separately. 1) Uncertainties are calculated for each SLR. 2) Uncertainties for the campaign average PSD are calculated, which are shown in Figure 2. We explain each of these steps below.

Additionally, the horizontal error bars on Figure 5 represent the minimum and maximum bin edges possible, using the bin centre and bin width uncertainties. Hence the actual uncertainty on the diameter centre point (which would be used in a typical progression of uncertainties from number concentration to volume concentration as stated by the reviewer), is much smaller than that shown on the plot. We show error bars to incorporate both bin centre and bin width in order to provide a realistic reflection of the real uncertainty in the sizing.

1) For each SLR, dN/dlogD uncertainties (vertical error bars) already includes bin width uncertainty *and* bin centre uncertainty, since they both contribute to the calculation of dlogD. dV/dlogD additionally takes account of the bin centre uncertainty again through the diameter cubed, as well as incorporating the uncertainty in dN/dlogD. Below is an example from one SLR, b932 R2, in the SAL.

In the figure below, we take the reviewer's data point close to 20 µm (diameter 18.7 µm). Here the uncertainties are as follows:

Bin centre uncertainty = 0.74 µm

Bin width uncertainty = 0.71 µm

Fractional uncertainty in dN/dlogD=0.28

For the uncertainty in dN/dlogD, the uncertainty from the bin width is 4 times larger than that from the uncertainty in the measured number concentration, so that the bin width error dominates the total error.

Fractional uncertainty in dV/dlogD=0.29

For the uncertainty in dV/dlogd, the uncertainty from the bin centre is around half that of the uncertainty propagated from dN/dlogD (which also already incorporates both bin width uncertainty and bin centre uncertainty). As a result, the fractional uncertainties in dN/dlogD and dV/dlogD are not very different in the two plots below, since they are both dominated by uncertainties in the bin size.

In contrast, for the CDP data point centred at 3.9 µm, the fractional uncertainties in bin centre and bin width are much larger (0.12, 0.05), which results in the much larger vertical error bar in both dN/dlogD and dV/dlogD.

[Figure]

Figure: dN/dlogD (left) and dV/dlogD (right) for an individual SLR (b932 R3) in the SAL.

2) Secondly, uncertainties are combined to provide the SAL and MBL average PSD, e.g. as shown in Figure 2.

In order to do this, the random and systematic errors from each SLR are processed separately, as recommended by Baumgardner et al. (2017). This is because the systematic error (from sample area uncertainty and bin size uncertainty) does not reduce by increasing the sample size, while the random errors (from counting and discretization uncertainties) can be reduced by increasing the sample size. This is done by standard analytical error propagation. It turns out that for the CDP, beneath 20 µm diameter, the total uncertainties are dominated by the bin size uncertainties, while above this size the counting error becomes significant as well.

We have added information to the caption for Figure 2, "Horizontal error bars represent maximum bin width due to uncertainties in both bin centre and bin width." We have also added the following text to Section 2.3.2:

"For the CDP at d < 20 µm bin size uncertainty was found to dominate the total uncertainty in dN/dlogD and dV/dlogD. Horizontal error bars in Figure 2 represent the maximum uncertainty in bin edges, derived from uncertainties in both bin centre and bin width. Uncertainties in bin size contribute to uncertainties in both dN/dlogD and dV/dlogD, and therefore the relative uncertainties do not change significantly between the two panels."

Figure 2 is hard to understand also from another aspect: Either the measurement points are all at the lower boundary of the error, or the error always includes zero. If they are at the lower boundary, please explain why. In particular with a non-zero counting error (which is necessarily two-sided), this seems impossible. If the error always includes zero, then the statement in P9 L27, "agree within the error bars" is meaningless, as all data could be zero.

In Figure 2 (and also Figure 5), only upper error bounds are shown, since for several points the lower error bars exceed the minimum on the log scale axis and impede visibility of the data points. This is now stated in the caption for both figures.

P8 L31 and P9 L4: if a particle was not counted for the given reasons – how was this underestimation of the concentration then accounted for? Decreasing proportionally the sample volume?

Yes, it is accounted for. The effective array width (EAW) changes depending on whether the all-in or centre-in approach is used. The EAW is used in calculation of the sample area (McFarquhar et al., 2017), and therefore the number concentration and its uncertainties. We added, "The sample area is adjusted for the effective array width, which is different depending on whether 'all-in' or 'centre-in' is used (McFarquhar et al., 2017), and therefore the calculated number concentrations account for this."

P9 L6: For what reason the projected area equivalent diameter was not used, which appears to be a common standard in shadowing techniques (i.e. light microscopy)?

With hindsight, and ample data processing time and computation time, it would have been interesting to additionally process the OAP data using projected area equivalent diameter. However, a limitation of using this metric with the OAPs is that particles can sometimes appear hollow (e.g. due to being out of focus). Under these circumstances using projected area equivalent diameter would be a significant underestimate, and therefore it is not clear that this is the 'best' metric for the OAPs. We have added the words, "Area-equivalent diameters were not calculated because particles can sometimes appear hollow on the OAPs (McFarquhar et al., 2017) which would lead to undersizing."

P9 L11-21: Please be more explicit on the error handling. Was is done by analytical error propagation, by Monte Carlo simulation, ...? What were the distribution assumptions(Poisson?)? How was the counting error treated, if measurement points with only 4 particles were regarded? If error bars are given: do they show a certain confidence interval, one (or more) standard deviations, or maximum errors?

Error handling was done by standard analytical error propagation. Counting error was calculated as $n^{1/2}$, where n is number of particles counted in the SLR. Size bins containing less than 4 particles in the whole SLR were disregarded, and therefore counting error was not calculated for these points. Error bars show maximum errors. This paragraph now reads:

"Uncertainties in number concentration for all size probes are propagated from 1Hz measurements, through to means over SLRs, through to the AER-D campaign averages. For all probes, random errors (due to counting and discretization error) and systematic errors (due to sample area uncertainty and bin size centre and width from Mie singularities) were accounted for in their contribution to total number concentration errors, and propagated by standard analytical error propagation. I.e. random error can be minimized by increasing the sample size (averaging across the campaign), while systematic error remains constant. For the CDP at d < 20 µm bin size uncertainty was found to dominate the total uncertainty in dN/dlogD and dV/dlogD. Horizontal error bars in Figure 2 represent the maximum uncertainty in bin edges, derived from uncertainties in both bin centre and bin width. Uncertainties in bin size contribute to uncertainties in both dN/dlogD and dV/dlogD, and therefore the relative uncertainties do not change significantly between the two panels. All error bars represent maximum uncertainty."

P9 L30-35: Are there any arguments beyond consistency with previous measurements for disposing (if I understood correctly) the CIP15 CC data? It is stated that the 2DS XY metric is better – better in terms of what? Please explain.

Yes, as stated on p9 L29, the "the CC metric will oversize a non-spherical particle," since the circumscribing circle, by definition, does not take any account of a particles' non-sphericity. We refer specifically to the metric, and not the instrument here.

P10 L1: This type of effective diameter seems usually (Hinds 1999) to be called the Sauter mean diameter (apparently defined by Sauter 1926) or mean volume-surface diameter (Hinds 1999). As different effective diameters exist, I suggest referring to one of the mentioned denominations and referring in addition to a more fundamental paper (McFarquhar et al. 1998).

We have changed the effective diameter reference to Hansen and Travis (1974), and added the Hinds (1999) reference as well as adding 'volume-surface diameter.'

P10 L4-7: Given the high counting uncertainty associated with a single count of the maximum-sized particle, wouldn't there be a better option, e.g., fitting a log-normal distribution or power law into the data? At least, both mentioned approaches should be compared to allow for consistent comparison.

Since we disregard SLRs where fewer than 4 particles were detected, our counting uncertainty will be a maximum of 50% if 4 particles are detected, and reduce accordingly when more are detected. Actually, only 2 SLRs detected fewer than 6 particles in the maximum size, so the errors are substantially less. Fitting a lognormal or a power law to the data is not necessarily useful as the size distribution shape will not necessarily follow these fits.

P11 L12 (and below): As magnifications are relative to the used screen, a measure like nm per pixel or pixels per smallest particle would be more meaningful.

Values are: x2,000: 55.9 nm/pixel, x10,000: 11.0 nm/pixel. These have been added to the manuscript.

P11 L14-19: Wouldn't it make more sense using the same metrics as described for the shadowing instruments above?

In an ideal world, yes. The OAP software processing is computationally intensive, and ellipse fitting to the time-of-flight data is not currently an option. Since more realistic metrics could be derived from the filter samples, we chose to use these (rather than a less appropriate metric consistent with the OAP data), hoping to inform knowledge on dust properties for the dust aerosol community. In the future, with the benefit of hindsight, processing the OAP data differently would be considered. See also response to P9 L6 above.

We tested the impact of processing the filters PSDs differently, mimicking the XY and CC metrics as much as possible. However it should be noted that even when processing the filters data like this, the two are still not equivalent. This is because the 2DS and CIP15 data are 2-D projections of a 3-D particle orientations in the atmosphere. While the filters data are also 2-D projections, the particles likely fall with their largest axis parallel to the filter sample. Despite this, using the XY metric for the filters data did not produce any noticeable changes in the PSD. Contrastingly, the CC metric processing produced PSDs with a mode shifted towards larger particles, and often detected a maximum particle size one size bin greater than that shown in Figure 7.  These effects on the PSD mimic the differences seen between the 2DS CC and 2DS XY in Figure 2.

We have added a sentence to Section 2.4 to indicate this:

"Additionally, we tested the sensitivity of the filters PSD to using the mean XY and CC sizing methods applied to the OAP data (not shown). Using a mean XY method on the filters data did not produce significantly different results, while using the CC method was found to shift the PSD towards larger particles, similar to the findings from the OAP size metric comparisons."

P11 L25-29: Please state where you draw the lines between thenardite/sulfate and gypsum/sulfate.

No quantitative separation is used since the percentages vary depending on the contribution from the other oxides. However, typically when Thenardite (Gypsum) is classified, percentages of $Na_2O$ and $SO_3$ (CaO and $SO_3$) are within 10% of each other. We have added information about the classification of Thenardite to section 2.4 which was missing, and added information on the oxide percent.

P16 L18-20: This behavior was described and a similar explanation was given for the same geographical region and season a while ago (Jaenicke et al. 1978), and also some model approaches seem to predict that (Garrett et al. 2003), so maybe a short comparison with previous finding would make sense here and enhance the general significance of the findings.

Thank you for these interesting references. We have added to this section the following, "This has also been suggested by Jaenicke and Schutz (1978) from aerosol surface observations at Sal, Cape Verde, where giant particles (d > 40 µm) were observed to arrive at the site a day after that of coarse dust particles (6 < d < 60 µm)." Garrett et al. (2003) appears to be more relevant to the introduction, since that work found that concentrations of d>10um particles were under predicted by models for transport across the Atlantic, so we have added that there.

P16 L28-30: This finding is surprisingly similar to the volume distributions shown for Cape Verde in winter time, comparing dusty and marine situations (Kandler et al.2011), so a comparison could show a broader relevance.

Thank you for this reference. Actually this finding is a little different from those of Kandler et al. (2011). In Kandler et al. (2011), for the dust case mean, the volume distribution peaks sharply at around 10 microns, whereas for our 'giant dust in the MBL' cases the volume distribution peaks at ~5 µm and ~30 µm. For the 'maritime' cases the PSDs shapes appear quite different too.

We have added a paragraph to Section 3.2 comparing the PSDs to those in the literature. We are also about to submit a follow-up paper which includes a thorough comparison of the AER-D PSDs to those in the literature, drawing together the latest results.

P17 L14-18: This would mean that the flow through the filter would have been ten times as high as the one used for calculating the size distributions – is this still in a physical probable range? Can there exist any aerosol concentration effects due to high velocity gradients during sampling?

Firstly, yes, this is plausible. There is nearly a factor of 10 difference in flow rate measured between different SLRs. Therefore it is possible that problems here could have strongly impacted the retrieved size distributions.

Secondly, non-isokinetic sampling can artificially increase number concentrations in certain size ranges. E.g. if the aspiration speed is larger than the aircraft speed then the submicron fraction will be artificially enhanced, and vice-versa. However, this is not consistent with the filters PSD being offset across all size ranges compared to the wing probes.

P17 L22: Smoothness of the curves could be related to different size intervals. The wing probes seem to have more size intervals (with higher counting uncertainties) in Figure 7.

This is true. We have added, "in part due to the broader size bins used" to this sentence.

P18 L23-32: Probably a median aspect ratio, instead of a modal value, would make interpretation less dependent on a single interval. In particular, as in the following the values are compared with median values from other sources, which otherwise can't be compared.

Thank you for this suggestion. We have calculated median values of the aspect ratios and added them to the panels in Figure 9. We have incorporated them into the discussion in this paragraph and re-written it. Although calculation of the median reveals higher values than the modal values, our values still appear a little lower than those from SAMUM1, 2 and AMMA, ranging from 1.16 to 1.54 for the full PSD.

P20 L24-25: How about sea-salt reacted with sulfuric acid?

Yes, this has been added.

P21 L7-12: How would black carbon have been identified in this work? Were all particle images manually inspected for fractal-like structures (doesn't come clear from the method section).

Yes, all images were manually inspected during the scanning procedure. Out of 6500 particles analysed, only one single black carbon particle was observed, in a chain, or fractal-like structure.

We have adjusted this text to reflect this more accurately, which now reads, "Additionally, in contrast to Liu et al. (2018), we do not detect any black carbon on the filter samples **in significant quantities**, which they find present predominantly between sizes of 0.1 to 0.6 μm. **During the analysis of 6500 particles, only one black carbon chain structure was observed.**" We also added the following sentence to the methodology section 2.4 for clarification, "Only one black carbon chain-like structure was observed during the analysis of over 6500 particles, and therefore this aerosol category is not included."

P22 L20: In particular for the internal mixing, there is plenty of room with respect to complexity for calculating an effective value (Nousiainen 2009; Lindqvist et al. 2014).

These citations have been mentioned in this sentence now.

P24 L6: In addition, also measured refractive indices show the dependency on iron (Moosmüller et al. 2012; Caponi et al. 2017), so it appears to be consistent in general.

We have added these citations to this section.

Figure S2: Where does the 'step function' of the imaginary part at 700 nm wavelength derived from?

This 'step function' at 700 nm originates from the refractive index dataset of hematite, which drops sharply from ~0.2 to ~0.0013 in the imaginary part at this wavelength.

======================================

Corrections

General: The order of the figures does not correspond with the order of their references in the text.

General: The table numbers in the text don't fit with the table numbers at the end.

Apologies for these errors, these have now all been corrected.

P3 L8-11: including into calculation/model?

This was calculated – the wording has been changed.

P4 L22: Why time-of-flight? Wasn't it an SP2 instrument?

We have changed the wording to 'real time measurements,' which is more accurate. (Yes, it was an SP2).

P4 23-25: While the link to the optical property measurement is clear, the reference to the ice nucleation measurements and to the modeling work doesn't seem to add something useful here.

We believe it is useful to tie together the papers relating to the same flights since the impacts of dust in the atmosphere are multiple. However, we have deleted the reference to the paper in preparation since it is not yet submitted.

P4 26-30: I suggest removing the paragraph, as the structure is standard.

Done

P5 L15-22: If results from SAVEX and CATS are not discussed here, these explanations should be removed.

These flights and results are discussed here. Certain flights covered multiple objectives. We prefer to retain this information since it explains the choice of location for each of the flights presented in this article and may be used in subsequent publications.

P8 L2: "ambiguities" instead of "singularities"?

Changed.

P9 L9: What does "Instead, ... sizing metric" refer to? Aren't the previous sentences the section about the sizing metric?

Here we intended to refer to the sizing metric of XY versus CC, pointing out that the choice of XY vs CC is the main controller of PSD in this case, as opposed to the choice of an 'all-in' versus a 'centre-in' approach. We have clarified this by adding, '(i.e. XY versus CC)' to this sentence.

P10 L8-9: "do not have a minimum detection concentration level, but at low particle concentrations the sampling statistics simply become poor": In fact, this is the detection limit. With poor sampling statistics the (counting) error becomes high in comparison with the signal. If it is decided to omit data below a certain signal/error ratio, the detection limit is introduced (which is a common procedure).

This is actually what we intended to say. To clarify, we have added to this sentence, ", introducing an effective detection limit."

P10 L14: "PSD' size distribution" – doubling "size distribution"

Removed

P12 L10: "low SEM signal" low image contrast?

By 'low signal' here we refer to the energy, or number of photons emitted, not the image contrast. Since this is dependent on particle volume, for smaller particles the number of photons emitted is lower, or non-existent for very small particles, and the analysis and interpretation becomes difficult. We have added, "(fewer photons emitted for smaller volume particles)" to this sentence.

P15 L5: increased?

Changed

P16 L24: There seems to be no (b) in Figure 6.

Changed

P18 L5: In Figure 7, in most cases the volume maximum is below 10 m, in one above (filter size distribution). "Dominating" therefore doesn't seem to be appropriate for the largest particles with respect to the mass.

This has been changed to, "can contribute substantially to"

P18 L12: Sentence ends with a comma.

Changed.

P24 L28: Most of section 4 is rather a summary than conclusions, so the section should be termed accordingly.

We have followed the guidelines from ACP at https://www.atmospheric-chemistry-and-physics.net/for_authors/manuscript_preparation.html for manuscript composition, which indicates that 'Conclusion' should be used. Also in our final paragraph we make some broader recommendations and comments.

Table 1: Different reference formats. Maybe explain instrument abbreviations, e.g., as addition in Table 3?

The reference formats have been corrected. We have created an appendix containing the instrument acronyms, and added this to the caption of Table 1.

Table 3: Check caption

This has been corrected.

Table 4: different time formats between table 2 and 4 (?)

Table 4 has been changed to comply with the given ACP date and time terminology guidelines.

Table 5: 'IT' number format (E+01 etc.)

These have been changed to the $1.16 \times 10^1$ format.

Figure 2: Both y axes should have the same number format.

The axis for the right hand panel has been changed to be consistent.

---

## Author Comment (AC2) · 9 Nov 2018

**Response to Reviewers**

**Anonymous Referee #2**

Ryder et al. discuss airborne measurements of Saharan dust performed between Cabo Verde and the Canary Islands in August 2015 with focus on several aspects of the dust, for example the coarse mode size distribution. I think the paper is interesting and fits well into the scope of ACP. Overall, the paper is in a useful shape but there are many, mainly minor, things the authors should improve before the paper is ready for final publication.

We thank reviewer 2 for their comments and are pleased that they find the paper interesting. We have responded to the minor comments below and corrected or altered them in the manuscript. In particular the question about why we did not process the CIP15 data with the XY metric has led to additional reprocessing of some of this data, which has been informative to the manuscript. Further details are given below. We apologize for incorrect cross-referencing of tables and figures, which have all now been corrected. We have added clarification to our flight patterns section to better explain the data.

General comments:

Abstract: Maybe one sentence with results for the MBL could be added.

We have added, "Within the MBL, mean effective diameter ($d_{eff}$) and volume median diameter (VMD) were 4.6 μm and 6.0 μm respectively, giant particles with a mode at 20-30 μm were observed, and composition was dominated by quartz and alumino-silicates at d > 1 μm."

On p3l6 you define coarse and giant mode dust. However you do not follow this definition, e.g. at p7l15, p9l32, p12l27, p25l25. Please make sure that the paper is self consistent.

We have changed the introduction to reflect the references throughout the rest of the text. It now reads, "Coarse and giant mode dust (defined here as d>2.5 μm and d>20 μm, respectively)…"

Often the citation type, e.g. '(Ryder et al., 2013)' vs. 'Ryder et al. (2013)', is not correct.

These have been corrected throughout the manuscript.

You did not define the flight legs. At p11l5 you mention 'R2' for the first time, but you did not introduce this properly. In Fig. 4 an undefined 'P2' etc appears, probably also refering to flight legs. I suggest to add a proper introduction of the nomenclature and to add a table (maybe in the supplement) with more details about the flight legs considered in the paper (e.g. name of leg, start time, end time, duration, height, etc. ).

We refer the reviewer to Section 2.1, 'Flight Patterns,' where in the final paragraph profiles and straight-and-level flight legs are explained and defined. We have added an explanation of R and P which refer to straight and level runs (SLRs) and profiles respectively, as suggested by the reviewer. We have also changed the terminology throughout the paper to use SLR rather than 'flight leg,' since SLR relates more closely to the 'R' abbreviation and is also the abbreviation used by the UK research aircraft community. Full time and altitude information for profiles and SLRs can be accessed online. We have added, "Full information about profile and SLR times and altitude are available from the Centre for Environmental Data Analysis (see Data Availability)."

You write SSA in the text but use omega_0^550 in the figures. Please use only one of these.

We prefer to use or $\omega_0$ for the figures, so have added this to the first appearance of SSA in the introduction, "decreasing the single scattering albedo (SSA or $\omega_0$)."

Sometimes for the imaginary part values a 'i' was added after the value, sometimes not. This should be made consistent, perferably removing 'i' everywhere because the 'imaginary part' is a real value; see also your definition at p8l18.

The 'i's' have been removed from the text.

Often, the main text refers to the wrong table number.

We apologize for this and have changed the cross-references.

Specific comments:

p1l24: In my view 'during' should be replaced by 'at'.

Changed

p1l28: 'constituting up to 40% of dust mass': As you mention this number in the abstract, it should also be mentioned in Sect. 3.3.

We have added to Section 3.3, "In the extreme, up to 90% of dust mass can be found at sizes greater than 5 µm and up to 40% at sizes greater than 20 µm."

p2l3f.: It is unclear what 'this complex evolution' refers to. Suggestion: '... to capture correctly both the dust composition and the size distribution including their changes during transport in order ...'

We have rephrased this sentence.

p3l27: 'preceeding the AER-D flights' comes a bit surprisingly. Please reformulate without refering to AER-D which is introduced only later.

We have removed the mention of AER-D and rephrased the sentence.

p3l33: I think 'however' could be removed here.

Done

p4l1: 'Mie theory conversion': it is not very clear what is meant. Please reformulate.

This has been changed to, "the scattering cross-section to particle size relationship is non-monotonic."

p5l7f.: 'The dust events sampled 550 nm AODs from 0.4 to 0.8' should be reformulated.

This now reads, "The dust events revealed AODs at 550 nm from…"

p5l19: 'aerosol structure' is a bit unclear. You probably mean the vertical distribution (structure) of the aerosol.

Correct, we have changed this

p5l20: 'nearer the ...' could be replaced by 'closer to the'.

Done

p5l26: 'Figure 1b' does not exist.

Changed to Figure 1

p5l28: 'flight' could be added before b920 and b924.

Done

p7l3: The wavelength list may be a bit confusing. Maybe you can just write the wavelengths in parentheses after the instruments?

This now reads, "Scattering measurements were made by a TSI 3563 integrating nephelometer (at wavelengths of 450, 550 and 700 nm). Absorption measurements were made by a Radiance Research Particle Soot Absorption Photometer (PSAP) at 567 nm."

p7l25: 'aerosol' should be replaced by 'particle' to make the sentence more general. Particles with 6.2mm are usually not aerosol particles but much more likely some kind of hydrometeors.

Done

p8l18: PCASP and CDP do not operate at 550nm. This should be mentioned here including your assumption that the refractive index does not change between 550 nm and the instrument's wavelengths.

We have altered this paragraph which now reads, "Bin sizes also depend on the choice of refractive index applied. In this work, a complex refractive index ($n^{550} = m^{550} - ik^{550}$) of 1.53-0.001i was used to determine the PCASP and CDP bin sizes, as determined from Section 2.5 for 550 nm. Since the PCASP and CDP operate at wavelengths of 633 and 658 nm, we assume a constant refractive index across these wavelengths. This is supported by the relatively flat spectral refractive index shape at these wavelengths indicated in Figure S2."

p8l28: You could write ' ... in two different ways, resulting in different sizing metrics.' This would help the readers in the subsequent paragraph.

This has been changed to, "Thus to investigate some of these uncertainties, the 2DS data was processed in two different ways, using two different sizing metrics."

p9l3f.: Were the particles rotated such that one dimension is minimized and the other maximized? Or were x and y measured for each imaged particle without such rotation?

No, the particles are not rotated. We added the sentence, "The x and y dimensions are measured along the probe array, i.e. the particle is not rotated to minimize or maximize either dimension."

p9l6: '... though diameters will be lower than an area-equivalent diameter for example, if the particle is an ellipse.' looks wrong. For example, assume x=1 and y=2. Then D_XY=(x+y)/2=1.5. The area-equivalent diameter however is D_area=(1*2)^0.5=1.414 which is smaller than 1.5. This could be a reason why also the mean XY method somewhat overestimates the 'real' particle size.

This was a typo, 'lower' should have been 'larger.' We have added and changed this paragraph as follows: "The mean XY method is considered to give a more representative diameter for non-spherical particles than the CC metric. If the particle image is an ellipse, the mean XY diameter will be larger than an area-equivalent diameter, as used by the filter sample data. However, the OAP images capture 2-D image projections of the particles in their atmospheric orientation, while the filter samples are will be collected with their largest surface lying parallel to the filter sample, and therefore may be oversized in this context."

p9l8: I wonder if there is a reason why you don't use the mean XY method (instead of the CC method) for the CIP15 in this discussion paper?

This is a good point, and we would have done so initially in an ideal world. Unfortunately much of the OAP processing was initially done by different institutions for each OAP, and used institutional conventions, often selected for consistency between different fieldwork campaigns and optimized for ice/cloud particles, which allowed the whole of the ICE-D data to be processed with one assumption (over half the ICE-D flights were cloud flights).

However, we have now run additional processing on the CIP15 data using a centre-in, mean XY metric, although unfortunately it has not been possible to do this for all the AER-D flights analysed in this article. Two contrasting examples are shown below.

[Figure]

It is clear that the biggest difference in the OAP size range stems from the choice of size metric (XY vs CC), rather than the instrument (2DS vs CIP15). The difference is not noticeable when the giant mode is smaller (right hand panel). The difference in mean $d_{eff}$ from all SLRs analysed between the 2DS XY and the CIP15 XY is under 1%. Differences for the CIP15 CC are 4.5%, and for the 2DS XY under 1%. These uncertainties, and particularly that for the XY metric, are smaller than the 5% error already applied due to the uncertainty due to choice of refractive index for the OPCs.

However, the impact of OAP metric and instrument on $d_{max}$ is larger. Differences from the 2DS XY $d_{max}$ to CIP15 XY $d_{max}$ are +6%, to the CIP15 CC +37%, and to the 2DS CC +21%. The upper error of 6% (based on the instrumental differences using the more realistic XY metric) is now incorporated in the upper error bars on $d_{max}$ shown in Figure 8b. These are small relative to the existing error of 5μm when $d_{max}$ is around 20 μm, but comparable to the error when $d_{max}$ is larger, and can be seen for example in the now asymmetric error bar for the points at $d_{max}$ = 80 μm.

New Figure 8b:

[Figure]

The following text has been added to section 2.3.2, "Some CIP15 AER-D data were also processed using a centre-in, mean XY metric, but unfortunately it was not possible to process data for all the SLRs with this method. Therefore this data was used to inform on instrumental differences between the 2DS and CIP15 when processed with the same size metric (XY mean). It was found that the impact on the full PSD was very small

(d$_{eff}$ differed by under 1%), but that d$_{max}$ was up to 6% larger with the CIP15 XY compared to the 2DS XY. The upper uncertainty of 6% in d$_{max}$ was therefore propagated in combination with the other uncertainties in d$_{max}$."

p9l12: Which flight leg length do these 10ˆ-5 cmˆ-3 correspond to, approximately?

This number concentration corresponds to a flight leg length of 132 km or approximately 20 mins of flight time on the FAAM aircraft. This information has been included.

p9l20: It is unclear what '... errors due to bin size from ...' means. Please reformulate this sentence.

This sentence has been deleted.

p9l28f.: 'as expected when the particles are non-spherical (section 3.2)' could be replaced by 'as excepted for dust'.

This sentence has been re-worded.

p10l9f.: This looks quite similar to p9l10ff. Maybe you could bring both together or at least refer here to the previous text (e.g. 'Therefore, we remove, as mentioned, cases where fewer ...').

Done

p10l17: 'of around 0.2 to 1 m' should be replaced by 'smaller than 2-3m' when considering Fig S1.

Done

p11l5: 'R2', 'R5' are not defined.

These are now explained by the additions to the 'Flight Patterns' section.

p11l15: The areas of the ellipses were larger than the areas of the particle 2D projections because you used circumscribed ellipses. How large is this difference? Why didn't you use the area of the projection itself to determine the area-equivalent diameter?

We did not use image contrast and brightness levels with automatic processing since this would have meant that not all particles would be detected, since not all particles had similar contrast and brightness. This technique would have allowed measurement of projected area. Instead images were analysed manually by circumscribing an ellipse to the particles. This method was selected in order to maximise the number of particles analysed, and as it is operator independent, systematic and reproducible. However, the particles were not notably jagged at their edges.

Secondly, p11 L17-19, ("Note that this technique may oversize the particle volume, particularly where the shape is a platy silicate with a tendency to fall with its largest surface parallel to the substrate; e.g. Chou et al. (2008) found the height of dust particles examined under SEM to be around one third of their major axis length.") is an *additional* process which may lead to the filters PSD being oversized compared to the wing probes, which relates to the orientation in which plate-like particles fall flat on a filter substrate.

This paragraph now reads, "It was not possible to use automated image contrast to calculate projected particle area because of a high degree of variability in particle contrast. Our filters sizing this technique may oversize the particle size for two reasons. Firstly, the area of a fitted ellipse may be larger than a projected particle area, though the particles were not noticeably jagged around their edges. Secondly, our method may oversize particle volume where the shape is a platy silicate and has a tendency to fall with its largest surface parallel to the substrate. For example, Chou et al. (2008) found the height of dust particles examined under SEM to be around one third of their major axis length."

p12l29: Shouldn't '0.0001' be replaced by '0.0005'? Otherwise I don't understand this description.

This has been rewritten to better explain, and now reads, "a Mie scattering code is used to generate optical properties at 550 nm, using the ACC PSD, with refractive index of $m^{550}$=1.53 and $k^{550}$ incrementing in steps of 0.0005 from 0.0005 to 0.006, but with an additional smallest value of 0.0001 which was required for the MBL SLRs."

p13l1: 'Figure 8b' shows something else.

This sentence has been deleted. The correct figure is referred to in the next sentence (Figure 3).

p13l9: 'between 0.0015i to 0.0025i' doesn't fit to the mode value of 0.001.

Thank you for pointing this out, there was a typo and it now reads '0.0005 to 0.0025.'

p13l11: Why do you use 'volume fraction' here and 'number fraction' for the external mixing case (p13l30)? How big is the difference between both cases?

Volume fraction is appropriate for the internal mixing assumption since the volume of each mineral determines its contribution to the total refractive index. For the external calculations, our scattering code requires input in the form of number concentration. Internally, the code calculates the surface area, since this determines the aerosol scattering properties. If the internal mixing assumption is used, but with number fraction, the results are very different because the smallest sized particles dominate the number concentrations, and have a greater contribution from different compositions such as sulfate and salt (as shown in Figure 11).

p15l16: I suggest to briefly discuss the difference from Marenco et al., who find maximum AODs of 2.0.

We have added, "Note that lidar-derived AODs and extinction shown by Marenco et al. (2018) are slightly lower than those shown here, which may be due to different extinction properties of the dust at the lidar wavelength of 355 nm, and the Rosemount inlet enhancement effects shown in Figure S2, or the differences between a lidar curtain and sloped aircraft in-situ profile."

p15l28: 'each flight leg': As Fig. 5 shows only a single leg inside the SAL for each flight I assume that there was only one flight leg in the SAL for each flight? However, then Fig. 2 and 5 seem to not fit to each other as mentioned in a separate point below. In general, the legs should be described better.

This is not the case. Successive flights included 2, 1, 6, 2 and 3 legs in the SAL. Much of the data overlies itself for some parts of Figure 5b so that it might be inferred that fewer legs were performed. We do not intend for the reader to be able to distinguish each individual flight leg from panel a, and the points discussed in the text can seen. We have added information to the caption of Figure 5 to state how many SLRs are shown. With the addition of information to the 'Flight Patterns' section, the legs should be more understandable.

p15l33: I do not really understand this sentence. Would the absense or presence of the coarse mode not always have an effect on the overall shape of the size distribution?

Here we intended to explain that during AER-D, when dust concentrations were higher, the coarse and giant mode particle concentrations increased together, as well as the concentration of particles in the accumulation mode, which meant that the overall shape of the PSD remained the same. This contrasts to Fennec, where coarse and giant particles often increased in concentration but without the smaller particle concentration increasing. In that case, the shape of the size distribution *did* change. A few changes and additions have been made to this paragraph to state this more clearly, and also in the conclusion.

p16l1: 'peak volume concentration' is unclear. I suggest to write 'The peak of the volume distribution during …'.

Done

p16l8f: 'Figure 5b; green, orange and red'

Changed

p17l3: ' may be aligned horizontally in the atmosphere ... ': I suggest to add here a reference to Ulanowski et al. (2007), DOI:10.5194/acp-7-6161-2007, who made some simulations on this topic (see e.g. Fig. 9 of that paper).

Done

p18l10: How does 'Particles sized over 20 m diameter were detected in 100%' fit to Fig. 8b which shows that there are cases with D_max=20m?

This has been changed to '20 um or larger.'

p18l14: 'Figure 8c' does not exist.

Changed to Figure 8b

p19l4: particles

Changed

p19l10: 4643 has probably too many significant digits. I suggest 'around 4600'.

Done

p19l12: 'decreases' is maybe the wrong word here. I suggest 'is lower'.

Now reads, "is around a factor of ten lower than the total mass"

p19l15: 'PM2.5' in not defined. As it is used nowhere else, I suggest to just write 'the accumulation mode'.

Done

p20l16: 'as they only include iron when detectable as single-iron particles': As far as I understand this sentence, 'they' should be replaced by 'we' and 'detectable as single iron particles' by something like 'iron was the dominant component of a particle'.

Done

p22l17: There is one 'is' too much.

Changed

p22l28: 'coarse mode present' could be replace by 'coarse particles'.

Done

p22l29: 'so' could be removed.

Done

p23l1f: The sentence could be improved by removing 'same' and adding 'also' after 'mode'.

We thank the reviewer for this suggestion. After further consideration, we think that the sentence sounds better as it was and prefer to leave this unchanged.

p23l4: 'as dominate' is unclear.

'as' changed to 'which'

p23l7: 'RI' is not defined.

Now defined in Section 2.3.2, first occurrence of refractive index.

p23l10: Suggestion: 'The variability of the optical properties of dust in the SAL is probably mainly determined by ...'.

The wording here has been changed.

p23l14: 'the variability of the' should be inserted before 'optical'.

Done

p23l26: I suggest to write 'the variation of the SSA as function of composition, represented by k550', because this order is more logical and also better fits to Fig. 13 considering that the vertical axis usually shows the dependent variable (y=f(x)).

Done.

p24l1: 'optical property' could be replaced by 'SSA' to be more specific.

Done

p24l13ff: It is not clear how this fits to p13l29f where you write that you use the same size distribution (only number-weighted) for all components. In addition, you could consider a size dependency not only in case of external mixtures but also in case of internal mixtures.

Regarding number-weighting – see response to point above (p13l11).

Actually, we have calculated the size dependency of SSA for both the internal and external mixing cases, but perhaps this was not clearly described. These points can be seen as the large diamonds in Figure 13b. We have added a second panel to figure 14 to show the same values but for the full PSD, for the iterated RI, internal mixing RI and external mixing RI, and now discuss this in the text.

p24l21: 'In contrast to Fennec observations of the full PSD and associated optical properties over the Sahara,' could be removed. Maybe the information about the location could be added somewhere on line 22.

Done

p24l30: I suggest to delete 'during August 2015' and to add instead a new sentence like 'The flights were performed in August 2015 between Cape Verde and the Canary Islands.'.

Done

p25l7: 'to be' could be removed.

We prefer to leave the wording here.

p25l14: 'Deff for the SAL the mean (minimum, maximum) was' should be reformulated.

Done

p25l26: 'giant MBL mode particles' could be reformulated.

'MBL mode' has been deleted.

p25l30: calculate

Changed

p26l6: 'slightly lower' is an understatement because the 'base value' of the aspect ratio is 1.0. Then your value is only about half of the literature value.

We have deleted 'slightly.'

p26l6: 'and quartz' could be removed when considering Fig. 11.

We changed the wording to, "alumino-silicate particles dominated the composition at sizes above 0.5 µm, followed by quartz."

p26l28: 'was extremely scattering' should be replaced by 'was only very weakly absorbing'.

These two statements are essentially the same, therefore we leave the wording as it is.

p26l33: I suggest to start a new sentence after 'dust' and to write 'Particles larger than expected from sedimentation processes alone are found.'.

Done

Table 1: The reference style is not consistent.

Changed

Table 1: During 'SALTRACE' also the 'CAS-DPOL' instrument was used, measuring up to 50m (Weinzierl et al., 2017).

This has been corrected.

Table 2: The 'General Flight Aims and Conditions' do not very well fit to Table 1 of Marenco et al. Furthermore, you write 'b923', and Marenco et al. 'B923'. Maybe this could be more harmonized between both papers.

We have added 'CATS underflight' to Table 2 to b920, and changed 'SAVEX' to 'SAVEX-D' to align better with Marenco et al. Table 1. Other than this the details reflect the same information. We use 'b' lowercase as this is the official terminology used by FAAM for flight numbers, and will consider adjusting the text in Marenco et al.

Table 4: Negative latitude values don't make sense here.

These were errors and have been corrected.

Table 4: Longitude and latitude values for b923/b924 do not fit to Fig. 1.

These have been corrected.

Table 6: Sometimes you write 'D_eff' and sometimes 'd_eff'.

Corrected to $d_{eff}$.

Table 6: With 'derived RI' you mean the refractive index you iterated to fit the optical measurements? If yes, 'iterated RI' would be more specific.

Changed

Table 7: What means 'assuming internal mixing' here? In my understanding, the mixing state (internal/external mix) is only relevant for optical calculations but not for the derivation of the refractive index from filter samples.

In order to calculate the refractive index from the filter sample composition, only the internal mixing assumption can be used. In the external mixing assumption, the calculations step straight from the refractive index of the individual mineral components to their optical properties, and then to the optical properties of the total aerosol sample. They do not provide a refractive index for the total aerosol sample as an intermediate step. This is why the external mixing case cannot be shown on Figure 13b.

Figure 4: What is 'SLRs'?

As stated above, we now explain SLR in Section 2.1, 'Flight Patterns,' and use SLR throughout the manuscript instead of 'flight leg.'

Figure 5a: The dV/dlogD value for b924 (green) at the largest three size bins is more than ten times higher than the corresponding average value shown in Fig. 2. However, there are only five flights and SAL flight lags. How do these figures relate? How did you calculate the average in Fig. 2? See also my comment on p15l28.

As stated above, there are 14 flight legs shown in Figure 5a. Thus the weighting towards the b924 data is much less than 1 in 5. Please see response to reviewer 1 for the details of creating the campaign SAL average shown in Figure 2 and error processing, and associated text added to the manuscript.

Figure 6: You write '6(a)' while there is no '6(b)'.

a has been deleted

Figure 7: The dashed lines are not very well visualized and the description is missing in the legend (at a reference to p17l12 should be added).

We have added this to the legend, and plotted the dashed line with open circles to be clearer.

Figure 9: 'Aspect ratios histograms as a function of number fraction of particles' is not clear. You mean 'number fraction of particles as function of aspect ratio'?

Yes, we have changed this as suggested.

Figure 10c: It looks like there is a height dependence of the fraction of D>5m particles within the SAL. I think this height dependence should be briefly discussed in Section 3.3.

Thank you for this suggestion. We have added the following to this section:

"Additionally there appears to be a trend with altitude shown in Figure 10c: the mean mass fraction at d > 5 μm decreases steadily from 0.75 at the surface to 0.23 at 5 km altitude. A decrease is also evident in panel d with the largest fractions being found towards the bottom of the SAL (excluding the MBL). These decreases with dust mass as a function of altitude are somewhat in contrast to the homogeneous distribution of dust size throughout the SAL shown in Figure 8. This may be due to the data shown in Figure 10c coming from profiles rather than SLRs, such that more data is available, and also that although $d_{eff}$ represents the full size distribution, as such it is relatively insensitive to smaller changes in the coarse and giant particle concentration. Either way, there is clearly evidence of coarser dust particles being more prevalent towards the bottom of the SAL (and also in the MBL), indicating deposition processes occurring."

Figure 11: Relative 'n particles' for '10.0 to 40.0' shows no big difference between 'b920 R2' and 'b928 R2', so it is a bit unclear why you mention here 'B928 R2 (top right) contained giant mode MBL particles.' which should also be true for b920.

We refer to the wing probe measurements which show the substantial giant mode in the MBL for b928 R2 but not for b920 R2. This is now clarified in the caption.

Figure 11: The last sentence could be 'Errorbars are counting uncertainties.'.

Changed.

Figure 13: 'size-specific RI is used': Is this explained in the main text?

The text has been changed to be clearer here.

---

## Author Comment (AC3) · 9 Nov 2018

**Response to Reviewers**

**Anonymous Referee #3**

The study characterizes the dust properties during the beginning of trans-Atlantic transport of dust particles. It presents new airborne measurements of dust size distribution, composition, shape, and optical properties within the Saharan Air Layer (SAL) and the Marine Boundary Layer (MBL) taken during the AERosol Properties – Dust (AER-D) fieldwork campaign in August, 2015. In their 6 flights, the authors used wing-mounted optical particle counters and shadow probes to measure dust sizes between 0.1 and 100 m diameter, a nephelometer and an absorption photometer to measure dust optical properties, and an in-cabin filter collection system to collect dust samples.

The focus of the study is to highlight the presence and contribution of coarse and giant mode dust particles to the dust size distribution, mass loading, shape, composition, refractive indices and optical properties. The authors found that within the SAL, dust particles with diameter (D) greater than 20m are detected in 100% of the cases, and those with D>40m are detected about 36% of the cases. Of the dust particles detected, 14% of the masses are for dust particles with size D<2.5m, 60% for size D>5m, and about 10% for D>20m. In addition, the authors also found the following: the shape of the measured particle size distribution does not vary significantly between dust layers; the modal aspect ratios are in between 1.2 to 1.4; the real part of dust refractive index in both SAL and MBL is within 1.47 to 1.49, but the imaginary part is between 0.0012 - 0.003i in the MBL and between 0.0004 - 0.0005i within the SAL. They also found that the single-scattering albedo (SSA) at 550nm decreases in the SAL when the measured coarse and giant dust particles are included in the calculation. However, they concluded that the variability of the SSA is not controlled by the dust size distribution, but by the variability in dust composition, contrary to previous studies.

Observational datasets for the coarse and giant dust particles, reported in this paper, are very important to better constrain dust properties in climate models. Current climate models over-estimate the fine-mode dust particles and under-estimate the coarse-mode particles, leading to uncertainties in the estimation of dust optical properties. This is largely due to inadequate observational constrains, and only few similar measurements of size-resolved dust properties are publicly available, with few obtained during the summer time period. Hence, high-quality measurements with a wider particle size range, like those reported in this study, are needed.

We thank the reviewer for their comments, and are pleased that they consider our observations high-quality and important for better constraining dust properties in climate models. In response to the three main comments, we have shortened sentences for clarity as much as possible throughout the whole manuscript, provided a detailed response to the back trajectories point, and expanded on the applicability of Mie theory in this work. Details are given below.

The paper is generally well written, and I believe it also meets the ACP standards. I recommend it for publication, if the authors can address the following comments:

1. Reading through the paper, some parts of it are rather confusing. This is primarily because some of the sentences are too long, making the reading of the paper a bit tiring. The long sentences also sometimes obscure the point the author may want to pass across. I encourage the author to look more closely into each sentence, separating the long ones to multiple short sentences, where necessary. While few of these sentences are highlighted below, I cannot point to all the instances and I hope the author will do the due diligence in addressing this comment throughout the paper.

Pg 14 Lines 6-8, 14-16. Pg 15 Line 1-4. Pg 17 Line 1-4, 10-12. Pg 18 Line 22-26. Pg 20 Line 2-5. Pg 25 lines 16-19

In all cases apart from one, these sentences have been shortened. We have also been through the whole paper and shortened long sentences where possible.

2. Pg 6: The authors should provide a more objective assessment of the dust source areas. While HYSPLIT back-trajectory understandably are associated with uncertainty at the trajectory endpoints, it is still a reasonable method to determine the age of the dust particles, especially when the alternative is subjective. This is particularly useful for the dust particles in the SAL, where such trajectory can easily be estimated along a constant potential temperature surface, therefore avoiding possible influence of the convective events within the boundary layer. Doing it this way, may give a more close and objective approximation of the dust age, to which the SEVIRI images can eventually confirm. Free-tropospheric dust aerosols generally preserve their temperature for a considerable distance from the source region. Isentropic trajectories are therefore suitable above the boundary layer (e.g. Merrill et al., 1986).

From the HYSPLIT website (https://ready.arl.noaa.gov/HYSPLIT_traj.php), the figure below shows an example of the isentropic back-trajectory for flight #b932 starting on 20/Aug/2015 at 12Z for an arbitrary height of 2800 m above sea level. This height corresponds approximately to the highest extinction in your Fig. 4. The figure is a 3-day back-trajectory and it appears to suggest that the starting point after 3 days is approximately in the same area as suggested by SEVIRI in you Figure 1. This calculation can be repeated for different height within the SAL, and can also be combined with the SEVIRI images to give a more objective estimate of the dust sources, the age and the starting location.

In addition, the figure below uses the NCEP reanalysis dataset. It may be useful, however, to use a better quality meteorological dataset, like ERA-Interim with relatively higher resolution, to drive the HYSPLIT back-trajectories. ECMWF assimilates meteorological data from radiosondes that launch from few but important stations over north Africa. This may reduce the uncertainty even further, giving some more credence to the methodology.

We appreciate the reveiwer's efforts to investigate one of our cases with HYSPLIT. We have to point out, however, that the trajectory endpoint location was incorrect as the b932 dust investigated was centred at around 20W, 20N (see red flight track in Figure 1) rather than at the Cape Verde Islands themselves. Below we show a HYSPLIT back trajectory that was used as part of the analysis that in part led to the discussion in Section 2.2, using the correct location and time for the in situ sampling of b932. Three starting heights were chosen to cover the altitudes of the dust layer sampled. We also show two SEVIRI images indicating the two dust uplift times for this case.

It can be seen that the back trajectories do not capture the actual uplift location, times or transport path of the dust over the Sahara at all. The back trajectories would lead us to infer dust uplift times/locations of 00UTC on 19 Aug in central Mauritania at ~20N (blue) and ~22N (red), and at ~12UTC on 17 Aug in northern Chad. While in fact the SEVIRI images show that uplift occurred firstly near the Mali/Algeria border between 17 Aug 1000 UTC to 0100 UTC 18 Aug, and then again over Northern Mali at 1200-1400 UTC on 18 Aug. Therefore we believe that the uplift locations suggested by the back trajectories are incorrect, and that we have a better representation for these events using the SEVIRI imagery. This should not be interpreted as a negative appreciation of back trajectories in general.

Secondly, according to HYSPLIT timings, the dust would be inferred to be aged >36h (lower layer, red/blue) and >96h (upper layer, green). The SEVIRI imagery dust ages are 45-47 h for the Northern Mali uplift, and 58-73 h for the Mali/Algerian border uplift.

The transport pathway of the dust is clearly visible on the satellite images. The dust is transported from the Mali/Algeria border region to the northwest, before moving southeastwards to the coastal area. This pathway is not captured at all by the back trajectories.

The image below shows back trajectories calculated using the GDAS 0.5 degree meteorology dataset. We also ran the same case with the NCEP reanalysis data, and the results were similar. To our knowledge, ERA-Interim data is not available with the web-based HYSPLIT trajectory model.

[Figure]

[Figure]

As stated in Section 2.2, we had already examined all the HYSPLIT back trajectories for our dust cases, and compared them to the SEVIRI imagery. ("HYSPLIT back trajectories were also run, but are not used to determine source location or age here, since in every case they indicated a transport path differing from that shown by the SEVIRI imagery.") Occasionally the dust source location was similar to that from SEVIRI, even though the indicated transport pathway was different. However, the back trajectories also always showed either too fast or too slow transport, which would lead to an incorrect assignment of dust age. Additionally, another limitation of back trajectories is that they only indicate when an air mass nears the surface, but do not reflect potential uplift conditions (e.g. surface wind strength, soil conditions...).

We have rewritten the last paragraph of Section 2.2 to explain this more clearly.

We agree with the reviewer that back trajectories, following a constant potential temperature, can be a very useful tool for situations like dust in the SAL. However, for the large part of the time between dust uplift and sampling during AER-D, the dust was over the Sahara and within the Saharan Boundary Layer, which can extend up to 5 km altitude. The SEVIRI images show that convection and clouds were frequently present along the transport pathway of the dust. Therefore over the Sahara our dust events cannot be assumed to follow isentropic trajectories, and the difference between the HYSPLIT results and the SEVIRI imagery confirms this.

Several Fennec studies examined model biases over the Sahara, including the effects of unresolved cold pools. Garcia-Carreras et al. (2013) found a "crucial role of convective cold pools in explaining model tropospheric temperature bias," and suggest that, "the misrepresentation of moist convective processes can affect continental-scale biases, altering the West African monsoon circulation," even when additional radiosonde data is assimilated. Engelstaedter et al. (2015) found that models have errors in moisture distribution which "is likely to have consequences for simulations of Saharan thermodynamics and dust emissions caused by convection-driven cold pools." Roberts et al. (2017) find that ERA Interim winds are systematically underestimated over the Sahara. Despite the assimilation of a few radiosondes over the Sahara, models still struggle to resolve Saharan meteorology (Garcia-Carreras et al., 2013), which has knock on effects on dust back trajectories which rely on model fields. We have added citations of these papers in Section 2.2.

Given the challenges models still face in simulating clouds, convection and dynamics over the Sahara, and the clear visibility of the dust transport pathways in the SEVIRI imagery, the uncertainty and subjectivity in the SEVIRI methodology was perhaps overstated in Section 2.2. The last paragraph of Section 2.2 has been rewritten to try to better reflect this, as follows:

"The SEVIRI imagery is not able to give altitude-resolved information, and can be subjective, particularly when dust loadings are light, at low altitude or in a moist environment, making dust appear less pink and more difficult to identify (Brindley et al., 2012). This is more evident in the dust tracked for flights b932 and b934 where dust loadings were lower. This introduces a small level of uncertainty into both the source locations and dust ages, which we account for by giving generous error bars to the dust uplift times and source locations. HYSPLIT back trajectories (Draxler and Hess, 1998; Stein et al., 2015) were also run for the AER-D dust events. In only one of the five dust events was the dust source location similar to that observed in the SEVIRI imagery. In every case the back trajectories indicated a transport path and transport time different to that shown by the SEVIRI imagery. Although the SEVIRI methodology has its limitations, the back trajectory method results were clearly not compatible with the information from SEVIRI. Therefore back trajectories are not used to determine source location or age here. Additionally, another limitation of back trajectories is that they only indicate when an air mass nears the surface, but do not reflect potential uplift conditions (e.g. surface wind strength or soil conditions). It has been shown that models and reanalyses are currently unable to adequately represent convective events and winds over the Sahara and Sahel, particularly due to the challenges of representing cold pools. For example, Garcia-Carreras et al. (2013) examine the role of convective cold pools and suggest that "the misrepresentation of moist convective processes can affect continental-scale biases, altering the West African monsoon circulation." Many other publications have examined the misrepresentation of Saharan convective events (Marsham et al., 2011; Heinold et al., 2009; Sodemann et al., 2015; Trzeciak et al., 2017; Allen et al., 2015; Roberts et al., 2017; Engelstaedter et al., 2015). Since convective events are the drivers of dust uplift in all the AER-D cases we do not consider HYSPLIT back trajectories (with relatively low model resolution of half a degree) to be informative here due to the challenges the models face in representing Saharan circulation. Finally, we note that back trajectories are recommended to be used with caution for dust events over the summertime Sahara (Trzeciak et al., 2017)."

3. The authors should either carefully justify the application of the Lorenz-Mie theory for dust particles larger than 20m or use a more appropriate methodology for this size range. The manufacturer-provided size bin diameters were calibrated against polystyrene latex spheres, which the authors corrected to diameter of dust using Lorenz-Mie method (on PCASP and CDP). But Lorenz-Mie theory is only valid when the particle size is comparable to the wavelength (Bohren and Huffman, 1983). For coarse and giant dust particles with diameter

larger than 20 m, the application of Lorenz-Mie theory is no longer valid, and instead the geometric optics method may be useful (see Bi et al., 2009).

Firstly, although we process the entire CDP PSD using Mie theory, covering diameters up to around 50 microns, we only use the CDP data up to diameters of 20 microns, as at this point the CIP15 or 2DS data is given precedence. Therefore this is not an issue for the CDP data.

Secondly, we would like to point out that Mie theory is an exact solution for spheres of any size, and has no upper (or lower) limit, but converges to geometric optics for larger sizes and to Rayleigh scattering approximations at smaller sizes. (e.g. Petty, 2006). However, we assume that the reviewer is referring to the applicability to Mie code for the use of non-spherical particles. In this work we assume that the particles are spherical for optical property calculations. Sensitivity of SSA to the assumption of spherical particles was tested by Otto et al. (2009) and Johnson and Osborne (2011), who found that SSA changed by under 1% and 2% respectively when non-spherical particles were assumed. This is less than our error in SSA described in Section 2.3.1, and therefore we consider this an acceptable assumption. We have added this information and citations to the text, as well as a new methodology section 2.6 'Calculation of Optical Properties' to make this clear.

Petty, (2006) A first course in Atmospheric Radiation, Ch 12, Sundog Publishing, USA.

Specific Comments:

Pg 5, Line 7. Pg 7, Line 20. Pg 14, Line 4. Pg 16, line 25. Pg 18, line 2. Pg 21, line 15: The table numbers referenced here are wrong. Please check all other reference in the paper.

All table references have been corrected.

Pg 3, Line 9-10: Re-write for clarity.

Done

Pg 9, Line 8-9: I wonder if this difference between the "all-in" and the "center-in" is actually quantified. This text referenced here appear to be an assumption as suggested by the use of word "considered". If the latter is the case, I suggest this sentence should be re-written to clarify this point.

The sample area is actually adjusted depending on the particle size and whether all-in or center-in is used, since they both impact the effective array width and therefore the sample area, and hence the number concentration calculated (McFarquhar et al., 2017). We have added the sentence, "The sample area is adjusted for the effective array width, which is different depending on whether 'all-in' or 'centre-in' is used (McFarquhar et al., 2017), and therefore the calculated number concentrations account for this."

Pg 16, Line 24: There is no need for "6a", there is just one figure. Please also correct this in other places of the manuscript.

These have been corrected.

Pg 18 line 14: Figure 8c is not provided.

Corrected.

There is no definition of some acronyms – an example is the "SLR" acronym in the text or in Fig. 4. I suggest the author look through the paper and make sure every acronym is defined before use.

We have added more detailed flight manoeuver information and explanations to section 2.1, and also added acronym explanations as an appendix.

---

## Author Comment (AC4) · 9 Nov 2018

**Response to Reviewers**

**Anonymous Referee #4**

This manuscript presented a study of Saharan dust based on airborne observations made over the Eastern Tropical Atlantic near the western African coast. The measurements were targeted to characterize dust microphysical, chemical and optical properties, including size distribution, particle shape, mass loading, composition, refractive indices, and SSA. This study contrasted the dust properties in Saharan Air Layer and the marine boundary layer. The authors highlighted several important findings which will advance the current understanding and benefit later modeling studies. The manuscript is logically organized and well written. It is noted that the authors provided meticulous details about the instrument, data reduction, and uncertainty analysis. This reviewer believes that this manuscript shall be published after the authors considering a few suggested minor changes, which will not alter the major finds of this study.

We are pleased that the reviewer finds our results important and beneficial to modelling studies, and that they consider our data processing and uncertainty analysis meticulous. We thank the reviewer for their detailed comments which we hope have led to a clearer manuscript. We have responded to each comment in turn below. In particular, we thank the reviewer for their point about the relationship between SSA, k and $d_{eff}$, and have tried to clarify our aims here in confirming whether k or $d_{eff}$ is the main contributor to the variability in SSA. Full details are given below.

Minor Comments:

Page 4, Line 4: Please clarify that while light shadowing techniques are not impacted aerosol composition or Mie theory conversion issues, they still can be impacted by non-spherical particles.

Done

Page 5, Line 7: Change "Table 1" to "Table 2"

Done

Page 5, Line 8, please describe the AOD and clarify if the AOD is calculated over the dust layers. Please also make changes to the table caption so that it will be consistent with the text

The AOD is calculated across the entire aircraft profile, not just the dust. We have changed the text, the table does not need changing.

We added to Section 2.1, "The events sampled revealed accumulation mode AODs at 550 nm (see Section2.3.1)…"

We added to Section 2.3.1, "Accumulation mode AODs are calculated from aircraft profiles by integrating the scattering and absorption measurements between the minimum aircraft altitude (typically around 30 m above sea level) to the top of the profile (typically around 6 km). Therefore AODs represent both SAL and MBL aerosol."

Page 5, Line 26: Change "Figure 1b" to "Figure 1"

Changed

Page 6, Lines 18-19: The authors note that visually identifying and tracking dust plumes is subjective, difficult, and potentially error-prone. Would it be possible to instead obtain the underlying satellite data and apply an objective threshold?

As in the response to reviewer 3, we feel that we potentially over-emphasized the disadvantages of the SEVIRI method, and have changed the text accordingly.

Methods have been developed to automatically detect dust emission using the SEVIRI imagery, e.g. Ashpole and Washington (2012). To our knowledge none have been developed to track or back-track transported plumes. Applying a threshold to the underlying data may indeed have been possible, but would still be subject to variability due to underlying surface emissivity from surface temperature and surface type, altitude of dust in the atmosphere, obscuration by clouds.

Page 7, Line 5, please add a brief discussion on the choice using PSAP correction by Turnbull (2010) and difference between this correction and that by Virkkula, AS&T, 44:706-712, 2010

Turnbull (2010) reports on corrections necessary to the FAAM PSAP measurements based on the original work by Bond et al. (1999), and clarifications to this publication described in Ogren (2010), and clarifies any errors in calculations performed by Haywood and Osborne (2000).

Virkkula et al (2010) report corrections to the Virkkula et al (2005) publication, dealing with inconsistencies between a one and three-wavelength PSAP. The 2010 publication resolves the discrepancies. Since we employ a one-wavelength PSAP, not a three wavelength PSAP, we do not consider these publications in our corrections.

We have added the Bond et al. (1999) and Ogren (2010) references to the text.

Page 11, line 31, please provide a more quantitative criteria to define the word "dominant"

This has been changed to, "they were classified according to their dominant component type which made up the greatest oxide percentage."

Page 13, Line 1: Figure 8b is unrelated to SSAs; perhaps Figure 13b was intended?

This sentence has been deleted.

Page 13, Line 8: Please restate the rationale to hold the real part of the refractive index at 1.53, in the context that in Section 3.5, the real part is found to be 1.47-1.49 based on the filter sample composition.

We also tested the sensitivity of our results to different real parts of the refractive index: 1.48 and 1.58. We found that effective diameter (resulting from varying the RI used to correct the PCASP and CDP data) changed by under 5%. For the optical properties calculated, scattering and absorption each changed by around 4%, but SSA changed by under 0.5% since typically both scattering and absorption changed in the same direction.

Also, for the reasons stated in Section 3.5, we believe the filters real refractive index results to be biased low.

We added, "Although a value of 1.53 for $m^{550}$ is higher than that produced by the filter sample composition results (Section 3.5, 1.47-1.49) the filters result is likely biased low due to the reasons discussed in Section 3.5. We also performed a sensitivity test to using $m^{550}$ of 1.48 and 1.58, and found that deff changed by up to 5% and SSA by under 1%."

Page 13, Lines 14-20: This information might be better suited to a table, which could also include the actual refractive index used for each substance.

Although we appreciate this suggestion, since the spectral refractive index data is used rather than just a value at 550 nm, this would not fit into a table easily.

Page 14, Line 4: Change "Table 3" to "Table 4"

Done

Page 16, line 25. Please provide a brief discussion on how the "best-fit" compare to observed volume size distribution and number size distribution and Change "Table 4" to "Table 5"

Differences between the effective diameter calculated with the best-fit lognormal curves and the observed PSDs are between 10-15%. This relatively large error derives largely from the size range between diameters 3

to 20 µm measured by the CDP. The fluctuating nature of the CDP PSD does not reconcile it to an easy fit with lognormal curves. This has been added to this paragraph.

Table reference changed.

Page 18, Line 2: Change "Table 5" to "Table 6"

Done

Page 18, Lines 8 and 11: These two statements regarding a potential decrease in dmax with height seem contradictory. Please clarify to make them consistent.

This has been changed to, "There is no clear trend of $d_{max}$ decreasing with altitude."

Page 18, Line 14: There is no Figure 8c

Changed to 8b.

Page 19, Line 14: Please supply a reference for the dust density value.

The value comes from Tegen and Fung (1994), and is used by Woodward (2001) and Bellouin et al. (2011) as the default density value for dust aerosol in the Met Office Unified Model and also the radiation scheme within the model. References have been added.

Page 21, Line 15: Change "Table 6" to "Table 7"

done

Page 23, Line 26 (and Figure 13b): Some readers may wonder if finding a good agreement between the imaginary part of the refractive index and the SSA is expected, given the relationship between k, absorption, and SSA. The authors should consider the significance of confirming that the relationship exists in this case.

We are not trying to show or confirm the existence of a relationship between k, absorption and SSA. We acknowledge that this is an established relationship. However, the PSD can also impact the SSA. Therefore the SSA can be influenced by several factors, and our aim is to investigate which factors dominate the variability of the SSA. During Fennec, for example, the variability in the PSD was the dominant controller of the SSA, rather than k. Contrastingly, during AER-D we found that the PSD did not vary much, and therefore variability in k dominated the variability in SSA.

We have rewritten this paragraph to try and better explain this finding and its significance, and also in the conclusion. This paragraph now reads:

"Contrastingly to Figure 13a, Figure 13b shows that the SSA variability was strongly influenced by the variability in composition. This is the case for both accumulation mode observations of SSA, and for the full size distribution. It is not surprising that variability in $k^{550}$ influences absorption and therefore SSA. However, the SSA can be influenced by several factors, including the PSD. Our aim is to investigate which factors influence the variability of the SSA. Therefore it is notable that there is so little variation in the PSD during AER-D that the composition (or $k^{550}$) is the main factor contributing to the variability of the SSA. This finding is notably the opposite from that found during Fennec, where the size distribution was the dominant controller of optical properties. Liu et al. (2018) show that hematite content is important in the ICE-D/AER-D samples as a controlling factor on optical properties. Moosmuller et al. (2012) and Caponi et al. (2017) also show dependencies of refractive index on iron content. This is consistent with our findings that the calculated refractive index from the filter samples is strongly influenced by the iron content and its absorbing properties. It appears that over the Sahara, variations in the PSD (affected by dust age) have an important impact on SSA, while over the ocean the impacts of composition (perhaps either by chemical aging or by sampling dust from different sources) become more important."

Page 24, Line 20: Change "Table 5" to "Table 6"

Done

Figure 4: This figure suggests the flight b924 and b934 did not have extensive sampling in MBL, please make changes in text accordingly

We are not sure why the reviewer thought this. Figure 4 shows aircraft profile observations. Each flight included one SLR in the MBL. We have added information to Section 2.1 (Flight Patterns) to further explain the aircraft manoeuvres. See also the response to reviewer 3.

Figure 6: The blue shading was very faint on my screen. Perhaps a darker shade, or even hatching, could be used instead. Also, as there are no other parts to this figure, "Figure 6(a)" should be changed to "Figure 6" (as well as in the associated text).

'a' has been removed. The shading appears fine on our screen and print-offs. We will closely monitor the readability of this figure in proof-reading, where the original eps files will be used rather than a convert to .png contained in the MS Word document.